# Dr Jekyll and Mr Hyde:
# The Strange Case of Off-Policy Policy Updates

**Romain Laroche**[*]
Microsoft Research Montréal, Canada

**Rémi Tachet des Combes**[*]
Microsoft Research Montréal, Canada

## Abstract

The policy gradient theorem states that the policy should only be updated in states that are visited by the current policy, which leads to insufficient planning in the *off-policy* states, and thus to convergence to suboptimal policies. We tackle this planning issue by extending the policy gradient theory to *policy updates* with respect to any state density. Under these generalized policy updates, we show convergence to optimality under a necessary and sufficient condition on the updates' state densities, and thereby solve the aforementioned planning issue. We also prove asymptotic convergence rates that significantly improve those in the policy gradient literature. To implement the principles prescribed by our theory, we propose an agent, Dr Jekyll & Mr Hyde (J&H), with a double personality: Dr Jekyll purely exploits while Mr Hyde purely explores. J&H's independent policies allow to record two separate replay buffers: one on-policy (Dr Jekyll's) and one off-policy (Mr Hyde's), and therefore to update J&H's models with a mixture of on-policy and off-policy updates. More than an algorithm, J&H defines principles for actor-critic algorithms to satisfy the requirements we identify in our analysis. We extensively test on finite MDPs where J&H demonstrates a superior ability to recover from converging to a suboptimal policy without impairing its speed of convergence. We also implement a deep version of the algorithm and test it on a simple problem where it shows promising results.

## 1 Introduction

Policy Gradient algorithms in Reinforcement Learning (RL) have enjoyed great success both theoretically [50, 43, 16, 36] and empirically [23, 38, 37, 27]. Their principle consists in optimizing an objective function $\mathcal{J}$ (the expected discounted return) through gradient steps, both being formally specified below [43]:

$$\mathcal{J}(\pi) \doteq \mathbb{E}\left[\sum_{t=0}^{\infty} \gamma^t R_t \middle| S_0 \sim p_0, A_t \sim \pi(\cdot|S_t), S_{t+1} \sim p(\cdot|S_t, A_t), R_t \sim r(\cdot|S_t, A_t)\right] \quad (1)$$

$$\nabla_\theta \mathcal{J}(\pi) = \sum_{s \in \mathcal{S}} d_{\pi,\gamma}(s) \sum_{a \in \mathcal{A}} q_\pi(s,a) \nabla_\theta \pi(a|s) \quad \text{and} \quad \theta_{t+1} \xleftarrow[proj]{} \theta_t + \eta_t \nabla_{\theta_t} \mathcal{J}(\pi_t), \quad (2)$$

using standard Markov Decision Process notations (recalled in App. A), where $\eta_t$ is the learning rate and $d_{\pi,\gamma}(s) \doteq \sum_{t=0}^{\infty} \gamma^t \mathbb{P}(S_t = s|\pi)$ the discounted state density, the policies $\pi \doteq \pi_\theta$ and $\pi_t \doteq \pi_{\theta_t}$ are implicitly parametrized by $\theta$ for conciseness, and $\xleftarrow[proj]{}$ denotes the projection onto the parameter space. We observe that the update established by the policy gradient theorem is proportional to the state density of the current policy. This is problematic, as it implies that no value improvement can be induced in *off-policy* states (*i.e.* states that are rarely visited). As a consequence,

---

[*]Equal contribution. Correspondence to: rolaroch@microsoft.com. All code available at http://aka.ms/jnh.

planning is inefficient in those states to the point of potentially compromising convergence to the optimal policy.

To address the planning issue in off-policy states, we generalize the policy gradient theory by considering the more general policy update:

$$U(\theta, d) \doteq \sum_{s \in \mathcal{S}} d(s) \sum_{a \in \mathcal{A}} q_\pi(s, a) \nabla_\theta \pi(a|s) \quad \text{and} \quad \theta_{t+1} \xleftarrow[proj]{} \theta_t + \eta_t U(\theta_t, d_t), \qquad (3)$$

where $d_t$ designates the state distribution on which the policy is updated. It does not have to match the distribution induced by $\pi$ and updates may therefore be off-policy. We note that $U(\theta, d)$ is not a gradient in the general case, hence the use of the term *update*. We prove in the direct: $\pi_\theta \doteq \theta$ and softmax: $\pi_\theta \stackrel{.}{\sim} \exp(\theta)$ parametrizations that following the exact policy updates induces a sequence of value functions $q_t \doteq q_{\pi_t}$ that is monotonously increasing, upper bounded, and thereby converges to a $q_\infty$. We then show that the condition "$\forall s \in \mathcal{S}, \sum_t \eta_t d_t(s) = \infty$" on the sequence $(d_t)$ is necessary and sufficient for $q_\infty$ to be optimal. Our result generalizes previous theorems to broader updates and milder assumptions [2, 22]. Finally, we significantly improve the existing asymptotic convergence rates. We show that $(q_t)$ converges to the optimal value i) exactly in finite time for the direct parametrization (previously in $\mathcal{O}\left(t^{-1/2}|\mathcal{S}||\mathcal{A}|(1-\gamma)^{-6}\right)$[2]), and ii) in $\mathcal{O}\left(t^{-1}|\mathcal{S}||\mathcal{A}|(1-\gamma)^{-2}\right)$ for the softmax parametrization, to be compared with $\mathcal{O}\left(t^{-1}|\mathcal{S}|^2|\mathcal{A}|^2(1-\gamma)^{-6}\right)$ [22].

Building on our theoretical results, we design a novel algorithm: Dr Jekyll and Mr Hyde (J&H). Its principle consists in training two independent policies: a pure exploitation one (Dr Jekyll) and a pure exploration one (Mr Hyde), and give control to either one for full trajectories. J&H's independent policies allow to record two separate replay buffers: one on-policy (Dr Jekyll's) and one off-policy (Mr Hyde's), and therefore to update J&H's models with any desired mixture of on-policy and off-policy updates. Beyond an algorithm, J&H introduces conditions based on our analysis that actor-critic algorithms should follow to properly plan off-policy. Furthermore, the separation of exploration and exploitation allows to stabilise the training of the exploitation policy while ensuring a full coverage of the state-action space through deep exploration [30].

We empirically validate our theoretical analysis and algorithmic innovation in both planning and RL settings. In the planning setting where we assume that $q_t$ is exactly known, we analyze the impact of the off-policiness of $d_t$ and conclude that, while it improves over classic policy gradient, the theoretical sufficient condition does not allow to converge in a reasonable amount of time in very hard planning tasks: we thereby recommend to enforce the stronger condition "$\forall s \in \mathcal{S}, \sum_t \eta_t d_t(s) \in \Omega(t)$". In the reinforcement learning domain, where $q_t$ must be estimated from the collected transitions, the off-policiness of J&H's policy updates allows by design to satisfy the recommendation as long as Mr Hyde is able to cover the whole state space. This leads J&H to significantly outperform all competing actor-critic algorithms in the hard planning tasks and to be competitive with gradient updates in simple planning problems. As a proof of concept, we also test J&H in a deep reinforcement learning setting and show that it outperforms various baselines on a simple environment.

The paper is organized as follows. Section 2 develops the policy update theory. Section 3 describes J&H and positions it with respect to the literature. Section 4 presents the experiments and their results. Finally, Section 5 concludes the paper. The interested reader may refer to the appendix for the proofs (App. B), the domains (App. C), and the full experiment reports (App. D, E, and F). Finally, the supplementary material contains the code for the experiments: algorithms and environments.

## 2 Theoretical analysis

First, we recall some background: policy gradient methods depend on a parametrization $\theta \in \mathbb{R}^{|\mathcal{S}| \times |\mathcal{A}|}$ of the policy. Like [2, 22], we will focus on the classic direct and softmax parametrizations:

- direct: $\pi(a|s) \doteq \theta_{s,a}$, for which $u_{s,a} = d_t(s) q_\pi(s, a)$ and the projection is on the simplex.

- softmax: $\pi(a|s) \doteq \dfrac{\exp(\theta_{s,a})}{\sum_{a'} \exp(\theta_{s,a'})}$, for which $u_{s,a} = d_t(s) \pi(a|s)(q_\pi(s, a) - v_\pi(s))$.

For both parametrizations, [2] proves that following the policy gradient $\nabla_\theta \mathcal{J}(\pi)$ from Eq. (2) eventually leads to the optimal policy in finite MDPs under the following assumptions and condition[2]:

> A1. The model $(p_0, p, r)$ is known.
>
> A2. The initial state-distribution covers the full state space: $\forall s \in \mathcal{S}, p_0(s) > 0$.
>
> C3. The learning rate is constant: $\eta_t = \frac{(1-\gamma)^3}{2\gamma|\mathcal{A}|}$ (direct) and $\eta_t = \frac{(1-\gamma)^3}{8}$ (softmax).

Those are stringent requirements. A1 implies that the values $q_\pi$ and state density $d_{\pi,\gamma}$ of the policy are known exactly. A2 is generally not satisfied in standard RL domains. Finally, verifying C3 makes learning slow to the point that it becomes impractical.

## 2.1 Convergence properties

In this section, we study, under A1, the convergence properties of the sequence of value functions $(q_t)$ induced by the update rule defined in Eq. (3). We note that $U(\theta, d)$ is not a gradient in general. Our first theoretical result is the monotonicity of the value function sequence $(q_t)$.

**Theorem 1** (**Monotonicity under the direct and softmax parametrization**). *Under A1, the sequence of value functions $q_t \doteq q_{\pi_t}$ and $v_t \doteq v_{\pi_t}$ are monotonously increasing.*

By the monotonous convergence theorem, Thm. 1 directly implies the convergence of $(q_t)$:

**Corollary 1** (**Convergence under the direct and softmax parametrization**). *Under A1, the sequence of value functions $q_t$ uniformly converges: $q_\infty \doteq \lim_{t\to\infty} q_t$.*

In order to go further and prove convergence to a global optimal value, we need to enforce an additional condition. Indeed, the update $U$ relies multiplicatively on $d_t$ which could be equal (or tend very fast) to 0 in some states, compromising the policy's ability to reach the optimal value. This argument is not only theoretical, it has been observed by many that policy gradient can get stuck in suboptimal policies, even with entropy regularization. We designed our chain domain to exhibit such behaviour (see Section 4.1).

Next, we show that $(q_t)$ converges to the optimal value function under some necessary and sufficient condition.

**Theorem 2** (**Optimality under the direct and softmax parametrization**). *Under A1, the following condition:*

> C4. *Each state $s$ is updated with weights that sum to infinity over time: $\sum_{t=0}^\infty \eta_t d_t(s) = \infty$,*

*is necessary and sufficient to guarantee that the sequence of value functions $(q_t)$ converges to optimality: $q_\infty = q_\star \doteq \max_{\pi \in \Pi} q_\pi$.*

*Proof sketch.* In both direct and softmax parametrizations, we assume there exists a state-action pair $(s, a)$ advantageous with respect to the state value limits $q_\infty$ and $v_\infty \doteq \lim_{t\to\infty} v_t$, that is: $\mathrm{adv}_\infty(s, a) \doteq q_\infty(s, a) - v_\infty(s) > 0$. We then prove that in this state $s$, the policy improvement yielded by the update is lower bounded by a linear function of the update weight $\eta_t d_t(s)$. Both parametrizations require different proof techniques and are dealt with in different theorems. Summing over $t$, we notice that this lower bounding sum diverges to infinity, which contradicts Corollary 1. We therefore infer that there cannot exist such a state-action pair, which allows us to conclude that no policy improvement is possible, and thus that the values are optimal: $q_\infty = \max_{\pi \in \Pi} q_\pi$.

For the necessity of C4, we show that the parameter update is upper bounded in both parametrizations by a term linear in the action gap. By choosing the reward function adversarially, we may set it sufficiently small so that the sum of all the gradient steps is insufficient to reach optimality. $\square$

Thm. 2 is impactful along five dimensions.

Practice of on-policy undiscounted updates: As Thm. 2 is applicable to any distribution sequence, it allows us in particular to consider the practice of using on-policy undiscounted updates: $d_t \doteq d_{\pi_t, 1}$.

---

[2]An assumption A# is a requirement on the environment or the application setting, while a condition C# is a requirement that may be enforced by a dedicated algorithm in any environment or setting.

Thus, it resolves a longstanding gap between the policy gradient theory and the actor-critic algorithm implementations [12, 7, 51, 17, 1, 19, 41]. This mismatch and its lack of theoretical grounding were identified in [45]. Later, [26] proved that the practitioners' undiscounted updates are not the gradient of any function and may be strongly biased under state aliasing. Recently, [53] studied the practical advantage of the undiscounted updates from a representation learning perspective. C4 encompasses both standard policy gradient and the undiscounted update rule: convergence to optimality of both is guaranteed as long as C4 is verified. Our analysis thus shows that the convergence properties of policy gradient and undiscounted updates require the same set of assumptions and conditions. Conversely, it is possible to prove convergence to sub-optimal policies of either when C4 is violated. In other words, Thm. 2 allows to reduce the study of specific algorithms to whether they verify C4.

Experience replay and off-policy updates: It also justifies the use of an experience replay for the actor, a trick also widely used in the literature to distributed the training over several agents [24, 34, 49, 14, 35]. Furthermore, while widely overlooked in the literature, we prove that off-policy updates have an even higher impact, since they are necessary to guarantee the convergence to optimality[3]. We leverage this discovery to introduce new design principles and a novel algorithm in Section 3.

Policy gradient theory: Next, Thm. 2 generalizes previous optimality convergence results along two axes. The initial state distribution $p_0$ is not required to cover the state space anymore (aka A2). A2 is unrealistic in most applications and cannot be enforced by an algorithm. Relaxing it is an important open problem [2], we disprove it in the full class of MDP. However, it might be possible to find a class of MDPs larger than the one verifying A2 where policy gradient also converges. Additionally, the condition on our learning rates is also much more flexible than C3.

Density vs. policy regularization: In contrast, C4 can be controlled by a dedicated algorithm. This theoretical result promotes the principle of density-based regularization [21, 32], at the expense of policy-based regularization that cannot guarantee that C4 is satisfied and sometimes fails to plan over the whole state space (our chain domain experiments in Section 4 illustrate it well).

Towards a generalized policy iteration theorem: Finally, by combining Thm. 2, which analyzes policy improvement, with a thorough theoretical analysis of the policy evaluation step (discussed in Section 3.1), we see a path towards results describing conditions on both components of generalized policy iteration that guarantee convergence to optimality [4]. Furthermore, Thm. 2 gives sufficient conditions on the policy improvement step. The generalized policy iteration theorem was conjectured in [42] but never proved.

## 2.2 Convergence rates

Next, we give convergence rates for both softmax and direct parametrizations:

**Theorem 3** (**Asymptotic convergence rates under the direct parametrization**). *With direct parametrization, under A1 and C4, the sequence of value functions $q_t$ converges to optimality in finite time:*

$$\exists t_0, \text{ such that } \forall t \geq t_0, \quad q_t = q_\star. \tag{4}$$

C4 is required for optimality and convergence rate contrarily to the softmax parametrization. We significantly improve over previous bounds in $\mathcal{O}\left(t^{-1/2}|\mathcal{S}||\mathcal{A}|(1-\gamma)^{-6}\right)$[2]. We wish to emphasize that Theorem 3 does not say anything about how large $t_0$ is, the rate is purely asymptotic.

**Theorem 4** (**Asymptotic convergence rates under the softmax parametrization**). *With softmax parametrization, under A1, C4, and A8:*

*A8. The optimal policy is unique: $\forall s, \ q_\star(s, a_1) = q_\star(s, a_2) = v_\star(s)$ implies $a_1 = a_2$,*

*the sequence of value functions $q_t$ converges asymptotically as follows:*

$$\exists t_0, \text{ such that } \forall t \geq t_0, \quad v_\star(s) - v_t(s) \leq \frac{8|\mathcal{A}|(v_\top - v_\perp)}{(1-\gamma)\min_{s \in supp(d_{\pi_\star, \gamma})} \delta(s) \sum_{t'=t_0}^{t-1} \eta_{t'} d_{t'}(s)}, \tag{5}$$

---

[3]Other papers proving convergence to optimality of policy gradient rely on A2, which implies C4.

*where $\delta(s) = v_\star(s) - \max_{a \in \mathcal{A}/\{\pi_\star(s)\}} q_\star(s, a)$ is the gap with the best suboptimal action in state $s$, $v_\top$ (resp. $v_\bot$) is the maximal (resp. minimal) value, and $\mathrm{supp}(d_{\pi_\star, \gamma})$ denotes the support of the distribution of the optimal policy.*

If $(d_t)$ satisfies the additional mild condition that the support of $d_\star$ is covered *on average*:

C5. $\forall s \in \mathrm{supp}(d_\star), \lim_{t \to \infty} \frac{1}{t} \sum_{t'=0}^{t} \eta_{t'} d_{t'}(s) \doteq e_\bot(s) > 0$,

then we obtain the following convergence rate in $\mathcal{O}(t^{-1})$:

$$\exists t_0, \text{ such that } \forall t \geq t_0, \quad v_\star(s) - v_t(s) \leq \frac{8|\mathcal{A}|(v_\top - v_\bot)}{(t - t_0)(1 - \gamma) \min_{s \in \mathrm{supp}(d_{\pi_\star, \gamma})} \delta(s) e_\bot(s)}. \quad (6)$$

C5 is implied by A2 and C3. Alternately, assuming $\eta_t \geq \eta_\bot > 0 \, \forall t$, it is verified by a uniform distribution in which case $e_\bot(s) = \frac{\eta_\bot}{|\mathcal{S}|}$, or by an on-policy distribution $d_{\pi_t}$ that converges to the distribution of some optimal policy $d_{\pi_\star}$ in which case $e_\bot(s) = \eta_\bot d_{\pi_\star}(s)$. Also, note that C4 is required for optimality, but not for convergence rates with respect to $q_\infty$. Thm. 4 establishes asymptotic rates in $\mathcal{O}\left(t^{-1}|\mathcal{S}||\mathcal{A}|(1 - \gamma)^{-2}\right)$, to be compared with $\mathcal{O}\left(t^{-1}|\mathcal{S}|^2|\mathcal{A}|^2(1 - \gamma)^{-6}\right)$ [22]. The gap is partially explained by dropping C3 and allowing any learning rate (see App. A.6 for a thorough comparison). It is possible, see Thm. 7 of App. B, to show that under A2, a condition on the learning rate, and a bounding condition on $d_t$ ($\exists d_\top \geq d_\bot > 0$ such that $\forall s, t, \, d_\top \geq d_t(s) \geq d_\bot$), we have: $t_0 \leq \mathcal{O}\left(\frac{|\mathcal{S}|}{(1-\gamma)^7} \frac{d_\top}{d_\bot} \frac{1}{\min_s \delta(s)}\right)$.

## 2.3 Related work

[11, 15, 52] study the off-policy, continuing setting paired with the average value objective. That objective is either expressed with a state visitation coming from the behavioral policy (the excursion objective) or from the trained policy (the alternative life objective). Its gradient can be computed exactly or approximated and used to optimize the objective. The weights assigned to each state during an update stem from the gradient computation (exactly as in the discounted return gradient we study). It is an interesting question, left for future work, to understand whether those weights have desirable properties from a convergence standpoint. In particular, it is possible that a condition akin to C4 exists in the continuing objective case.

From a proof technique perspective, [2] studies gradient ascent and, as a byproduct, can rely on standard optimization results: strong convexity and convergence of gradients to 0. We cannot do so as our updates are not gradients anymore. More precisely, Lemma C2 in [2] uses the strong convexity of the objective function to prove that following the gradient results in a value improvement (assuming a learning rate sufficiently small). We provide a more general (we do not need the condition on the learning rate) and, in our admittedly biased opinion, more elegant proof of that lemma in our Theorem 1. Second, their Lemma C5 uses (i) the convergence of the gradient to 0 (a standard result with gradient as-/des-cent), and (ii) the assumption that all states have a non-vanishing density to infer that, in the limit, the learnt policy does not assign any mass to states that have a non-zero advantage. Neither (i) nor (ii) hold in our setting, we had to leverage C4 instead.

# 3 Dr Jekyll and Mr Hyde: an actor-critic with convergence guarantees

## 3.1 Conditions for an actor-critic algorithm to converge to optimality

C4 states the necessary and sufficient condition for optimal planning with the exact model. In a reinforcement learning setting however, the model is not known; the policy must be learnt from samples collected in the environment. As a consequence, A1 cannot be made anymore, we need to rely on an exploration condition instead:

C6. Each state-action pair is explored infinitely many times: $\forall s, a, \lim_{t \to \infty} n_t(s, a) = \infty$,

where $n_t(s, a)$ is the count of samples collected for the state-action pair $(s, a)$ at time $t$.

In finite MDPs, C6 provides a necessary and sufficient condition for having an unbiased estimator of the value. As is customary, we call this estimator the critic and denote it $\mathring{q}_t$. C6 is necessary because

the required statistical precision can only be achieved when the number of samples tends to infinity; it is sufficient as guaranteed by many off-policy policy evaluation algorithms from the literature [31, 46]. Classic concentration bounds, such as Hoeffding's inequality, tell us that with $\mathcal{O}(1/\xi^2)$ samples in every state, a $\xi$-accurate critic can be achieved. Under C4 and C6, there thus exists a timestep after which updates based on $\mathring{q}_t$ are sufficiently close to the true ones for a continuity argument to show that the policy improves eventually[4] (as guaranteed by theory under A1).

Our results till now have assumed a finite MDP and models with sufficient capacity. In larger domains, those assumptions do not hold anymore, and therefore our theory does not apply. For instance, [26] proves that using the on-policy undiscounted update $d_{\pi_t,1}$ instead of the policy gradient can lead to highly suboptimal policies under state aliasing. Their counter-example applies to our policy updates as well. Nevertheless, we believe that state aliasing is a worst-case scenario and conjecture that it does not happen to neural networks thanks to their high expressive capacity. Also note that this concern with respect to distribution shift is general and could be formulated for any neural model, including supervised models, or in RL the purely value-based ones such as DQN that are frequently trained off-policy. The consensus in the literature is that neural models do not suffer too much from distribution shift as long as the testing set distribution is well covered by the training set. Since appropriate coverage of the state-action space is actually the final objective of our off-policy policy updates, we expect minimal impact on this dimension, though it does remain to be formally demonstrated. Our empirical results from Section 4.4 support this conjecture. The theoretical study of the function approximation setting is left for future work.

### 3.2 Our solution to enforce C4 and C6

**Enforcing C4**  C4 defines the necessary and sufficient condition for asymptotic convergence to optimality. However, we will see in Section 4.2 that in difficult planning tasks, C4 can be insufficient for convergence to happen in a reasonable amount of time. Below, we therefore introduce C4-s, a condition stronger[5] than C4, as well as two techniques for C4-s to be verified:

C4-s. $\forall s \in \mathcal{S}, \quad \lim_{T \to \infty} \frac{1}{T} \sum_{t=0}^{T} \eta_t d_t(s) \geq d_\perp(s) > 0.$

The first technique to enforce C4-s is to apply the approximate expected policy update [39, 20, 10]:

$$\widehat{U}(\theta, s) \doteq \sum_{a \in \mathcal{A}} \mathring{q}(s, a) \nabla_\theta \pi(a|s). \tag{7}$$

The expected policy update is deterministic in the sense that it does not require sampling actions. As there exists a deterministic optimal policy, this in turn implies that the learning rate $\eta_t$ does not need to satisfy the second Robbins-Monro condition: $\sum_t \eta_t^2 < \infty$, and thus may be constant $\eta_t \doteq \eta$.

The second technique to enforce C4-s is to ensure that the density of updates $d_t$ does not decay to 0 for any state $s$. This can be obtained by maintaining a constant proportion $o_t$ of off-policy actor updates in order to cover the whole state space. We propose to do so by recording two replay buffers: one with on-policy samples, *i.e.* samples collected with an exploitation policy, and another with off-policy samples, *i.e.* samples collected with an exploration policy.

To the best of our knowledge, this type of prioritized experience replay has never been used in this fashion, nor to this effect. We note that off-policy policy gradient is a concept that exists in the literature, but it refers to the application of policy gradients from batch samples [21]. Following the success of DQN [24], experience replays have also been used in actor-critic methods [49] which bears some resemblance to our suggestion. Finally, off-policy actor-critic with shared prioritized experience replay [34] has been applied to large-scale experiments with distributed agents [14, 35].

**Enforcing C6**  Many RL algorithms (actor-critic or purely value-based) ensure some form of exploration. They broadly form two groups: dithering exploration (*e.g.*, epsilon greedy, softmax, entropy regularization [47, 42, 13]), and deep exploration (*e.g.*, UCB, thompson sampling, density constraints [28, 3, 29, 32]). Strong cases have been made against dithering exploration, arguing in particular its inability to ensure visits to all states and therefore convergence to optimality [30].

---

[4]A formal proof would require dealing with several technical challenges due to the stochasticity of the value updates that break the monotonicity of the estimator accuracy. We leave it for future work.

[5]C4-s is identical to C5, but applied to the full state space $\mathcal{S}$ (it thus implies C5).

---

**Input:** Scheduling of exploration ($\epsilon_t$), of off-policiness ($o_t$) and of actor learning rate ($\mathring{\eta}_t$).

1: Initialize Dr Jekyll's replay buffer $\mathring{D} = \emptyset$, actor $\mathring{\pi}$, and critic $\mathring{q}$.       ▷ exploitation agent
2: Initialize Mr Hyde's replay buffer $\tilde{D} = \emptyset$, and policy $\tilde{\pi}$.       ▷ exploration agent
3: Set the behavioural policy and working replay buffer to Dr Jekyll's: $\pi_b \leftarrow \mathring{\pi}$ and $D_b \leftarrow \mathring{D}$.
4: **for** $t = 0$ to $\infty$ **do**
5:     Sample a transition $\tau_t = \langle s_t, a_t \sim \pi_b(\cdot|s_t), s_{t+1} \sim p(\cdot|s_t, a_t), r_t \sim r(\cdot|s_t, a_t)\rangle$.
6:     Add it to the working replay buffer $D_b \leftarrow D_b \cup \{\tau_t\}$.
7:     **if** $\tau$ was terminal, **then** $(\pi_b, D_b) \leftarrow (\tilde{\pi}, \tilde{D})$ w.p. $\epsilon_t$, $(\pi_b, D_b) \leftarrow (\mathring{\pi}, \mathring{D})$ otherwise.
8:     **if** Update step, **then**
9:         $\tau \doteq \langle s, a, s', r \rangle \sim \tilde{D}$ w.p. $o_t$, $\tau \doteq \langle s, a, s', r \rangle \sim \mathring{D}$ otherwise.
10:        Update Mr Hyde's policy $\tilde{\pi}$ on $\tau$.     ▷ *e.g.* with $Q$-learning trained on UCB rewards
11:        Update Dr Jekyll's critic $\mathring{q}$ on $\tau$.     ▷ *e.g.* with SARSA update
12:        Expected update of Dr Jekyll's actor $\mathring{\pi}$ in state $s$.     ▷ Eq. (7)
13:     **end if**
14: **end for**

---

Algorithm 1: Dr Jekyll & Mr Hyde algorithm. After initialization of parameters and buffers, we enter the main loop. At every time step, an action, chosen by the behavioral policy, is executed in the environment to produce a transition $\tau_t$ (line 5). $\tau_t$ is stored in the replay buffer of the personality in control (either Dr Jekyll or Mr Hyde, line 6). If the trajectory is done, the algorithm samples a new personality to be in control during the next one (line 7). Then, the updates of the models start (line 8). The updates for Mr Hyde's policy $\tilde{\pi}$ and Dr Jekyll's critic $\mathring{q}$ are underspecified, and may be any algorithm in the literature satisfying the exploration (for Mr Hyde) or unbiased (for Dr Jekyll) conditions. When ($\epsilon_t$) = 0, J&H amounts to on-policy expected updates from a single replay buffer.

In spite of these arguments (that we recall in App. A.5), there are still only very few actor-critic algorithms that realize deep exploration [9, 33]. We conjecture that this relates to the following observations: i) exploration involves a moving objective, hence a non-stationarity of the desired policy, ii) actor-critic algorithms have a structural inertia in their policy (in contrast to value-based methods that can completely switch policy when an action's value overtakes another's). As a consequence, a dual exploration-exploitation actor-critic algorithm takes a lot of time to switch from deep exploration to exploitation, and back, making it inefficient. To avoid this issue, we propose to train two policies, a pure exploration one and a pure exploitation one. As an added benefit, they will be used to constitute the on-policy and off-policy replay buffers introduced in the second technique above.

### 3.3 Dr Jekyll and Mr Hyde algorithm (J&H)

The objective of this section is to introduce a novel algorithm that satisfies conditions C4 (or C4-s) and C6 by design. To do so, we maintain a mixture of two policies:

- Dr Jekyll $\mathring{\pi}_t$ is a pure exploitation policy,
- Mr Hyde $\tilde{\pi}_t$ is a pure exploration policy.

At the beginning of a new trajectory, Mr Hyde $\tilde{\pi}_t$ (resp. Dr Jekyll $\mathring{\pi}_t$) is chosen with probability $\epsilon_t$ (resp. $1 - \epsilon_t$) and used to generate the full trajectory. Dr Jekyll $\mathring{\pi}_t$ is trained with the update rule of Eq. (3), where $d_t$ is defined by prioritized sampling over the experience replays, $q_t$ is replaced with an unbiased estimator $\mathring{q}$ of the value of $\mathring{\pi}_t$, and $\mathring{\theta}_t$ are the parameters of $\mathring{\pi}_t$. $\tilde{\pi}_t$ is a pure exploratory policy, designed to verify almost surely: $\forall (s, a) \in \mathcal{S} \times \mathcal{A}$, $\lim_{t \to \infty} n_{|\tilde{D}|}(s, a) \geq \tilde{d}_\perp > 0$,, where $n_{|\tilde{D}_t|}(s, a)$ is the count of samples $(s, a)$ in its experience replay. We note that a uniform policy satisfies this condition, but with a very small constant $\tilde{d}_\perp$ which compromises convergence in a reasonable time. Any deep exploration algorithm should be guaranteeing it[6]. In our finite MDP experiments, we implement $q$-learning with UCB rewards [3]: $\tilde{\pi}_t \doteq \operatorname{argmax}_{a \in \mathcal{A}} \tilde{q}_t$ where $\tilde{q}_t$ is trained to predict the expectation of the discounted sum of rewards: $\tilde{r}(s, a) \doteq \frac{1}{\sqrt{n_t(s,a)}}$ (see full Alg.

---

[6]Efficient exploratory algorithms are a challenge in large environments. The design of such algorithms is beyond the scope of this paper.

3 in App. E). In our deep learning experiments, Hyde is a Double-DQN [48] trained to maximize an exploration bonus based on Random Network Distillation [6] (Section 4.4).

The advantages of separating the exploration from the exploitation policy are the following. First, it is easier to specify the exploration/exploitation trade-off and get full control on the exploration requirements for condition C6: $\forall s, a, \sum_t d_{\pi_t}(s, a) = \infty$. Second, Jekyll's actor $\mathring{\pi}$ is not optimized under a moving objective which would otherwise induce a high level of instability on the policy. Third, one can define the on-policiness/off-policiness $o_t$ by recording two separate replay buffers $\mathring{D}$ and $\tilde{D}$, for trajectories respectively controlled by Dr Jekyll and Mr Hyde. This offers full control on condition C4/C4-s and on the asymptotic behaviour of $\sum_t \eta_t d_t(s)$.

These various observations lead to the design of the Dr Jekyll and Mr Hyde algorithm (J&H), formally detailed in Alg. 1.

## 4 Experiments

### 4.1 Domains

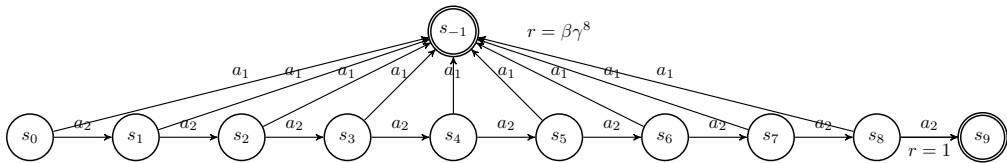

Figure 1: Deterministic chain MDP. Initial state is $s_0$. Reward is 0 everywhere except when accessing final states $s_{-1}$ and $s_9$. Reward in $s_{-1}$ is set such that $q(s_0, a_1) = \beta q_\star(s_0, a_2)$, with $\beta \in [0, 1)$.

**Chain Domain** The chain domain is designed to measure the ability of algorithms to overcome immediate rewards pushing the policy gradient towards suboptimal policies. In every state $s_k$, the agent has the opportunity to play action $a_1$ and receive an immediate reward of $\beta \gamma^{|S|-2}$, or to play $a_2$ and progress to next state $s_{k+1}$ without any immediate reward. A reward of 1 is eventually obtained when reaching state $s_{|S|-1}$. Fig. 1 represents a chain of size 10. We report the normalized performance: $\overline{\mathcal{J}}_\pi = \frac{\mathcal{J}_\pi - \mathcal{J}_\perp}{\mathcal{J}_\star - \mathcal{J}_\perp}$, where $\mathcal{J}_\star$ is the optimal performance and $\mathcal{J}_\perp \doteq q(s_0, a_1) = \beta \gamma^{|S|-2}$.

**Random MDPs** The random MDPs domain is designed to test the algorithms in situations where exploration is not an issue. Indeed, by its design of stochastic transition functions, random MDPs will have a non-null chance to visit every state whatever the behavioural policy. It is therefore a domain where we expect policy gradient updates to perform well, perhaps optimally, and hope that our modified updates still perform comparably. We reproduce the random MDPs environment published in App. B.1.3 of [18]. We report the normalized performance: $\overline{\mathcal{J}}_\pi = \frac{\mathcal{J}_\pi - \mathcal{J}_u}{\mathcal{J}_\star - \mathcal{J}_u}$, where $\mathcal{J}_\star$ is the optimal performance and $\mathcal{J}_u$ is the performance of the uniform policy. The full description of both domains is available in App. C.

### 4.2 Finite MDP policy planning (A1)

We test performance against time, learning rate $\eta$ of the actor, MDP parameters $|S|$ and $\beta$, off-policiness $o_t$, and policy entropy regularization weight $\lambda$, with both direct and softmax parametrizations, on the chain and random MDPs. The full report is available in App. D.

On the chain domain, we confirm that enforcing updates with $d_t \doteq o_t d_u + (1 - o_t) \frac{d_{\pi_t, \gamma}}{\|d_{\pi_t, \gamma}\|_1}$ including an off-policy component $o_t > 0$ on a uniform state distribution $d_u(s) \doteq \frac{1}{|S|}$ helps path discovery and policy planning, while on-policy updates fail at converging to the optimal policy, even with policy entropy regularization. We also observe that while $o_t \in \Omega(t)$ enforces C4, and therefore guarantees convergence to optimality, it may not happen in a reasonable amount of time, and a constant off-policiness is preferable. By sweeping over the chain parameters $\beta$ and $|S|$, we observe that on-policy updates are sensitive to both: even with a small $\beta = 0.1$, a reasonably sized chain $|S| = 15$ cannot be solved. In contrast, $o_t = 0.5$ converges fast to optimality even with $\beta = 0.95$ and $|S| = 25$.

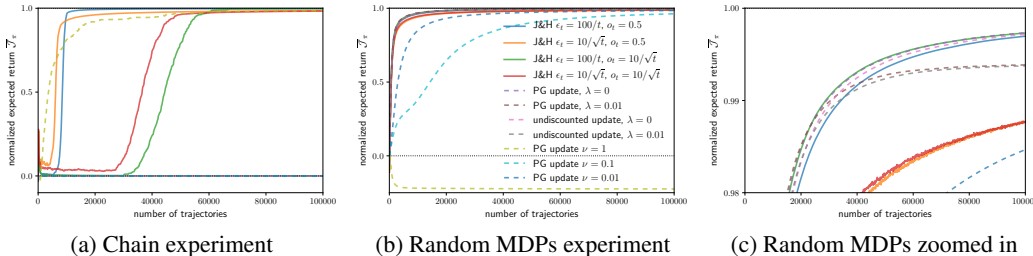

| | |
| (a) Chain experiment | (b) Random MDPs experiment | (c) Random MDPs zoomed in |

Figure 2: RL experiments: normalized expected return vs. number of trajectories (200+ runs).

On the random MDPs domain, we observe empirically that discounted and undiscounted updates perform well but oftentimes stagnate at 99% of the optimal performance; $d_t$ with an off-policy component performs better and gets even closer to optimality. Finally, we note that purely uniform updates: $o_t = 1$ slow down training in the random MDPs experiment. We conclude that it is best to include both on-policy and off-policy components in $d_t$. These experiments also allow us to observe the biased convergence implied by policy entropy regularization.

### 4.3 Finite MDP reinforcement learning experiments

With the softmax parametrization, we tested J&H with various scheduling for $\epsilon_t$ and $o_t$ against on-policy (PG/undiscounted) updates, with/without policy entropy regularization (hyperparameter $\lambda$), and with/without UCB critic (UCB hyperparameter $\nu$). The full description of the algorithms implementations is available in App. E.1. Our policy planning experiments suggested that $o_t$ was best constant at $0.5$, the same holds in our RL experiments. Fig. 2a reveals that on-policy updates with/without policy entropy regularization cannot solve the chain experiment of size 10 with $\beta = 0.8$. In contrast, the task is solved by on-policy updates with a strong UCB bonus $\nu = 1$ for the critic, and by all J&H implementations. Concerning J&H scheduling hyperparameters, we observe that $o_t = 0.5$ allows to identify the optimal policy significantly faster, and $\epsilon_t = \frac{100}{t}$ is sufficient and allows to converge faster asymptotically than $\epsilon_t = \frac{10}{\sqrt{t}}$. Fig. 2b shows that all algorithms are able to solve the random MDP task. However, UCB critics with high $\nu \in \{0.1, 1\}$ (including the only on-policy setting that succeeded on the chain task) fail to do so properly: they eventually will, but explore too much within the experimental time. If we look more closely at the tight convergence of every algorithm on Fig. 2c, we observe that the policy entropy regularization incurs a bias and that off-policiness of updates of J&H does not slow down convergence compared to vanilla on-policy updates. Note that J&H with $\epsilon_t = \frac{10}{\sqrt{t}}$ performs worse only because we report the expected performance of the mixture of Dr Jekyll and Mr Hyde. The performance of Dr Jekyll alone is comparable, and even slightly better than vanilla on-policy updates (see Fig. 9b in App. E).

For completeness, we also test performance against time, learning rates of the actor $\eta$ and critic $\eta_c$, MDP parameters $|\mathcal{S}|$ and $\beta$, off-policiness $o_t$, exploration $\epsilon_t$, and critic initialization $q_0$ with the softmax parametrization on the chain and random MDPs domains. See App. E for the full report.

### 4.4 Deep reinforcement learning experiments

To conclude our empirical evaluation, we implemented a deep version of J&H, as described in Algorithm 1, by parametrizing the agents using deep neural networks (see App. F for full details).

Dr Jekyll is a policy network, with a standard architecture, trained using the updates described in Eq. (7). As previously mentioned, any exploration algorithm can be used for Mr Hyde. In our experiments, we chose Random Network Distillation [6, RND] to generate exploration bonuses. Two networks, one random, the target, and a second one, the predictor, are used to assess the novelty of an observed state via the distance between the output of the target and the output of the predictor on that state. Each time a given state is evaluated, the predictor is trained to predict the output of the target. The more a state is seen, the smaller the prediction error will be. The error is used as a reward signal for Mr Hyde, a standard Double-DQN [48, DDQN] trained to maximize it. By doing so, Mr Hyde is incentivized to explore parts of the state space that have not been visited much yet.

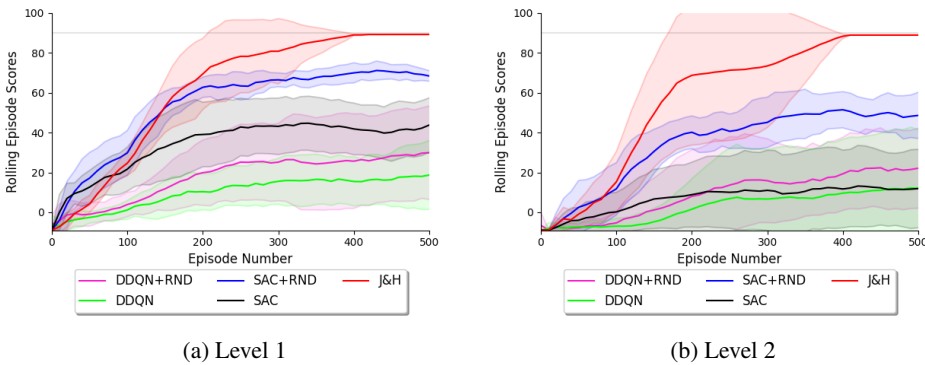

|  (a) Level 1 | (b) Level 2 |

Figure 3: Deep RL experiments: score vs. number of training episodes, averaged over 10 seeds.

We train J&H on a version of the Four Rooms environment [44], a 15x15 grid split into four rooms (see App. F for the exact layout). The agent, placed at random initially, needs to navigate to a fixed goal location, where it is granted a positive reward. For each step taken in the environment, the agent incurs a small negative reward. We consider two levels for the task. In level 1 (the original version of the game), the initial state distribution covers the entire state space, which corresponds to A2 being verified. To make exploration harder, we also consider an initial state distribution that does not contain any state from the room where the goal is located, and call that task level 2. We compare J&H to DDQN and Soft-Actor Critic [13, SAC]. For fair comparison, we also train these agents on the environment rewards augmented with the RND rewards used to train Mr Hyde, represented by the curves labelled DDQN+RND and SAC+RND.

Results can be found in Fig. 3. On level 1, we see that J&H reaches quite fast the maximal score of 90, The baselines do not perform as well and fail to converge to the optimal policy. Interestingly, even with A2 verified, adding an RND bonus led to increased performance. We also notice that overall SAC outperforms DDQN. The same observations can be made on level 2. We note that on some seeds, J&H takes more time to learn the optimal policy, leading to visible plateaus and temporary high variance. However, a powerful property underlined by our experiments and offered by the decoupling of exploration from exploitation is the ability of J&H, unhindered by dithering and/or conflicting reward signals, to converge to optimality. Comparatively, the baselines get stuck in a mixed exploration/exploitation behavior and fail to reach the maximum score.

## 5    Contributions and limitations

Contributions: We study a planning issue in actor-critic algorithms and tackle it by extending the policy gradient theory to *policy updates* with respect to any state density. Under these generalized policy updates, we show convergence to optimality under a necessary and sufficient condition on the updates' state densities. We also significantly improve previous asymptotic convergence rates. We implement the principles prescribed by our theory in a novel algorithm, Dr Jekyll & Mr Hyde (J&H), with a double personality: Dr Jekyll purely exploits while Mr Hyde purely explores. J&H's independent policies allow to record two separate replay buffers: one on-policy (Dr Jekyll's) and one off-policy (Mr Hyde's), and therefore to update J&H's models with a mixture of on-policy and off-policy updates. Beyond J&H, we define conditions for actor-critic algorithms to satisfy the requirements from our analysis. We extensively test J&H on finite MDPs and deep RL where it demonstrates superior planning abilities, and at least comparable asymptotic rates, to its competitors.

Limitations and future work: On the theory side, our non-asymptotic convergence rates are limited to the softmax and require A2 and C3. Although we are convinced our arguments hold, we did not develop the formal convergence proof in the RL setting (see second paragraph of Section 3.1). Finally, our theory does not tackle function approximation. In that context, we believe techniques from [2] could be useful. On the empirical side, we wish to emphasize that our deep RL experiments are to be considered as a proof of concept of the relevance of J&H's principles in that setting. A thorough study, encompassing more environments and baselines, potentially both in discrete and continuous actions spaces, is left for future work.

## Acknowledgments and Disclosure of Funding

We would like to thank Shangtong Zhang for our fruitful discussions and Alessandro Sordoni for helping us presenting and organizing the paper.

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
