# A MDP notations and basic knowledge on policy gradients

## A.1 General notations

- $\Delta_{\mathcal{X}}$ is the simplex over set $\mathcal{X}$.
- $|\mathcal{X}|$ is the cardinality of set $\mathcal{X}$.
- $X \sim d$ means that random variable X is sampled from distribution $d$.
- $\mathbb{E}[f(X)|X \sim d]$ denotes the expectation of $f(X)$ when random variable $X$ is sampled from distribution $d$.
- $\mathbb{P}[X > x|Y = y]$ denotes the probability that random variable $X$ is greater than $x$ conditioned to the fact that random variable $Y$ equals $y$.
- Subscript $\top$ (resp. $\bot$) denotes the maximal (resp. minimal) value in a range. For instance, $r_\top$ denotes the maximal reward.
- $f(\mathcal{X})$ denotes the sum of $f(x)$ over set $\mathcal{X}$: $f(\mathcal{X}) \doteq \sum_{x \in \mathcal{X}} f(x)$.

## A.2 Markov Decision Processes

A Markov Decision Process (MDP) is a tuple $m = \langle \mathcal{S}, \mathcal{A}, p, p_0, r, \gamma \rangle$, where:

- $\mathcal{S}$ is the set of states,
- $\mathcal{A}$ is the set of actions,
- $p(s'|s, a) : \mathcal{S} \times \mathcal{A} \to \Delta_{\mathcal{S}}$ is the probability of accessing state $s'$ after performing action $a$ in state $s$,
- $p_0 \in \Delta_{\mathcal{S}}$ is the initial state probability,
- $r(s, a) : \mathcal{S} \times \mathcal{A} \to \mathbb{R}$ is the (possibly stochastic) reward function,
- and $\gamma$ is the discount factor.

A stochastic policy $\pi(a|s) : \mathcal{S} \to \Delta_{\mathcal{A}}$ determines the probability of performing action $a$ in state $s$ and therefore determines the behaviour of an agent. The objective of a planning algorithm is to maximize the expected return:

$$\mathcal{J}(\pi) \doteq \mathbb{E}\left[\sum_{t=0}^{\infty} \gamma^t R_t \,\middle|\, S_0 \sim p_0, A_t \sim \pi(\cdot|S_t), S_{t+1} \sim p(\cdot|S_t, A_t), R_t \sim r(\cdot|S_t, A_t)\right]. \quad (8)$$

## A.3 Values and advantages

More generally, the state (resp. state-action) value function $v_\pi : \mathcal{S} \to \mathbb{R}$ (resp. $q_\pi : \mathcal{S} \times \mathcal{A} \to \mathbb{R}$) is the expected return starting from state $s$ (resp. after performing action $a$ in state $s$), when following policy $\pi$ afterwards:

$$v_\pi(s) \doteq \mathbb{E}\left[\sum_{t=0}^{\infty} \gamma^t R_t \,\middle|\, S_0 = s, A_t \sim \pi(\cdot|S_t), S_{t+1} \sim p(\cdot|S_t, A_t), R_t \sim r(\cdot|S_t, A_t)\right], \quad (9)$$

$$q_\pi(s, a) \doteq \mathbb{E}\left[\sum_{t=0}^{\infty} \gamma^t R_t \,\middle|\, S_0 = s, A_0 = a, A_t \sim \pi(\cdot|S_t), S_{t+1} \sim p(\cdot|S_t, A_t), R_t \sim r(\cdot|S_t, A_t)\right]. \quad (10)$$

We have several notorious identities, for instance:

$$\mathcal{J}(\pi) = \mathbb{E}\left[v_\pi(S)|S_0 \sim p_0\right], \quad (11)$$
$$v_\pi(s) = \mathbb{E}\left[q_\pi(s, A)|A \sim \pi(\cdot|s)\right], \quad (12)$$
$$q_\pi(s, a) = \mathbb{E}\left[R + \gamma v_\pi(S')|S' \sim p(\cdot|s, a), R \sim r(\cdot|s, a)\right]. \quad (13)$$

We also define the advantage of an action $a$ (resp. a policy $\pi$) over a value $v$ in a given state $s$:

$$\text{adv}_v(s, a) \doteq \mathbb{E}\left[R + \gamma v(S')|S' \sim p(\cdot|s,a), R \sim r(\cdot|s,a)\right] - v(s) \qquad (14)$$

$$\text{resp.} \quad \text{adv}_v(s, \pi) \doteq \mathbb{E}\left[\text{adv}_v(s, A)|A \sim \pi(\cdot|s)\right]. \qquad (15)$$

By extension, we talk about advantage over a stochastic policy and write $\text{adv}_\pi$ to denote the advantage over the value induced by this stochastic policy $\text{adv}_\pi \doteq \text{adv}_{v_\pi}$. For notational simplicity, we write $v_t, q_t, \text{adv}_t$ instead of $v_{\pi_t}, q_{\pi_t}$ and $\text{adv}_{\pi_t}$. A policy $\pi'$ is said to be advantageous over a value or another policy if its advantage is positive in all states $s \in \mathcal{S}$. During our analysis, we will constantly use the policy improvement theorem:

**Theorem 5.** *(Policy Improvement Theorem, see, e.g., Eq 4.7 & 4.8 in [42])*
*If $\forall s \in \mathcal{S}$, $\pi'$ is advantageous over $\pi$:*

$$\text{adv}_\pi(s, \pi') \geq 0, then \, \forall s \in \mathcal{S}, v_{\pi'}(s) \geq v_\pi(s). \qquad (16)$$

*If the first inequality is strict in a given state $s$, so is the second inequality on $s$.*

### A.4 Densities and policy gradients

For a given stochastic policy $\pi$, we define its state visit density as:

$$d_{\pi,\gamma}(s) \doteq \sum_{t=0}^{\infty} \gamma^t \mathbb{P}\left[S_t = s|S_0 \sim p_0, A_t \sim \pi(\cdot|S_t), S_{t+1} \sim p(\cdot|S_t, A_t)\right] \qquad (17)$$

The canonical policy gradient is:

$$\nabla_\theta \mathcal{J}(\pi) \doteq \sum_s d_{\pi,\gamma}(s) \sum_a q_\pi(s,a) \nabla_\theta \pi(a|s). \qquad (18)$$

The update vector often used in practice is:

$$\sum_s d_{\pi,1}(s) \sum_a q_\pi(s,a) \nabla_\theta \pi(a|s). \qquad (19)$$

We consider a more general update vector

$$U(\theta, d) \doteq \sum_s d(s) \sum_a q_\pi(s,a) \nabla_\theta \pi(a|s),$$

where $d : \mathcal{S} \to \mathbb{R}^+$ may be any positive function over $\mathcal{S}$.

The policy gradient methods depend on a parametrization $\theta \in \mathbb{R}^{|\mathcal{S}| \times |\mathcal{A}|}$ of the policy. In our finite MDPs theory, we focus on the two most common ones:

- direct: $\pi(a|s) \doteq \theta_{s,a}$ with update: $u_{s,a} = d(s)q_\pi(s,a)$ and projection on the simplex.

- softmax: $\pi(a|s) \doteq \dfrac{\exp(\theta_{s,a})}{\sum_{a'} \exp(\theta_{s,a'})}$ with update: $u_{s,a} = d(s)\pi(a|s)\text{adv}_\pi(s,a)$.

### A.5 Limits of on-policy updates with and without policy entropy regularization

The problem with on-policy updates (updates that are issued from a density over the state visitation of the current policy) is the conflicting objective of exploring and converging. If we want to converge, then mechanically the updates on off-density states will dry out. Let us assume that, in order to discover a better policy, the agent needs to go off-policy from state $s$, and that the state $s'$ where there is a positive advantage that would lead to a policy flip is actually encountered $h$ timesteps after going off-policy. If we assume that exploration is performed through dithering (typically policy entropy regularization in actor-critic algorithms), ie with a small exploration probability $\epsilon_t$ to take the action leading to $s'$, then state $s'$ will be visited only with probability $\epsilon_t^h$. As a consequence, to ensure that $\sum_t d_{\pi_t}(s) = \infty$, as we show to be necessary in Thm. 2, $\epsilon_t$ would need to decrease in $\Omega(t^{-1/h})$ which is very slow (and $h$ would be a hyperparameter strongly dependent on the environment). The overall issue comes from the myopia of exploration, a problem already observed in other works that promote deep exploration such as [30].

Only a handful of works have investigated deep exploration in actor-critic methods [6, 9]. We believe that they face difficulties due to the inertia of the policy network. Indeed, the policy gradient requires

on-policy updates which implies that the policy network must constantly adapt when switching from an exploration strategy to an exploitation strategy and back, which can be time-consuming and inefficient. Our fix consists in splitting the exploration and exploitation behaviour into two policies: Mr Hyde and Dr Jekyll. Mr Hyde performs pure deep exploration and will eventually collect, for each state-action, transitions in the replay buffer in a sufficient amount to obtain statistical significance. Dr Jekyll purely exploits what it learnt from a mixture of on-policy and off-policy policy updates, the latter granting a faster adaptation to the off-policy discoveries made during exploration.

### A.6 Comparison of bounds established in Thm. 4 with previous bounds

[22] established (to the best of their and our knowledge) the first convergence-rate result for softmax policy gradient for MDPs. As far as we know, there has not been others since. We start by recalling the differences between our setting and theirs:

- They have an all-time upper bound of the sub-optimality $\mathcal{J}(\pi_\star) - \mathcal{J}(\pi_t) \leq \epsilon$, while we only provide an asymptotic result of the form: $\exists t_0$, such that $\forall t > t_0$, $\mathcal{J}(\pi_\star) - \mathcal{J}(\pi_t) \leq \epsilon$.

- Our result is more general in the sense that it is applicable to any policy update scheme satisfying C4, while theirs only apply to policy gradient in settings (A2 and C3) that guarantee C4 and C5.

- Our theorem assumes A8 (see Rem. 1), requiring that there exists a unique optimal policy (*i.e.* $\nexists$ $(s, a_1 \neq a_2) \in \mathcal{S} \times \mathcal{A}^2$, such that $v_\star(s) = q_\star(s, a_1) = q_\star(s, a_2)$). Although [22] states that their approach works with multiple optimal policies, they did not prove it formally and the extension of their proofs from a unique optimal policy to multiple ones is not clear to us. It seems that they would face the same kind of difficulties we have (basically a not well-defined action gap with moving $q_t$). We discuss the limitations with respect to A8 in more details in Rem. 1 of App. B.2.3.

Now we recall their theorem under our notations:

**Theorem 6** (Thm. 4 of [22]). *Assuming A1, A2, C3, and A8 hold, then for all $t \geq 1$:*

$$v_\star(s) - v_t(s) \leq \frac{16|\mathcal{S}|(r_\top - r_\perp)}{t(1-\gamma)^6 \inf_{s \in \mathcal{S}, t \geq 1} \pi_t(\pi_\star(s)|s)^2} \cdot \left\| \frac{(1-\gamma)d_{\pi_\star,\gamma}}{p_0} \right\|_\infty^2 \cdot \left\| \frac{1}{p_0} \right\|_\infty, \quad (20)$$

*where $\pi_\star : \mathcal{S} \to \mathcal{A}$ is an arbitrary deterministic optimal policy.*

And we recall our result under A1, C5, and A8:

$$\exists t_0, \text{ such that } \forall t \geq t_0, \quad v_\star(s) - v_t(s) \leq \frac{8|\mathcal{A}|(v_\top - v_\perp)}{(t - t_0)(1-\gamma)\min_{s \in \text{supp}(d_\star)} \delta(s)e_\perp(s)}. \quad (21)$$

C5. $\forall s \in \text{supp}(d_\star), \lim_{t \to \infty} \frac{1}{t} \sum_{t'=0}^t \eta_{t'} d_{t'}(s) \doteq e_\perp(s) > 0,$

Now we look at each classic dependency:

- Time $t$: both are in $\mathcal{O}(t^{-1})$ but ours uses an offset $t_0$, hence its qualification of asymptotic convergence rate.

- State set size $|\mathcal{S}|$: we have a hidden dependency in $\min_{s \in \text{supp}(d_\star)} \delta(s)e_\perp(s)$ which is in $\Omega(1/|\mathcal{S}|)$, so our bound is in $\mathcal{O}(|\mathcal{S}|)$. Their theorem has an additional state set size dependency in $\|\frac{1}{p_0}\|_\infty$, so their bound is in $\mathcal{O}(|\mathcal{S}|^2)$.

- Action set size $|\mathcal{A}|$: we are in $\mathcal{O}(|\mathcal{A}|)$. Their theorem has an implicit action set size dependency in $\inf_{s \in \mathcal{S}, t \geq 1} \pi_t(\pi_\star(s)|s)$, so their bound is in $\mathcal{O}(|\mathcal{A}|^2)$.

- Horizon/discount factor $\frac{1}{1-\gamma}$: we have a possible hidden dependency in $v_\top - v_\perp$ (depending on e.g. the sparsity of rewards), which yields a bound in $\mathcal{O}((1-\gamma)^{-2})$, to be compared with their dependency in $\mathcal{O}((1-\gamma)^{-6})$. They pay the cost for setting through C3 the learning rate $\eta$ in $\mathcal{O}((1-\gamma)^{-3})$. We interpret the last $1-\gamma$ difference by the fact that we only deal with asymptotic rates and do not have to account for propagation through the MDP. Note that while the horizon appears in $\|\frac{(1-\gamma)d_{\pi_\star,\gamma}}{p_0}\|_\infty^2$, this term is larger than 1 (it is the sup of a ratio of distributions), hence the $1-\gamma$ cannot be taken out.

- State initialization distribution $p_0$: there lies the beauty of policy updates, we do not have any dependency on it. However, we do show that policy gradient will not converge to the optimal solution in general if A2 is not satisfied, and in that case off-policy updates must be used. The dependency in $p_0$ is particularly strong: possibly cubic with $\|\frac{1}{p_0}\|_\infty$. The worst case happens in $\|\frac{(1-\gamma)d_{\pi_\star,\gamma}}{p_0}\|_\infty^2$ when a state is rarely seen at initialization but visited a lot by optimal policies .

Finally, we discuss the constants that are difficult to compare:

- Divergence from the optimal policy during training $\inf_{s\in\mathcal{S},t\geq1}\pi_t(\pi_\star(s)|s)$: this is a un-controlled term that shows that policy gradient is very sensitive to preliminary convergence to sub-optimal solution. This term can easily take infinitesimal values (see *e.g.* Fig. 1d of [22]), and the dependency is squared.

- Optimal action gap $\delta(s)$: we depend on this constant because we first prove the convergence speed in policy and only then in value. If the action gap is very small in some states, resulting in a loose bound, it may be useful to use different proof techniques to avoid the convergence in policy step.

- State density of policy updates $e_\perp(s)$: with a constant factor $\kappa$ on uniform policy updates, $e_\perp(s) \doteq \frac{\kappa}{|\mathcal{S}|}$, and we accounted for this hidden dependency in the state set size comparison above.

# B Theoretical results

## B.1 Assumptions and conditions

A1 The model $p_0, p, r$ is known,

A2 The initial state-distribution covers the full state space: $\forall s \in \mathcal{S}, p_0(s) > 0$,

C3 The learning rate is constant: $\eta_t = \frac{(1-\gamma)^3}{2\gamma|\mathcal{A}|}$ (direct) and $\eta_t = \frac{(1-\gamma)^3}{8}$ (softmax),

C4. Each state $s$ is updated with weights that sum to infinity over time: $\sum_{t=0}^{\infty} \eta_t d_t(s) = \infty$,

C4-s. $\forall s \in \mathcal{S}, \quad \lim_{T \to \infty} \frac{1}{T} \sum_{t=0}^{T} \eta_t d_t(s) \geq d_\perp(s) > 0$.

C5. $\forall s \in \text{supp}(d_\star), \lim_{t \to \infty} \frac{1}{t} \sum_{t'=0}^{t} \eta_{t'} d_{t'}(s) \doteq e_\perp(s) > 0$,

C6. Each state-action pair is explored infinitely many times: $\forall s, a, \lim_{t \to \infty} n_t(s, a) = \infty$.

C7. There exists $d_\top \geq d_\perp > 0$ such that $\forall s, t, d_\top \geq d_t(s) \geq d_\perp$.

A8. The optimal policy is unique: $\forall s, \ q_\star(s, a_1) = q_\star(s, a_2) = v_\star(s)$ implies $a_1 = a_2$.

## B.2 Proofs

### B.2.1 Monotonicity and convergence

**Theorem 1** (**Monotonicity under the direct and softmax parametrization**). *Under A1, the sequence of value functions $q_t \doteq q_{\pi_t}$ and $v_t \doteq v_{\pi_t}$ are monotonously increasing.*

*Proof sketch.* First, we prove that, with both direct and softmax parametrizations, the update defined in Eq. (3) leads to a policy that is advantageous over the original one. By the policy improvement theorem, it means that the value function $v_t$ must be increasing, which also implies that $q_t$ is increasing. $\square$

*Proof.* The direct and softmax parametrizations are treated independently in Theorems 1a and 1b. $\square$

**Theorem 1a** (**Monotonicity under the direct parametrization**). *Under A1, the sequence of value functions $q_t \doteq q_{\pi_t}$ and $v_t \doteq v_{\pi_t}$ are monotonously increasing.*

*Proof.* Let us fix any $s \in \mathcal{S}$. We recall that

$$\pi_{t+1}(\cdot|s) = \mathcal{P}_{\Delta(\mathcal{A})}(\pi_t(\cdot|s) + \eta d(s)q_t(s, \cdot)). \tag{22}$$

By the definition of projections, this gives us:

$$\|\pi_t(\cdot|s) + \eta d(s)q_t(s, \cdot) - \pi_{t+1}(\cdot|s)\|^2 \ \leq \ \|\pi_t(\cdot|s) + \eta d(s)q_t(s, \cdot) - \pi_t(\cdot|s)\|^2 \tag{23}$$

$$\Longleftrightarrow \quad 2\langle\pi_t(\cdot|s) - \pi_{t+1}(\cdot|s), \eta d(s)q_t(s, \cdot)\rangle \ + \ \|\pi_t(\cdot|s) - \pi_{t+1}(\cdot|s)\|^2 \ \leq \ 0 \tag{24}$$

$$\Longleftrightarrow \quad \frac{1}{2}\|\pi_t(\cdot|s) - \pi_{t+1}(\cdot|s)\|^2 \ \leq \ \langle\pi_{t+1}(\cdot|s) - \pi_t(\cdot|s), \eta d(s)q_t(s, \cdot)\rangle \tag{25}$$

$$\Longrightarrow \quad 0 \ \leq \ \langle\pi_{t+1}(\cdot|s) - \pi_t(\cdot|s), \eta d(s)q_t(s, \cdot)\rangle. \tag{26}$$

Using the fact that $d(s) > 0$ and $\eta > 0$ and replacing the vectors by their values allows to infer:

$$\text{adv}_t(s, \pi_{t+1}) \doteq \sum_{a \in \mathcal{A}} (\pi_{t+1}(a|s) - \pi_t(a|s))q_t(s, a) \geq 0, \tag{27}$$

which is true for all $s$ and therefore allows to apply the policy improvement theorem, which concludes the proof of the monotonicity of the value through the direct parametrization policy update. $\square$

**Theorem 1b** (**Monotonicity under the softmax parametrization**). *Under A1, the sequence of value functions $q_t \doteq q_{\pi_t}$ and $v_t \doteq v_{\pi_t}$ are monotonously increasing.*

*Proof.* This lemma is a generalization of Lemma C.2 in [2], relaxing the assumption on the learning rate, as well as extending the result to our update rule. By the performance difference lemma (see e.g. Lemma 3.2 in [2]), it is sufficient to show that:

$$\forall s \in \mathcal{S}, \quad \text{adv}_t(s, \pi_{t+1}) \doteq \sum_a \pi_{t+1}(a|s)\text{adv}_t(s, a) \geq 0. \tag{28}$$

Let us define:

$$\mathcal{A}_s^\top = \{a \in \mathcal{A} | \text{adv}_t(s, a) \geq 0\}, \tag{29}$$
$$\mathcal{A}_s^\perp = \{a \in \mathcal{A} | \text{adv}_t(s, a) < 0\}. \tag{30}$$

For any $s$, we have (the weights in the exponential have all increased, and the advantages are non-negative):

$$\sum_{a \in \mathcal{A}_s^\top} e^{(\theta_t)_{s,a} + \eta_t d_t(s)\pi_t(a|s)\text{adv}_t(s,a)} \text{adv}_t(s, a) \geq \sum_{a \in \mathcal{A}_s^\top} e^{(\theta_t)_{s,a}} \text{adv}_t(s, a). \tag{31}$$

Similarly (the weights in the exponential have all decreased, and the advantages are negative):

$$\sum_{a \in \mathcal{A}_s^\perp} e^{(\theta_t)_{s,a} + \eta_t d_t(s)\pi_t(a|s)\text{adv}_t(s,a)} \text{adv}_t(s, a) \geq \sum_{a \in \mathcal{A}_s^\perp} e^{(\theta_t)_{s,a}} \text{adv}_t(s, a). \tag{32}$$

Summing gives:

$$\sum_a e^{(\theta_{t+1})_{s,a}} \text{adv}_t(s, a) \geq \sum_a e^{(\theta_t)_{s,a}} \text{adv}_t(s, a) = 0. \tag{33}$$

Normalizing the left-hand side by $\sum_a e^{(\theta_{t+1})_{s,a}}$ gives the policy and concludes the proof. $\square$

**Corollary 1** (**Convergence under the direct and softmax parametrization**). *Under A1, the sequence of value functions $q_t$ uniformly converges: $q_\infty \doteq \lim_{t \to \infty} q_t$.*

*Proof.* Since $\forall s, a, t, q_t(s, a) \leq \frac{1}{1-\gamma} r_\top$, the monotonous convergence theorem guarantees the existence of $q_\infty \in \mathbb{R}^{|\mathcal{S}| \times |\mathcal{A}|}$ that is the limit of the sequence of $q_t$. $\square$

### B.2.2 Optimality

**Theorem 2** (**Optimality under the direct and softmax parametrization**). *Under A1, the following condition:*

*C4. Each state $s$ is updated with weights that sum to infinity over time: $\sum_{t=0}^\infty \eta_t d_t(s) = \infty$,*

*is necessary and sufficient to guarantee that the sequence of value functions $(q_t)$ converges to optimality: $q_\infty = q_\star \doteq \max_{\pi \in \Pi} q_\pi$.*

*Proof sketch.* In both direct and softmax parametrizations, we assume that there exists a state-action pair $(s, a)$ that is advantageous over the state value limits $q_\infty$ and $v_\infty \doteq \lim_{t \to \infty} v_t$: $\text{adv}_\infty(s, a) \doteq q_\infty(s, a) - v_\infty(s) > 0$. Then, we prove that in this state, the policy improvement yielded by the update is lower bounded by a linear function over the update weight $\eta_t d_t(s)$ applied in state $s$. Next, we sum over $t$ and notice that this lower bounding sum diverges to infinity, which contradicts Corollary 1.

We may therefore infer that there cannot exist a state-action pair that is advantageous over $q_\infty$ and $v_\infty$, which allows us to conclude that no policy improvement is possible, and thus that the values are optimal: $q_\infty = \max_{\pi \in \Pi} q_\pi$.

For the necessity of C4, we show that the parameter update is upper bounded in both parametrizations by a term linear in the action gap. By choosing the reward function adversarially, we may set it sufficiently small so that the sum of all the gradient steps are insufficient to reach optimality. $\square$

*Proof.* The sufficient conditions for direct and softmax parametrizations are treated independently in Thm. 2a and 2b. The necessary condition is proven in Thm. 2c. $\square$

**Theorem 2a** (**Optimality under the direct parametrization**). *Under A1 and C4, the sequence of value functions $q_t$ converges to optimality:*

$$q_\infty = q_\star \doteq \max_{\pi \in \Pi} q_\pi. \tag{34}$$

*Proof.* Let us assume that $v_\infty < v_\star$, then, by the policy improvement theorem, there must be some state $s$ for which an advantage $q_\infty(s, a_\top) - v_\infty(s) = \epsilon(s) > 0$ over $\pi_t$ exists, with $a_\top \in \mathcal{A}_\top(s) = \arg\max_{a \in \mathcal{A}} q_\infty(s, a)$. Let us define the state value-gap $\delta(s) := q_\infty(s, a_\top) - \max_{a_\perp \in \mathcal{A}_\perp(s)} q_\infty(s, a_\perp) > 0$, with $\mathcal{A}_\perp(s) := \mathcal{A}/\mathcal{A}_\top(s)$.

Since we proved that $q_t \to_{t \to \infty} q_\infty$, there exists $t_0$ such that for all $t \geq t_0$ and $a \in \mathcal{A}$, $q_\infty(s, a) - q_t(s, a) \leq \frac{\delta(s)}{2}$. This guarantees two things for any $t \geq t_0$:

$$\forall a \in \mathcal{A}_\top(s), q_t(s, a) \geq q_\infty(s, a_\top) - \frac{\delta(s)}{2}, \tag{35}$$

$$\forall a \in \mathcal{A}_\perp(s), q_t(s, a) \leq q_\infty(s, a_\top) - \delta(s). \tag{36}$$

Remembering that

$$\pi_{t+1}(\cdot|s) \doteq \mathcal{P}_{\Delta(\mathcal{A})}(\pi_t(\cdot|s) + \eta_t d_t(s) q_t(s, \cdot)), \tag{37}$$

the above inequalities allow us to apply Lemma 1 to $\pi_t(\cdot|s)$ with $\mathcal{A}_\top(s)$ the set of coordinates $\{1, \ldots, k\}$, $\alpha = \eta_t d_t(s)(q_\infty(s, a_\top) - \frac{\delta(s)}{2})$ and $\beta = \eta_t d_t(s)(q_\infty(s, a_\top) - \delta(s))$, giving:

$$\sum_{a \in \mathcal{A}_\top(s)} \pi_{t+1}(a|s) \geq \min\left(1, \sum_{a \in \mathcal{A}_\top(s)} \pi_t(a|s) + \frac{\alpha - \beta}{2}\right) \tag{38}$$

$$= \min\left(1, \sum_{a \in \mathcal{A}_\top(s)} \pi_t(a|s) + \eta_t d_t(s) \frac{\delta(s)}{4}\right). \tag{39}$$

By assumption, we know that $\sum_{a \in \mathcal{A}_\top(s)} \pi_t(a|s) \leq \lim_{t' \to \infty} \sum_{a \in \mathcal{A}_\top(s)} \pi_{t'}(a|s) < 1$, we may conclude that $\forall t \geq t_0$:

$$\sum_{a \in \mathcal{A}_\top(s)} \pi_{t+1}(a|s) - \sum_{a \in \mathcal{A}_\top(s)} \pi_t(a|s) \geq \eta_t d_t(s) \frac{\delta(s)}{4}. \tag{40}$$

Hence,

$$\lim_{t' \to \infty} \sum_{a \in \mathcal{A}_\top(s)} \pi_{t'}(a|s) - \sum_{a \in \mathcal{A}_\top(s)} \pi_{t_0}(a|s) \geq \sum_{t'=t_0}^{\infty} \eta_{t'} d_{t'}(s) \frac{\delta(s)}{4}. \tag{41}$$

Since $\sum_{t'=t}^{\infty} \eta_{t'} d_{t'}(s) = \infty$ and the left-hand side is bounded, this contradicts the assumption that there exists an advantageous action in state $s$. This property must be true in every state $s$, we may therefore conclude that $v_\infty$ is optimal. $\square$

**Theorem 2b** (**Optimality under the softmax parametrization**). *Under A1 and C4, the sequence of value functions $q_t$ converges to optimality:*

$$q_\infty = q_\star \doteq \max_{\pi \in \Pi} q_\pi. \tag{42}$$

*Proof sketch.* This theorem generalizes Thm. 5.1 of [2] and our proof borrows many of their ideas. The generalization could have been made by adapting some of the lemmas their proof relies on, but we believe that the full rewriting of the proof has its merits: it is self contained, and ends up being significantly shorter ( 3 pages versus 10 pages). In this proof sketch, we stack without proving them the arguments used in the full proof that follows.

We start by fixing state $s$ and by partitioning the action set according to their value $q_\infty(s,a)$ in $s$ as compared to $v_\infty(s)$: smaller $\mathcal{A}_s^<$, equal $\mathcal{A}_s^=$, or larger $\mathcal{A}_s^>$. We assume towards a contradiction that there must be some $s$ for which $\mathcal{A}_s^>$ is non-empty, as otherwise, the policy improvement theorem tells us that $v_\infty$ is optimal. In such a state, by the uniform convergence theorem, there is a timestep $t_0$ from which for all $t \geq t_0$, the values $q_t$ are $\xi$-close to $q_\infty$. We choose $\xi$ to be smaller than any action gap in state $s$. We then notice that if an action in $\mathcal{A}_s^=$ gets a policy smaller than that of an action in $\mathcal{A}_s^>$, then it remains this way in the future (it undergoes smaller updates). We further partition $\mathcal{A}_s^=$: $\mathcal{A}_s^{\leqq}$ contains the actions for which the policy gets smaller than the policy in all actions of $\mathcal{A}_s^>$, $\mathcal{A}_s^{\geqq}$ contains the actions for which this event never happens. We call $t_1 \geq t_0$, the timestep when $\forall a^= \in \mathcal{A}_s^{\leqq}, a^> \in \mathcal{A}_s^>, \pi_{t_1}(a^=|s) \leq \pi_{t_1}(a^>|s)$.

Then, we observe that $\mathcal{A}_s^{\geqq}$ cannot be empty because this set is the only one that contains actions that may prevent full convergence on $\mathcal{A}_s^>$. We then look at the evolution of the ratio of policy mass on $\mathcal{A}_s^>$ versus $\mathcal{A}_s^< \cup \mathcal{A}_s^{\leqq}$ and show that this ratio is monotonously increasing. We then establish an upper bound on it by arguing that otherwise it would imply that the sum of the updates on $\mathcal{A}_s^{\geqq}$ would become negative. This cannot happen because actions in $\mathcal{A}_s^{\geqq}$ need to diverge to infinity to get some policy mass, and none of them can compensate by diverging to $-\infty$, as it would otherwise have been partitioned in $\mathcal{A}_s^{\leqq}$. We conclude the proof by showing that if the parameter of an action in $\mathcal{A}_s^{\geqq}$ diverges to $+\infty$, then the parameters of actions in $\mathcal{A}_s^>$ will also diverge to $+\infty$. This implies the aforementioned ratio diverges to $\infty$ contradicting it being upper bounded. Hence $\mathcal{A}_s^>$ must be empty in every state and consequently $v_\infty$ optimal. $\qquad\square$

*Full proof.* We recall the update rule with softmax parametrization:

$$(\theta_{t+1})_{s,a} = (\theta_t)_{s,a} + \eta_t d_t(s)\pi_t(a|s)\mathrm{adv}_t(s,a). \tag{43}$$

We start from Corollary 1 that proves the existence of $v_\infty(s)$ and $q_\infty(s,a)$ to which $v_t(s)$ and $q_t(s,a)$ converge from below for any state and action. We then consider the following partition of the actions in state $s$:

$$\mathcal{A}_s^> = \{a \in \mathcal{A}|q_\infty(s,a) > v_\infty(s)\} \tag{44}$$
$$\mathcal{A}_s^= = \{a \in \mathcal{A}|q_\infty(s,a) = v_\infty(s)\} \tag{45}$$
$$\mathcal{A}_s^< = \{a \in \mathcal{A}|q_\infty(s,a) < v_\infty(s)\}. \tag{46}$$

If $\forall s, \ \mathcal{A}_s^> = \emptyset$, the policy improvement theorem tells us that $q_\infty$ is optimal and our work is done.

We thus assume towards a contradiction that $\mathcal{A}_s^>$ is non-empty for a certain state $s$. We define $\delta_\infty(s)$ as the smallest positive improvement of $q_\infty$ over $v_\infty$ in state $s$:

$$\delta_\infty(s) := \min_{a \in \mathcal{A}_s^>} q_\infty(s,a) - v_\infty(s). \tag{47}$$

From the convergence of $v_t(s)$ and $q_t(s,a)$, we know that for any $\xi > 0$, there exists $t_0$ such that for any $s, a$ and $t \geq t_0$:

$$v_\infty(s) - \frac{1}{2}\xi \leq v_t(s) \leq v_\infty(s), \tag{48}$$
$$q_\infty(s,a) - \xi \leq q_t(s,a) \leq q_\infty(s,a). \tag{49}$$

Moving forward, we choose $\xi \leq \frac{1}{2}\min\{\delta_\infty(s), v_\infty(s) - \max_{a\in\mathcal{A}_s^<} q_\infty(s,a)\}$ and let $t_0$ be the corresponding timestep.

This allows us to further partition the set $\mathcal{A}_s^=$ as:

$$\mathcal{A}_s^{\leqq} \doteq \{a \in \mathcal{A}_s^=|\exists t_a \geq t_0, \forall t' \geq t_a, \forall a^> \in \mathcal{A}_s^>, \theta_{t'}(s,a^>) \geq \theta_{t'}(s,a)\} \tag{50}$$
$$\mathcal{A}_s^{\geqq} \doteq \mathcal{A}_s^= \setminus \mathcal{A}_s^{\leqq}. \tag{51}$$

We note that trivially: $\mathcal{A} = \mathcal{A}_s^{\leqq} \cup \mathcal{A}_s^{\geqq} \cup \mathcal{A}_s^> \cup \mathcal{A}_s^<$. We let $t_1 = \max_{a\in\mathcal{A}_s^{\leqq}} t_a$ and in the following consider $t \geq t_1$. We recall the notation $f(\mathcal{X}) \doteq \sum_{x\in\mathcal{X}} f(x)$ for any function $f$ and set $\mathcal{X}$.

The updates $u_t(s,a) \doteq \pi_t(a|s)\mathrm{adv}_t(s,a)$ on those various sets can be bounded as follows.

On $\mathcal{A}_s^>$:

$$u_t(s, \mathcal{A}_s^>) \doteq \sum_{a^> \in \mathcal{A}_s^>} u_t(s, a^>) \tag{52}$$

$$= \sum_{a^> \in \mathcal{A}_s^>} \pi_t(a^>|s)(q_t(s, a^>) - v_t(s)) \tag{53}$$

$$\geq ((v_\infty(s) + \delta_\infty(s) - \xi) - v_\infty(s)) \sum_{a^> \in \mathcal{A}_s^>} \pi_t(a^>|s) \tag{54}$$

$$= (\delta_\infty(s) - \xi)\pi_t(\mathcal{A}_s^>|s). \tag{55}$$

On $\mathcal{A}_s^{\geq}$:

$$u_t(s, \mathcal{A}_s^{\geq}) \doteq \sum_{a^{\geq} \in \mathcal{A}_s^{\geq}} u_t(s, a^{\geq}) \tag{56}$$

$$= \sum_{a^{\geq} \in \mathcal{A}_s^{\geq}} \pi_t(a^{\geq}|s) \left( \sum_{b^< \in \mathcal{A}_s^< \cup \mathcal{A}_s^{\leq}} \pi_t(b^<|s)(q_t(s, a^{\geq}) - q_t(s, b^<)) \right.$$

$$+ \sum_{b^> \in \mathcal{A}_s^>} \pi_t(b^>|s)(q_t(s, a^{\geq}) - q_t(s, b^>)) \tag{57}$$

$$\left. + \sum_{b^{\geq} \in \mathcal{A}_s^{\geq}} \pi_t(b^{\geq}|s)(q_t(s, a^{\geq}) - q_t(s, b^{\geq})) \right)$$

$$\leq \sum_{a^{\geq} \in \mathcal{A}_s^{\geq}} \pi_t(a^{\geq}|s) \left( \pi_t(\mathcal{A}_s^<|s) + \pi_t(\mathcal{A}_s^{\leq}|s) \right) (v_\infty(s) - v_\perp)$$

$$- \sum_{a^{\geq} \in \mathcal{A}_s^{\geq}} \pi_t(a^{\geq}|s)\pi_t(\mathcal{A}_s^>|s)(\delta_\infty(s) - \xi) \tag{58}$$

$$+ \sum_{(a^{\geq}, b^{\geq}) \in \mathcal{A}_s^{\geq} \times \mathcal{A}_s^{\geq}} \pi_t(a^{\geq}|s)\pi_t(b^{\geq}|s)(q_t(s, a^{\geq}) - q_t(s, b^{\geq})) \tag{59}$$

$$= \pi_t(\mathcal{A}_s^{\geq}|s) \left( \left( \pi_t(\mathcal{A}_s^<|s) + \pi_t(\mathcal{A}_s^{\leq}|s) \right) (v_\infty(s) - v_\perp) - \pi_t(\mathcal{A}_s^>|s)(\delta_\infty(s) - \xi) \right) \tag{60}$$

$$\leq \left( 1 - \pi_t(\mathcal{A}_s^{\geq}|s) \right) (v_\infty(s) - v_\perp). \tag{61}$$

Finally, on $\mathcal{A}_s^<$:

$$u_t(s, \mathcal{A}_s^<) \doteq \sum_{a^< \in \mathcal{A}_s^<} u_t(s, a^<) \leq 0. \tag{62}$$

This gives:

$$(\theta_{t'})_{s, \mathcal{A}_s^>} \geq (\theta_t)_{s, \mathcal{A}_s^>} + (\delta_\infty(s) - \xi) \sum_{t''=t}^{t'-1} \eta_{t''} d_{t''}(s)\pi_{t''}(\mathcal{A}_s^>|s), \tag{63}$$

$$(\theta_{t'})_{s, \mathcal{A}_s^{\geq}} \leq (\theta_t)_{s, \mathcal{A}_s^{\geq}} + (v_\infty(s) - v_\perp) \sum_{t''=t}^{t'-1} \eta_{t''} d_{t''}(s) \left( 1 - \pi_{t''}(\mathcal{A}_s^{\geq}|s) \right), \tag{64}$$

$$(\theta_{t'})_{s, \mathcal{A}_s^<} \leq (\theta_t)_{s, \mathcal{A}_s^<}. \tag{65}$$

We treat the case of $\mathcal{A}_s^{\leq}$ slightly differently, and will instead use the following inequalities.

$\forall a^> \in \mathcal{A}_s^>, \; a^{\leq} \in \mathcal{A}_s^{\leq}, \; \theta_t(s, a^>) \geq \theta_t(s, a^{\leq}), \; \pi_t(s, a^>) \geq \pi_t(s, a^{\leq}), \text{ and } u_t(s, a^>) \geq 2u_t(s, a^{\leq}).$

The first two inequalities stem directly by construction of $\mathcal{A}_s^{\leqq}$. For the last one, we have:

$$
\begin{aligned}
u_t(s, a^>) = \pi_t(a^>|s)\mathrm{adv}_t(s, a^>) &\geq \pi_t(s, a^{\leqq})\mathrm{adv}_t(s, a^>) \\
&= \pi_t(s, a^{\leqq})(q_t(s, a^>) - v_t(s)) \\
&\geq \pi_t(s, a^{\leqq})(v_\infty(s) + \delta_\infty - \xi - v_t(s)) \\
&\geq \pi_t(s, a^{\leqq})(\delta_\infty - \xi) \\
&\geq \xi\pi_t(s, a^{\leqq}) \\
&\geq 2\pi_t(s, a^{\leqq})\mathrm{adv}_t(s, a^{\leqq}) = 2u_t(s, a^{\leqq}).
\end{aligned}
\tag{66}
$$

In the last line, we used: $\mathrm{adv}_t(s, a^{\leqq}) = q_t(s, a^{\leqq}) - v_t(s) \leq q_\infty(s, a^{\leqq}) - v_t(s) = v_\infty(s) - v_t(s) \leq \frac{\xi}{2}$.

$\underline{\mathcal{A}_s^{\geqq} \neq \emptyset.}$ If $\mathcal{A}_s^{\geqq}$ is empty, Eq. (63), Eq. (65) and Eq. (66) imply that $\pi_t(\mathcal{A}_s^{\geqq}|s)$ increases monotonously with $t$. C4 then guarantees that $(\theta_{t'})_{s,\mathcal{A}_s^{\geqq}}$ diverges to $\infty$. Consequently, $\pi_t(\mathcal{A}_s^{\geqq}|s)$ will converge to 1 and $v_t(s)$ will tend to a value larger than $v_\infty(s) + \delta_\infty(s)$ which is impossible. Moving forward, we consider $\mathcal{A}_s^{\geqq} \neq \emptyset$ non-empty.

$\underline{\text{Mass on } \mathcal{A}_s^{>} \text{ and } \mathcal{A}_s^{<} \cup \mathcal{A}_s^{\leqq}.}$ We now look at the evolution of $\mathrm{ratio}_t(\mathcal{A}_s^{>}, \mathcal{A}_s^{<} \cup \mathcal{A}_s^{\leqq}) \doteq \frac{\pi_t(\mathcal{A}_s^{>}|s)}{\pi_t(\mathcal{A}_s^{<}\cup\mathcal{A}_s^{\leqq}|s)}$:

$$
\mathrm{ratio}_{t+1}(\mathcal{A}_s^{>}, \mathcal{A}_s^{<} \cup \mathcal{A}_s^{\leqq}) = \frac{\sum_{a\in\mathcal{A}_s^{>}} \exp\left((\theta_{t+1})_{s,a}\right)}{\sum_{a\in\mathcal{A}_s^{<}\cup\mathcal{A}_s^{\leqq}} \exp\left((\theta_{t+1})_{s,a}\right)}
\tag{67}
$$

$$
= \frac{\sum_{a\in\mathcal{A}_s^{>}} \exp\left((\theta_t)_{s,a} + u_t(s,a)\right)}{\sum_{a\in\mathcal{A}_s^{<}\cup\mathcal{A}_s^{\leqq}} \exp\left((\theta_t)_{s,a} + u_t(s,a)\right)}
\tag{68}
$$

$$
\geq \frac{\sum_{a\in\mathcal{A}_s^{>}} \exp\left((\theta_t)_{s,a}\right)}{\sum_{a\in\mathcal{A}_s^{<}\cup\mathcal{A}_s^{\leqq}} \exp\left((\theta_t)_{s,a}\right)}
\tag{69}
$$

$$
\geq \mathrm{ratio}_t(\mathcal{A}_s^{>}, \mathcal{A}_s^{<} \cup \mathcal{A}_s^{\leqq})
\tag{70}
$$

Eq. (70) establishes the monotonicity of the ratio and shows that $\forall t' \geq t$:

$$
\pi_{t'}(\mathcal{A}_s^{>}|s) \geq \pi_{t'}(\mathcal{A}_s^{<} \cup \mathcal{A}_s^{\leqq}|s)\mathrm{ratio}_t(\mathcal{A}_s^{>}, \mathcal{A}_s^{<} \cup \mathcal{A}_s^{\leqq}).
\tag{71}
$$

$\underline{\text{Eq. (60) remains positive.}}$ Eq. (70) allows us to show that Eq. (60) cannot become negative. Let us assume towards a contradiction that it does become negative at a given time $t$, ie

$$
\left(\pi_t(\mathcal{A}_s^{<}|s) + \pi_t(\mathcal{A}_s^{\leqq}|s)\right)(v_\infty(s) - v_\perp) - \pi_t(\mathcal{A}_s^{>}|s)(\delta_\infty(s) - \xi) \leq 0.
$$

The monotonicity of the above ratio then guarantees that it will remain negative for all $t' > t$. In other words, for all $t' > t$, $u_{t'}(s, \mathcal{A}_s^{\geqq}) \leq 0$.

If the sum of updates on actions in $\mathcal{A}_s^{\geqq}$ is negative, two things can happen:

- Either the actions in $\mathcal{A}_s^{\geqq}$ all have parameters $\theta$ that are upper bounded. This means that $\pi_t(\mathcal{A}_s^{>}|s)$ will eventually converge to 1. In this case, we can reuse the arguments used to disprove $\mathcal{A}_s^{\geqq} = \emptyset$, and reach the same contradiction.

- Or at least one action in $\mathcal{A}_s^{\geqq}$ has a parameter $\theta$ whose $\limsup$ is $+\infty$. In that case, since the sum of updates on the set $\mathcal{A}_s^{\geqq}$ is negative, at least one action in $\mathcal{A}_s^{\geqq}$ must have a $\theta$ whose $\liminf$ is $-\infty$. Letting $a$ denote that action, there thus exists a time $t$ at which $\forall a^> \in \mathcal{A}_s^{>}, \theta_t(s, a^>) > \theta_t(s, a)$. Following the reasoning from Eq. (66), the updates to those various actions will guarantee that the property holds for all $t' \geq t$, which contradicts the construction of $\mathcal{A}_s^{\geqq}$.

Thus, we assume from now on that Eq. (60) is positive.

Boundedness of the ratio and lower bound on $\pi_{t'}(\mathcal{A}_s^>|s)$. The positivity of Eq. (60) gives us an upper bound on the ratio: $\text{ratio}_t(\mathcal{A}_s^>, \mathcal{A}_s^< \cup \mathcal{A}_s^{\doteq}) \leq \frac{v_\infty(s) - v_\perp}{\delta_\infty(s) - \xi}$, and equivalently the interesting inequality:

$$\pi_t(\mathcal{A}_s^< \cup \mathcal{A}_s^{\doteq}|s)(v_\infty(s) - v_\perp) \geq \pi_t(\mathcal{A}_s^>|s)(\delta_\infty(s) - \xi) \tag{72}$$

$$\iff \quad \pi_t(\mathcal{A}_s^< \cup \mathcal{A}_s^{\doteq}|s)(v_\infty(s) - v_\perp + \delta_\infty(s) - \xi) \geq (\pi_t(\mathcal{A}_s^>|s) + \pi_t(\mathcal{A}_s^< \cup \mathcal{A}_s^{\doteq}|s))(\delta_\infty(s) - \xi) \tag{73}$$

$$\iff \quad \pi_t(\mathcal{A}_s^< \cup \mathcal{A}_s^{\doteq}|s) \geq \frac{(1 - \pi_t(\mathcal{A}_s^{\doteq}|s))(\delta_\infty(s) - \xi)}{v_\infty(s) - v_\perp + \delta_\infty(s) - \xi}. \tag{74}$$

Combining it with Eq. (71), we get for all $t' > t$:

$$\pi_{t'}(\mathcal{A}_s^>|s) \geq \pi_{t'}(\mathcal{A}_s^< \cup \mathcal{A}_s^{\doteq}|s)\text{ratio}_t(\mathcal{A}_s^>, \mathcal{A}_s^< \cup \mathcal{A}_s^{\doteq}) \tag{75}$$

$$\geq \underbrace{\frac{\text{ratio}_t(\mathcal{A}_s^>, \mathcal{A}_s^< \cup \mathcal{A}_s^{\doteq})(\delta_\infty(s) - \xi)}{v_\infty(s) - v_\perp + \delta_\infty(s) - \xi}}_{\kappa > 0} \left(1 - \pi_{t'}(\mathcal{A}_s^{\doteq}|s)\right). \tag{76}$$

The policy cannot converge to $\mathcal{A}_s^{\doteq}$. Let us assume that the policy converges on $\mathcal{A}_s^{\doteq}$:

$$\lim_{t \to \infty} \pi_t(\mathcal{A}_s^{\doteq}|s) = 1. \tag{77}$$

From Eq. (55), we know that $(\theta_t)_{s,\mathcal{A}^>}$ increases with time (and we assumed that $\mathcal{A}_s^>$ is non-empty). This implies that $(\theta_t)_{s,\mathcal{A}^{\doteq}} \to \infty$. We may therefore infer from Eq. (64) that:

$$\sum_{t=0}^{\infty} \eta_t d_t(s) \left(1 - \pi_{t'}(\mathcal{A}_s^{\doteq}|s)\right) = \infty, \tag{78}$$

then:

$$(\theta_{t'})_{s,\mathcal{A}_s^>} \geq (\theta_t)_{s,\mathcal{A}_s^>} + \sum_{t''=t}^{t'-1} \eta_{t''} d_{t''}(s)\pi_{t''}(\mathcal{A}_s^>|s)(\delta_\infty(s) - \xi) \tag{79}$$

$$\geq (\theta_t)_{s,\mathcal{A}_s^>} + \kappa(\delta_\infty(s) - \xi)\sum_{t''=t}^{t'-1} \eta_{t''} d_{t''}(s)\left(1 - \pi_{t''}(\mathcal{A}_s^{\doteq}|s)\right). \tag{80}$$

Thus: $(\theta_{t'})_{s,\mathcal{A}_s^>} \to \infty$, which implies the divergence of $\text{ratio}_t(\mathcal{A}_s^>, \mathcal{A}_s^< \cup \mathcal{A}_s^{\doteq})$, and a contradiction with the upper bound established earlier: we cannot have $\pi_t(\mathcal{A}_s^{\doteq}|s) \to 1$.

Conclusion. We thus know that some policy mass must remain outside of $\mathcal{A}_s^{\doteq}$. Because the ratio is bounded and increasing, some mass must be assigned to both $\mathcal{A}_s^>$ and $\mathcal{A}_s^< \cup \mathcal{A}_s^{-\infty}$. Equation 63 thus implies that $(\theta_t)_{s,\mathcal{A}_s^>}$ must diverge to $\infty$, and the ratio with it, leading to a final contradiction. We conclude that there does not exist a state where policy improvement is possible, in other words, the policy is optimal. $\qquad \square$

**Theorem 2c** (**Necessity of C4 for optimality**). *Assumption C4 is necessary to guarantee that the sequence of value functions $v_t$ converges to optimality.*

*Proof.* We consider a minimal MDP with one state: $\mathcal{S} \doteq \{s\}$ and two terminal actions: $\mathcal{A} \doteq \{a_1, a_2\}$. We assume that $\sum_t \eta_t d_t(s) = m < \infty$ and $r_1 = 1$. Now, we prove that we may choose $r_2 < 1$ such that the sequence $(v_t)$ does not converge to optimality: $\lim_{t \to \infty} v_t < v_\star = 1$.

**Direct parametrization:** the update is:

$$u_{s,a_1} = d_t(s)q_\pi(s, a_1) \tag{81}$$

$$= d_t(s) \tag{82}$$

$$u_{s,a_2} = d_t(s)q_\pi(s, a_2) \tag{83}$$

$$= d_t(s)r_2 \tag{84}$$

$$(\theta_{t+1})_{s,a_1} = \min\left(1, (\theta_t)_{s,a_1} + \eta_t d_t(s)\frac{1 - r_2}{2}\right) \tag{85}$$

$$(\theta_{t+1})_{s,a_2} = \max\left(0, (\theta_t)_{s,a_2} + \eta_t d_t(s)\frac{r_2 - 1}{2}\right). \tag{86}$$

Starting from $(\theta_0)_{s,a_1}$ (resp. $\theta_0(a_2|s)$), $\theta_t(a_1|s)$ (resp. $\theta_t(a_2|s)$) hits 1 (resp. 0), if and only if:

$$\sum_{t=0}^{\infty} \eta_t d_t(s)\frac{1 - r_2}{2} \geq 1 - (\theta_0)_{s,a_1} \tag{87}$$

$$\Longleftrightarrow \qquad m(1 - r_2) \geq 2 - 2(\theta_0)_{s,a_1} \tag{88}$$

$$\Longleftrightarrow \qquad r_2 \leq 1 - \frac{2}{m}\left(1 - (\theta_0)_{s,a_1}\right). \tag{89}$$

If we adversarially choose $r_2 = 1 - \frac{1}{m}\left(1 - (\theta_0)_{s,a_1}\right)$, then the optimality is never reached.

**Softmax parametrization:** the update is:

$$u_{s,a_1,t} = d_t(s)\pi_t(a_1|s)\text{adv}_t(s, a_1) \tag{90}$$

$$= d_t(s)\pi_t(a_1|s)(1 - \pi_t(a_1|s) - \pi_t(a_2|s)r_2) \tag{91}$$

$$= d_t(s)\pi_t(a_1|s)(1 - \pi_t(a_1|s))(1 - r_2) \tag{92}$$

$$\leq d_t(s)\frac{1 - r_2}{4} \tag{93}$$

$$u_{s,a_2,t} = d_t(s)\pi_t(a_2|s)\text{adv}_t(s, a_2) \tag{94}$$

$$= d_t(s)\pi_t(a_2|s)(r_2 - \pi_t(a_1|s) - \pi_t(a_2|s)r_2) \tag{95}$$

$$= d_t(s)\pi_t(a_1|s)(1 - \pi_t(a_1|s))(r_2 - 1) \tag{96}$$

$$\geq d_t(s)\frac{r_2 - 1}{4}. \tag{97}$$

Therefore, we know that, starting from $(\theta_0)_{s,a_1}$, $\theta_t$ is upper bounded:

$$(\theta_t)_{s,a_1} \doteq (\theta_0)_{s,a_1} + \sum_{t'=0}^{t} \eta_{t'} u_{s,a_1,t} \tag{98}$$

$$\leq (\theta_0)_{s,a_1} + \sum_{t'=0}^{t} \eta_{t'} d_t(s)\frac{1 - r_2}{4} \tag{99}$$

$$\leq (\theta_0)_{s,a_1} + m\frac{1 - r_2}{4}. \tag{100}$$

$$\tag{101}$$

Similarly, we know that, starting from $(\theta_0)_{s,a_2}$, $\theta_t$ is upper bounded:

$$(\theta_t)_{s,a_2} \doteq (\theta_0)_{s,a_2} + \sum_{t'=0}^{t} \eta_{t'} u_{s,a_2,t} \tag{102}$$

$$\geq (\theta_0)_{s,a_2} + \sum_{t'=0}^{t} \eta_{t'} d_t(s)\frac{r_2 - 1}{4} \tag{103}$$

$$\geq (\theta_0)_{s,a_2} - m\frac{1 - r_2}{4}. \tag{104}$$

$$\tag{105}$$

As a consequence, for any $0 \leq r_2 < 1$ and any $t$, we have:

$$v_\star(s) - v_t(s) = 1 - (1 - \pi_t(a_2|s)(1 - r_2)) \tag{106}$$

$$\geq \frac{\exp\left((\theta_0)_{s,a_2} - m\frac{1-r_2}{4}\right)}{\exp\left((\theta_0)_{s,a_1} + m\frac{1-r_2}{4}\right)}(1 - r_2) \tag{107}$$

$$\geq \exp\left((\theta_0)_{s,a_2} - (\theta_0)_{s,a_1} - \frac{m}{2}\right)(1 - r_2), \tag{108}$$

which is strictly positive and therefore concludes the proof. $\qquad\square$

### B.2.3 Convergence rates

**Theorem 4** (**Asymptotic convergence rates under the softmax parametrization**). *With softmax parametrization, under A1, C4, and A8:*

*A8. The optimal policy is unique:* $\forall s, \; q_\star(s, a_1) = q_\star(s, a_2) = v_\star(s)$ *implies* $a_1 = a_2$,

*the sequence of value functions $q_t$ converges asymptotically as follows:*

$$\exists t_0, \; such \; that \; \forall t \geq t_0, \quad v_\star(s) - v_t(s) \leq \frac{8|\mathcal{A}|(v_\top - v_\perp)}{(1-\gamma)\min_{s \in supp(d_{\pi_\star,\gamma})} \delta(s) \sum_{t'=t_0}^{t-1} \eta_{t'} d_{t'}(s)}, \tag{5}$$

*where $\delta(s) = v_\star(s) - \max_{a \in \mathcal{A}/\{\pi_\star(s)\}} q_\star(s, a)$ is the gap with the best suboptimal action in state $s$, $v_\top$ (resp. $v_\perp$) is the maximal (resp. minimal) value, and $supp(d_{\pi_\star,\gamma})$ denotes the support of the distribution of the optimal policy.*

**Remark 1.** *With A8, Thm. 4 provides bounds that improve the asymptotic rates that were stated before from $\mathcal{O}(|\mathcal{A}|^3)$ to $\mathcal{O}(|\mathcal{A}|)$.*

*We note that we were not able to convince ourselves that [22] (the only existing convergence rate for the softmax policy) properly handles the case of multiple optimal policies either. We do believe that their proof works in the bandit setting, corresponding to their Section 3.2.1. However the general MDP setting studied in Section 3.2.2 is problematic. More specifically, their proof of Lemma 8 for multiple actions (only found in the Appendix) does not apply to the sum over all optimal actions, but to the sum over all greedy actions with respect to the current policy. This prevents a direct application of the rest of their results and raises the question of whether the rate does hold in that case.*

*We do not know whether the issue is structural, but we do believe that our proof technique, which ignores RL optimization properties, will not be able to prove a bound in $\mathcal{O}(1/t)$ for problems with multiple optimal policies. Indeed, the problem arises when the optimization commits to one optimal action $a_1^\top$ at the expense of $a_2^\top$ in some state $s$ (because e.g. $q_t(s, a_2^\top) < v_t(s) < q_t(s, a_1^\top)$), up to a point where $\pi_t(s, a_1^\top) \gg \pi_t(s, a_2^\top)$, and $a_2^\top$ later experiences a significant jump in value due to efficient optimization in its subsequent states, inducing $q_t(s, a_1^\top) < v_t(s) < q_t(s, a_2^\top)$. Then, the parameter update will be extremely slow: proportional to the product of $\pi_t(s, a_2^\top)$ and $(1 - \pi_t(s, a_1^\top) + \pi_t(s, a_2^\top))$, which are two very small quantities. The policy convergence to $\mathcal{A}_s^\top$ will be much slower, even go backwards for some time, since $(\theta_t)_{s,a_1^\top}$ will decrease and $(\theta_t)_{s,a_2^\top}$ will be too small to have a significant impact on $\sum_{a^\top \in \mathcal{A}_s^\top} \pi_t(s, a^\top)$.*

*Proof sketch.* We consider the partition between the optimal and the suboptimal actions in every state $s$:

$$\mathcal{A}_s^\top = \{a \in \mathcal{A} | q_\star(s, a) = v_\star(s)\} \quad \text{and} \quad \mathcal{A}_s^\perp = \{a \in \mathcal{A} | q_\star(s, a) < v_\star(s)\}. \tag{109}$$

We define $\delta(s)$ as the gap with the best suboptimal action: $\delta(s) \doteq v_\star(s) - \max_{a \in \mathcal{A}_s^\perp} q_\star(s, a)$. Then, thanks to the uniform convergence and optimality proved in Corollary 1 and Thm. 2, we know that there exists $t_0$, such that $\forall t \geq t_0, \|q_\star - q_t\|_\infty \leq \frac{\delta(s)}{2}$. Then, we consider the sequence:

$$(X_t)_{t \geq t_0} \doteq \begin{cases} X_{t_0} = \max_{a \in \mathcal{A}_s^\top}(\theta_{t_0})_{s,a} - \max_{a \in \mathcal{A}_s^\perp}(\theta_{t_0})_{s,a} \\ X_{t+1} = X_t + \frac{\delta(s)}{8}\eta_t d_t(s) e^{-X_t} \end{cases} \tag{110}$$

and prove that $\forall t \geq t_0$, $X_t \leq \max_{a \in \mathcal{A}_s^\top}(\theta_t)_{s,a} - \max_{a \in \mathcal{A}_s^\perp}(\theta_t)_{s,a}$. Further analysis allows us to prove the following rate in policy convergence to suboptimal actions:

$$\sum_{a \in \mathcal{A}_s^\perp} \pi_t(a|s) \leq \frac{8|\mathcal{A}|}{\delta(s) \sum_{t'=t_0}^{t-1} \eta_{t'} d_{t'}(s)}, \tag{111}$$

which allows us to prove the following upper bound on the value convergence rate:

$$v_\star(s) - v_t(s) \leq \frac{8|\mathcal{A}|(v_\top - v_\perp)}{(1-\gamma) \min_{s \in \mathrm{supp}(d_{\pi_\star, \gamma})} \delta(s) \sum_{t'=t_0}^{t-1} \eta_{t'} d_{t'}(s)}. \qquad \square$$

*Full proof.* We consider the following partition of the actions in state $s$:

$$\mathcal{A}_s^\top = \{a \in \mathcal{A} | q_\star(s,a) = v_\star(s)\} \tag{112}$$
$$\mathcal{A}_s^\perp = \{a \in \mathcal{A} | q_\star(s,a) < v_\star(s)\}. \tag{113}$$

We define $\delta(s)$ as the gap with the best suboptimal action[7]:

$$\delta(s) := v_\star(s) - \max_{a \in \mathcal{A}_s^\perp} q_\star(s,a). \tag{114}$$

From the convergence of $v_t(s)$ and $q_t(s,a)$, we also know that for any $\xi > 0$, there exists $t_0$ such that for any $s, a$ and $t \geq t_0$:

$$v_\star(s) - \xi \leq v_t(s) \leq v_\star(s) \tag{115}$$
$$q_\star(s,a) - \xi \leq q_t(s,a) \leq q_\star(s,a). \tag{116}$$

We fix $\xi \doteq \frac{\delta(s)}{2}$. We then write the advantage as:

$$\mathrm{adv}_t(s,a) = \sum_{a' \neq a} \pi(a'|s) (q_t(s,a) - q_t(s,a')), \tag{117}$$

which gives the following bounds:

$$\forall a \in \mathcal{A}_s^\top, \quad \mathrm{adv}_t(s,a) \geq (1 - \pi_t(\mathcal{A}_s^\top|s)) (v_\star(s) - \xi - (v_\star(s) - \delta(s)))$$
$$+ (\pi_t(\mathcal{A}_s^\top|s) - \pi_t(a|s)) (v_\star(s) - \xi - v_\star(s)) \tag{118}$$
$$= \frac{\delta(s)}{2} (1 - \pi_t(\mathcal{A}_s^\top|s)) - \frac{\delta(s)}{2} (\pi_t(\mathcal{A}_s^\top|s) - \pi_t(a|s)), \tag{119}$$
$$\forall a \in \mathcal{A}_s^\perp, \quad \mathrm{adv}_t(s,a) = q_t(s,a) - v_t(s) \tag{120}$$
$$\leq q_\star(s,a) - (v_\star(s) - \xi) \tag{121}$$
$$\leq -\frac{\delta(s)}{2}. \tag{122}$$

The last term in Eq. (119) may imply a negative advantage for actions in $\mathcal{A}_s^\top$. This term is null if $\mathcal{A}_s^\top$ contains only one element $a_s^\top$ and this is the assumption (A8) we need to make to progress further.

Therefore, we have for the parameters:

$$(\theta_{t+1})_{s,a_s^\top} \geq (\theta_t)_{s,a_s^\top} + \eta_t d_t(s) \pi_t(a_s^\top|s) (1 - \pi_t(a_s^\top|s)) \frac{\delta(s)}{2}, \tag{123}$$

$$\forall a \in \mathcal{A}/\{a_s^\top\}, \quad (\theta_{t+1})_{s,a} \leq (\theta_t)_{s,a} - \eta_t d_t(s) \pi_t(a|s) \frac{\delta(s)}{2} \leq (\theta_t)_{s,a}. \tag{124}$$

Let us consider the following sequence:

$$(X_t)_{t \geq t_0} := \begin{cases} X_{t_0} = (\theta_{t_0})_{s,a_s^\top} - \max_{a \in \mathcal{A}/\{a_s^\top\}}(\theta_{t_0})_{s,a} \\ X_{t+1} = X_t + \frac{\delta(s)}{8} \eta_t d_t(s) e^{-X_t} \end{cases} \tag{125}$$

$$\tag{126}$$

---

[7]If there is no suboptimal action, then $\sum_{a \in \mathcal{A}_s^\top} \pi(a|s) = 1$ and we proved what we wanted.

We prove by induction that for all $t \geq t_0$, $X_t \leq (\theta_t)_{s,a_s^\top} - \max_{a \in \mathcal{A}/\{a_s^\top\}}(\theta_t)_{s,a}$.

We have equality at initialization. Let us now assume the property true for $t$ and prove it still holds for $t + 1$. Let $a_t^\perp$ denote the action in $\mathcal{A}/\{a_s^\top\}$ for which the parameter is maximal at time $t$:

$$a_t^\perp \in \operatorname*{argmax}_{a \in \mathcal{A}/\{a_s^\top\}} (\theta_t)_{s,a} = \operatorname*{argmax}_{a \in \mathcal{A}/\{a_s^\top\}} \pi_t(a|s). \tag{127}$$

We recall that $t \geq t_0$ guarantees that $v_\star - v_t \leq \xi$, which implies that $\pi_t(a_s^\top|s) \geq \frac{1}{2}$. Moreover,

$$1 - \pi_t(a_s^\top|s) = \frac{\sum_{a \in \mathcal{A}/\{a_s^\top\}} e^{(\theta_t)_{s,a}}}{\sum_{a \in \mathcal{A}} e^{(\theta_t)_{s,a}}} \tag{128}$$

$$\geq \frac{e^{(\theta_t)_{s,a_t^\perp}}}{2e^{(\theta_t)_{s,a_s^\top}}}. \tag{129}$$

We now compute:

$$(\theta_{t+1})_{s,a_s^\top} - \max_{a \in \mathcal{A}/\{a_s^\top\}} (\theta_{t+1})_{s,a} \geq (\theta_t)_{s,a_s^\top} - (\theta_t)_{s,a_t^\perp} + \eta_t d_t(s)\pi_t(a_s^\top|s)\left(1 - \pi_t(a_s^\top|s)\right)\frac{\delta(s)}{2} \tag{130}$$

$$\geq X_t + \eta_t d_t(s)\frac{1}{2}\frac{e^{(\theta_t)_{s,a_t^\perp}}}{2e^{(\theta_t)_{s,a_s^\top}}}\frac{\delta(s)}{2} \tag{131}$$

$$\geq X_t + \frac{\delta(s)}{8}\eta_t d_t(s)e^{-(\theta_t)_{s,a_s^\top}+(\theta_t)_{s,a_t^\perp}} \tag{132}$$

$$\geq X_t + \frac{\delta(s)}{8}\eta_t d_t(s)e^{-X_t} \tag{133}$$

$$= X_{t+1} \tag{134}$$

which concludes the induction.

Above, (130) is a direct application of (123) and (124), and (133) comes from the induction hypothesis and the fact that $x \mapsto x + \frac{\delta(s)}{8}\eta_t d_t(s)e^{-x}$ is increasing on $[\frac{\delta(s)}{8}\eta_t d_t(s), +\infty)$.

$X_t$ belonging to that interval is easily guaranteed for $t$ large enough as $X_t \to \infty$ and we assume $\eta_t d_t(s)$ to be bounded.

Let us now study the sequence $(X_t)_{t \geq t_0}$. To that end, we define the function $f(t)$ solution on $[t_0, +\infty)$ of the ordinary differential equation (note that $t$ is now a continuous variable):

$$\begin{cases} f(t_0) = X_{t_0} \\ \frac{df(t)}{dt} = \frac{\delta(s)}{8}\eta_t d_t(s)e^{-f(t)}, \end{cases} \tag{135}$$

where $\eta_t d_t(s)$ is the piece-wise constant function defined as $\eta_t d_t(s) = \eta_{\lfloor t \rfloor} d_{\lfloor t \rfloor}(s)$.

From the evolution equations of $X_t$ and $f(t)$, we see that $\forall t \in \mathbb{N}, X_t \geq f(t)$. Additionally, we have:

$$f(t) = \log\left(\frac{\delta(s)}{8}\int_{t_0}^t \eta_{t'} d_{t'}(s)dt' + e^{X_{t_0}}\right). \tag{136}$$

In particular, going back to $t \in \mathbb{N}$, we obtain:

$$X_t \geq \log\left(\frac{\delta(s)}{8}\int_{t_0}^t \eta_{t'} d_{t'}(s)dt' + e^{X_{t_0}}\right) = \log\left(\frac{\delta(s)}{8}\sum_{t'=t_0}^{t-1} \eta_{t'} d_{t'}(s) + e^{X_{t_0}}\right). \tag{137}$$

We can now write the following rate in policy convergence:

$$1 - \pi_t(a_s^\top | s) = \frac{\sum_{a \in \mathcal{A}/\{a_s^\top\}} e^{(\theta_t)_{s,a}}}{\sum_{a \in \mathcal{A}} e^{(\theta_t)_{s,a}}} \tag{138}$$

$$\leq \frac{(|\mathcal{A}| - 1) e^{(\theta_t)_{s,a_t^\perp}}}{e^{(\theta_t)_{s,a_s^\top}}} \tag{139}$$

$$\leq (|\mathcal{A}| - 1) e^{-X_t} \tag{140}$$

$$\leq |\mathcal{A}| \frac{1}{\frac{\delta(s)}{8} \sum_{t'=t_0}^{t-1} \eta_{t'} d_{t'}(s) + e^{X_t}} \tag{141}$$

$$\leq \frac{8|\mathcal{A}|}{\delta(s) \sum_{t'=t_0}^{t-1} \eta_{t'} d_{t'}(s)}. \tag{142}$$

On the value side, we further get:

$$v_\star(s) - v_t(s) = \mathbb{P}[A = a_s^\top | A \sim \pi_t(\cdot|s)] \times \gamma \mathbb{E}[v_\star(S') - v_t(S') | S' \sim p(\cdot|s, a_s^\top)] \tag{143}$$
$$+ \mathbb{P}[A \in \mathcal{A}/\{a_s^\top\} | A \sim \pi_t(\cdot|s)] (v_\star(s) - \mathbb{E}[q_t(s, A) | A \sim \pi_t(\cdot|s) \cap \mathcal{A}/\{a_s^\top\}])$$

$$\leq \gamma \mathbb{E}[v_\star(S') - v_t(S') | S' \sim p(\cdot|s, a_s^\top)] + \frac{8|\mathcal{A}|(v_\star(s) - \min_{a \in \mathcal{A}} q_t(s, a))}{\delta(s) \sum_{t'=t_0}^{t-1} \eta_{t'} d_{t'}(s)} \tag{144}$$

$$\leq \frac{8|\mathcal{A}|(v_\top - v_\perp)}{(1 - \gamma) \min_{s \in \text{supp}(d_{\pi_\star, \gamma})} \delta(s) \sum_{t'=t_0}^{t-1} \eta_{t'} d_{t'}(s)}, \tag{145}$$

where $v_\top$ (resp. $v_\perp$) stand for the maximal (resp. minimal) value, which is upper bounded by $\frac{r_\top}{1-\gamma}$ (resp. $\frac{r_\perp}{1-\gamma}$), often times much smaller (resp. larger), and where $\text{supp}(d_{\pi_\star, \gamma})$ denotes the support of the distribution of the optimal policy. This concludes the proof. $\square$

**Theorem 3** (**Asymptotic convergence rates under the direct parametrization**). *With direct parametrization, under A1 and C4, the sequence of value functions $q_t$ converges to optimality in finite time:*

$$\exists t_0, \text{ such that } \forall t \geq t_0, \quad q_t = q_\star. \tag{4}$$

*Proof.* This is a direct consequence of Lemma 1 and Eq. (39) in Thm. 2a, which shows that the probability of choosing an optimal action grows by updates that sum to infinity until it hits its ceiling probability of 1. Therefore, this ceiling will be hit in a finite time $t_1$ in all states. $\square$

**Lemma 1.** *Let $x \in \Delta_n$, the simplex of $\mathbb{R}^n$, and $y \in \mathbb{R}^n$. Assume that there exists $0 < k < n$ and $\alpha > \beta$ such that $\forall i \leq k, y_i \geq \alpha$ and $\forall i > k, y_i \leq \beta$, then*

$$\sum_{i \leq k} \mathcal{P}_{\Delta_n}(x + y)_i \geq \min\left(1, \sum_{i \leq k} x_i + \frac{\alpha - \beta}{2}\right). \tag{146}$$

*Proof.* Let us assume without loss of generality that $\forall i, y_i \geq 0$ (by e.g. subtracting from each of them $\min y_i$). Let us also define the following sets:

$$\mathcal{A}^\top = \{i \leq k\}, \tag{147}$$
$$\mathcal{A}^{\perp,+} = \{i > k \,|\, \mathcal{P}_{\Delta_n}(x + y)_i > 0\}, \tag{148}$$
$$\mathcal{A}^{\perp,-} = \{i > k \,|\, \mathcal{P}_{\Delta_n}(x + y)_i = 0\}. \tag{149}$$

We assume that $\mathcal{A}^{\perp,+} \neq \emptyset$, as otherwise we would have $\sum_{i \leq k} \mathcal{P}_{\Delta_n}(x + y)_i = 1$, and (146) would hold.

We are interested in the effect of $\mathcal{P}_{\Delta_n}$ on the mass of the coordinates of $x + y$ that belong to $\mathcal{A}^\top$. Given the assumption that $y$ is coordinate-wise non-negative, the Euclidean projection onto $\Delta_n$ will remove from the coordinates of $x + y$ the mass $\sum_i y_i$ that $x$ gained by adding $y$ to it. That mass will be removed as uniformly as possible (corresponding to the equality case in Cauchy-Schwarz

inequality), while ensuring that each coordinate remains positive. In our case, this means that $\mathcal{P}_{\Delta_n}$ will remove all the mass of the coordinates in $\mathcal{A}^{\perp,-}$, and then remove the remaining added mass uniformly from the other coordinates[8]:

- The mass added to the first $k$ coordinates is $\sum_{i \in \mathcal{A}^\top} y_i$.

- The total mass to remove is $K = \sum_i y_i$.

- The mass removed from $\mathcal{A}^{\perp,-}$ is $K_{\mathcal{A}^{\perp,-}} \doteq \sum_{i \in \mathcal{A}^{\perp,-}} x_i + y_i$, which is larger than $\sum_{i \in \mathcal{A}^{\perp,-}} y_i$.

- The remaining mass $K_{\mathcal{A}^\top} + K_{\mathcal{A}^{\perp,+}} = K - K_{\mathcal{A}^{\perp,-}}$ is smaller than $\sum_{i \in \mathcal{A}^\top \cup \mathcal{A}^{\perp,+}} y_i$ and removed uniformly from the coordinates in $\mathcal{A}^\top \cup \mathcal{A}^{\perp,+}$.

Formally, this gives us:

$$\sum_{i \in \mathcal{A}^\top} \mathcal{P}_{\Delta_n}(x+y)_i - \sum_{i \in \mathcal{A}^\top} x_i = \sum_{i \in \mathcal{A}^\top} y_i - \frac{|\mathcal{A}^\top|}{|\mathcal{A}^\top| + |\mathcal{A}^{\perp,+}|} (K_{\mathcal{A}^\top} + K_{\mathcal{A}^{\perp,+}}) \tag{150}$$

$$\geq \sum_{i \in \mathcal{A}^\top} y_i - \frac{|\mathcal{A}^\top|}{|\mathcal{A}^\top| + |\mathcal{A}^{\perp,+}|} \sum_{i \in \mathcal{A}^\top \cup \mathcal{A}^{\perp,+}} y_i \tag{151}$$

$$= \frac{|\mathcal{A}^{\perp,+}|}{|\mathcal{A}^\top| + |\mathcal{A}^{\perp,+}|} \sum_{i \in \mathcal{A}^\top} y_i - \frac{|\mathcal{A}^\top|}{|\mathcal{A}^\top| + |\mathcal{A}^{\perp,+}|} \sum_{i \in \mathcal{A}^{\perp,+}} y_i \tag{152}$$

$$\geq \frac{|\mathcal{A}^{\perp,+}||\mathcal{A}^\top|}{|\mathcal{A}^\top| + |\mathcal{A}^{\perp,+}|} \alpha - \frac{|\mathcal{A}^\top||\mathcal{A}^{\perp,+}|}{|\mathcal{A}^\top| + |\mathcal{A}^{\perp,+}|} \beta \tag{153}$$

$$= \frac{|\mathcal{A}^{\perp,+}||\mathcal{A}^\top|}{|\mathcal{A}^\top| + |\mathcal{A}^{\perp,+}|} (\alpha - \beta). \tag{154}$$

$\frac{|\mathcal{A}^{\perp,+}||\mathcal{A}^\top|}{|\mathcal{A}^\top| + |\mathcal{A}^{\perp,+}|}$ is an increasing function of $|\mathcal{A}^{\perp,+}|$ and $|\mathcal{A}^\top|$ and both those quantities are larger than 1. Hence:

$$\sum_{i \in \mathcal{A}^\top} \mathcal{P}_{\Delta_n}(x+y)_i - x_i \geq \frac{\alpha - \beta}{2}, \tag{155}$$

which concludes the proof. $\qquad\square$

### B.3 Non-asymptotic convergence rates under A2

We now define an extra assumption that allows us to get non-asymptotic convergence rates under A2.

C7. There exists $d_\top \geq d_\perp > 0$ such that $\forall s, t, d_\top \geq d_t(s) \geq d_\perp$.

We will be needing $\left\| \frac{1}{p_0} \right\|_\infty$ and note that under A2, it is a well-defined quantity.

Finally, we let $M_t \in \mathbb{R}^{|\mathcal{S}||\mathcal{A}| \times |\mathcal{S}||\mathcal{A}|}$ be the diagonal matrix defined by $M_t(s,a;s,a) = \frac{d_t(s)}{d_{\pi_t,\gamma}(s)}$ and 0 elsewhere. We note that:

$$U(\theta_t, d_t) = M_t \nabla_{\theta|\theta_t} \mathcal{J}(\pi_t). \tag{156}$$

We also denote by $\lambda_t = \min_s \frac{d_t(s)}{d_{\pi_t,\gamma}(s)}$ and $\Lambda_t = \max_s \frac{d_t(s)}{d_{\pi_t,\gamma}(s)}$ the smallest and largest eigenvalues of $M_t$, and by $\kappa_t = \frac{\Lambda_t}{\lambda_t}$ its condition number. We have the following inequalities:

$$\lambda_t \geq (1-\gamma)d_\perp, \qquad \Lambda_t \leq d_\top \left\| \frac{1}{p_0} \right\|_\infty, \qquad \kappa_t \leq \left\| \frac{1}{p_0} \right\|_\infty \frac{d_\top}{(1-\gamma)d_\perp}. \tag{157}$$

---

[8]We note that this can potentially result in some coordinates in $\mathcal{A}^\top$ becoming negative. As we are looking for an upper bound on the mass removed from $\mathcal{A}^\top$, this is not problematic (the mass removed in excess from $\mathcal{A}^\top$ will then be added back to it and removed from coordinates both in and out of $\mathcal{A}^\top$, resulting in a smaller decrease).

**Theorem 7** (**Convergence rate for the softmax parametrization under A2**). *With a softmax parametrization, under A2, C7 and A8, and with $\eta = \frac{(1-\gamma)^3}{8} \frac{1}{d_\top} \left\| \frac{1}{p_0} \right\|_\infty^{-1}$, we have the following convergence rate for any s:*

$$v_\infty(s) - v_t(s) \leq \frac{16|\mathcal{S}|C}{(1-\gamma)^7 t} \cdot \left\| \frac{d_{\pi_\star,\gamma}}{p_0} \right\|_\infty^2 \left\| \frac{1}{p_0} \right\|_\infty^2 \frac{d_\top}{d_\perp}, \tag{158}$$

*where $C = \frac{1}{\inf_{s\in\mathcal{S},t\geq 1} \pi_t(\pi_\star(s)|s)^2}$ and $\pi_\star$ is the unique (per A8) optimal policy .*

*Proof.* We note that this result is a generalization of the bound from [22] to our more general update rule. The extra $\left\| \frac{1}{p_0} \right\|_\infty \frac{d_\top}{(1-\gamma)d_\perp}$ term, coming from the upper bound on $\kappa_t$, is the price we pay for the generalization.

Our proof follows the logic from [22] and relies on the fact that $M_t$ is diagonal and positive-definite. Hence it performs some form of conditioning on the gradient, which will affect the convergence rate, but does not change the overall evolution of the policies. More formally, we start by using the strong convexity of $\mathcal{J}$ to get:

$$|\mathcal{J}(\pi_{t+1}) - \mathcal{J}(\pi_t) - \langle \nabla_\theta \mathcal{J}(\pi_t), \theta_{t+1} - \theta_t \rangle| \leq \frac{4}{(1-\gamma)^3} \|\theta_{t+1} - \theta_t\|^2. \tag{159}$$

Now, reusing notations from [22], we let $\delta_t = \mathcal{J}(\pi_*) - \mathcal{J}(\pi_t)$. Let us also remember that $U(\theta_t, d_t) = M_t \nabla_\theta \mathcal{J}(\pi_t)$, and that $\lambda_t$ and $\Lambda_t$ are respectively the smallest and largest eigenvalues of $M_t$. Then

$$\delta_{t+1} - \delta_t = \mathcal{J}(\pi_t) - \mathcal{J}(\pi_{t+1}) \tag{160}$$

$$\leq -\langle \nabla_\theta \mathcal{J}(\pi_t), \theta_{t+1} - \theta_t \rangle + \frac{4}{(1-\gamma)^3} \|\theta_{t+1} - \theta_t\|^2 \tag{161}$$

$$= -\eta_t \langle \nabla_\theta \mathcal{J}(\pi_t), M_t \nabla_\theta \mathcal{J}(\pi_t) \rangle + \frac{4\eta_t^2}{(1-\gamma)^3} \|M_t \nabla_\theta \mathcal{J}(\pi_t)\|^2 \tag{162}$$

$$= \sum_s \frac{d_t(s)}{d_{\pi_t,\gamma}(s)} \left( -\eta_t + \frac{4\eta_t^2}{(1-\gamma)^3} \frac{d_t(s)}{d_{\pi_t,\gamma}(s)} \right) \langle \nabla_{\theta_s} \mathcal{J}(\pi_t), \nabla_{\theta_s} \mathcal{J}(\pi_t) \rangle \tag{163}$$

$$\leq \sum_s \frac{d_t(s)}{d_{\pi_t,\gamma}(s)} \left( -\eta_t + \frac{4\eta_t^2}{(1-\gamma)^3} \Lambda_t \right) \langle \nabla_{\theta_s} \mathcal{J}(\pi_t), \nabla_{\theta_s} \mathcal{J}(\pi_t) \rangle \tag{164}$$

$$\leq -\sum_s \frac{d_t(s)}{d_{\pi_t,\gamma}(s)} \frac{(1-\gamma)^3}{16} \frac{1}{\Lambda_t} \langle \nabla_{\theta_s} \mathcal{J}(\pi_t), \nabla_{\theta_s} \mathcal{J}(\pi_t) \rangle \qquad (\eta_t = \frac{(1-\gamma)^3}{8} \frac{1}{\Lambda_t}) \tag{165}$$

$$\leq -\sum_s \frac{(1-\gamma)^3}{16} \frac{\lambda_t}{\Lambda_t} \langle \nabla_{\theta_s} \mathcal{J}(\pi_t), \nabla_{\theta_s} \mathcal{J}(\pi_t) \rangle \tag{166}$$

$$= -\frac{(1-\gamma)^3}{16} \frac{\lambda_t}{\Lambda_t} \|\nabla_\theta \mathcal{J}(\pi_t)\|^2. \tag{167}$$

We now apply Lemma 8 from [22], which holds under A2 and A8, to get

$$\delta_{t+1} - \delta_t \leq -\frac{(1-\gamma)^3}{16|\mathcal{S}|} \frac{1}{\kappa_t} \left\| \frac{d_{\pi_\star,\gamma}}{d_{\pi_t,\gamma}} \right\|_\infty^{-2} \cdot \left[ \min_s \pi_t(a_\star(s)|s) \right]^2 \cdot |\mathcal{J}(\pi_*) - \mathcal{J}(\pi_t)|^2 \tag{168}$$

$$\leq -\frac{(1-\gamma)^3}{16|\mathcal{S}|} \frac{1}{\kappa_t} \left\| \frac{d_{\pi_\star,\gamma}}{p_0} \right\|_\infty^{-2} \cdot \left[ \inf_{t,s} \pi_t(a_\star(s)|s) \right]^2 \cdot \delta_t^2, \tag{169}$$

where $a_\star(s)$ is the unique optimal action in $s$ (per A8). We know from Thm. 2b that for all $s$, $\pi_t(a_\star(s)|s) \to 1$. Thus $\inf_{t,s} \pi_t(a_\star(s)|s) > 0$ and we can define $1/C \doteq \left[ \inf_{t,s} \pi_t(a_\star(s)|s) \right]^2 > 0$.

Let us write $K_t \doteq \frac{(1-\gamma)^3}{16|\mathcal{S}|C} \frac{1}{\kappa_t} \left\| \frac{d_{\pi_\star,\gamma}}{p_0} \right\|_\infty^{-2}$. We have $\delta_{t+1} - \delta_t \leq -K_t \delta_t^2$. Exactly as in [22], we now proceed by induction to show that $\delta_t \leq \frac{1}{\min_{i \leq t} K_i t}$.

For $t = 2$, we clearly have $\delta_1 \leq \frac{1}{1-\gamma} \leq \frac{1}{2K_1}$. Now, $\delta_{t+1} \leq \delta_t - \min_{i \leq t} K_i \delta_t^2$. On $[0, \frac{1}{2\min_{i \leq t} K_i}]$, the function $x - \min_{i \leq t} K_i x^2$ is increasing, hence:

$$\delta_{t+1} \leq \frac{1}{\min_{i \leq t} K_i t} - \frac{\min_{i \leq t} K_i}{\min_{i \leq t} K_i^2 t^2} \leq \frac{1}{\min_{i \leq t} K_i (t+1)} \leq \frac{1}{\min_{i \leq t+1} K_i (t+1)}, \tag{170}$$

which concludes the induction.

In the end, we get:

$$\mathcal{J}(\pi_*) - \mathcal{J}(\pi_t) \leq \frac{16|\mathcal{S}|C}{(1-\gamma)^3} \max_{i \leq t} \kappa_i \left\| \frac{d_{\pi_\star,\gamma}}{p_0} \right\|_\infty^2 \frac{1}{t} \tag{171}$$

$$\leq \frac{16|\mathcal{S}|C}{(1-\gamma)^4} \left\| \frac{1}{p_0} \right\|_\infty \left\| \frac{d_\top}{d_\perp} \right\| \left\| \frac{d_{\pi_\star,\gamma}}{p_0} \right\|_\infty^2 \frac{1}{t}, \tag{172}$$

where in the last line we used the bound on $\kappa$ from Eq (157). Note that to match the definitions in the bound from [22], we can write is as:

$$\mathcal{J}(\pi_*) - \mathcal{J}(\pi_t) \leq \frac{16|\mathcal{S}|C}{(1-\gamma)^6 t} \left\| \frac{(1-\gamma)d_{\pi_\star,\gamma}}{p_0} \right\|_\infty^2 \left\| \frac{1}{p_0} \right\|_\infty \frac{d_\top}{d_\perp}.$$

Finally, we move from $\mathcal{J}(\pi_*)$ and $\mathcal{J}(\pi_t)$ to $v_\infty(s)$ and $v_t(s)$ which introduces an extra $\frac{1}{1-\gamma} \left\| \frac{1}{p_0} \right\|_\infty$ and concludes the proof. □

We can use the above result to compute the time at which the rates from Thm. 4 become valid.

**Corollary 2.** $t_0$ *from Thm.* 4 *can be chosen as follows:*

$$t_0 = \frac{32|\mathcal{S}|C}{(1-\gamma)^7} \cdot \left\| \frac{d_{\pi_\star,\gamma}}{p_0} \right\|_\infty^2 \cdot \left\| \frac{1}{p_0} \right\|_\infty^2 \frac{d_\top}{d_\perp} \frac{1}{\inf_s \delta(s)}. \tag{173}$$

*Proof.* We know that the rates from Thm. 4 kick in for $t_0$ such that for any $s, a$ and $t \geq t_0$:

$$v_\infty(s) - \frac{\delta(s)}{2} \leq v_t(s) \leq v_\infty(s),$$

$$q_\infty(s,a) - \frac{\delta(s)}{2} \leq q_t(s,a) \leq q_\infty(s,a).$$

From Thm. 7, it is straightforward to show that it is verified for:

$$t_0 \geq \frac{32|\mathcal{S}|C}{(1-\gamma)^7 t} \cdot \left\| \frac{d_{\pi_\star,\gamma}}{p_0} \right\|_\infty^2 \left\| \frac{1}{p_0} \right\|_\infty^2 \frac{d_\top}{d_\perp} \frac{1}{\inf_s \delta(s)}. \qquad \square$$

## C Domains

### C.1 Chain domain

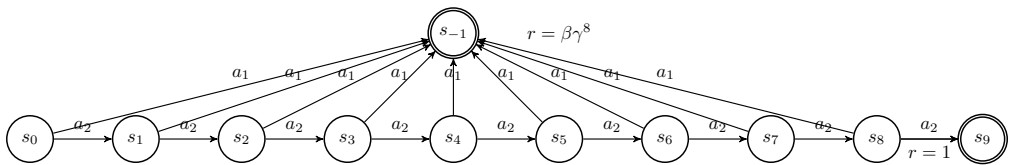

Figure 4: Deterministic chain MDP. Initial state is $s_0$. Reward is 0 everywhere except when accessing final states $s_{-1}$ and $s_9$. Reward in $s_{-1}$ is set such that $q(s_0, a_1) = \beta q_\star(s_0, a_2)$, with $\beta \in [0, 1)$.

The chain domain is designed to measure the ability of algorithms to overcome misleading rewards, *i.e.* immediate rewards that would guide the updates towards a suboptimal solution. In every state $s_k$, the agent has the opportunity to play action $a_1$ and receive an immediate reward of $\beta\gamma^{|\mathcal{S}|-2}$, or to play $a_2$ and progress to next state $s_{k+1}$ without any immediate reward. A reward of 1 is eventually obtained when reaching state $s_{|\mathcal{S}|-1}$. Fig. 4 represents a chain of size 10 (both terminal states $s_{-1}$ and $s_{\mathcal{S}-1}$ count as a single state). Formally, the deterministic MDP $\langle \mathcal{S}, \mathcal{A}, p, p_0, r, \gamma \rangle$ is constructed from two hyperparameters: the size $|\mathcal{S}|$ and the ratio $\beta$:

$$\mathcal{S} \doteq \{s_i\}_{i \in [\![0, |\mathcal{S}|-1]\!]} \tag{174}$$

$$\mathcal{A} \doteq \{a_1, a_2\} \tag{175}$$

$$p(\cdot | s, a) \doteq \begin{cases} \forall s, \ p(s_{|\mathcal{S}|-1} | s, a_1) = 1 \\ \forall i < |\mathcal{S}| - 1, \ p(s_{i+1} | s_i, a_2) = 1 \\ \text{the episode terminates when } s_{|\mathcal{S}|-1} \text{ is reached} \end{cases} \tag{176}$$

$$p_0(s_0) \doteq 1 \tag{177}$$

$$r(s, a) \doteq \begin{cases} \forall s, \ p(s, a_1) = \beta\gamma^{|\mathcal{S}|-2} \\ \forall i < |\mathcal{S}| - 2, \ r(s_i, a_2) = 0 \\ r(s_{|\mathcal{S}|-1}, a_2) = 1 \end{cases} \tag{178}$$

$$\gamma \doteq 0.99. \tag{179}$$

There are two remarkable policies:

- Low-hanging-fruit policy: $\begin{cases} \pi_\perp(a_1 | s) = 1 \\ v_\perp(s) = \beta\gamma^{|\mathcal{S}|-2} \end{cases}$

- Optimal policy: $\begin{cases} \pi_\star(a_2 | s) = 1 \\ v_\star(s_i) = \gamma^{|\mathcal{S}|-2-i} \end{cases}$

The policy gradient is attracted by the low-hanging-fruit policy because of its shorter horizon, even though it is sub-optimal.

In order to account for different experimental settings where the size of the chain $|\mathcal{S}|$ or the value ratio $\beta$ differ, we report the experimental results in terms of normalized expected return, formally computed as:

$$\overline{\mathcal{J}}_\pi = \frac{\mathcal{J}_\pi - \mathcal{J}_\perp}{\mathcal{J}_\star - \mathcal{J}_\perp}, \tag{180}$$

where $\mathcal{J}_\star = v_\star(s_0)$ is the performance of the optimal policy and $\mathcal{J}_\perp = v_\perp(s_0)$ the performance of the low-hanging-fruit policy. Note in particular that $\overline{\mathcal{J}}_\star = 1$ and $\overline{\mathcal{J}}_\perp = 0$.

### C.2 Random MDPs domain

The Random MDPs domain is designed to test the algorithms in situations where the exploration is not an issue. Indeed, by its design of highly stochastic transition functions, Random MDPs will have a non-null chance to visit every state whatever the behavioural policy is. This is therefore

a domain where we would expect the policy gradient update to perform well, if not the best, and hope our modified updates still perform comparably. We reproduce the Random MDPs environment published in Section B.1.3 of [18] (also in [25, 40]), for which we recall the specifications hereafter.

The Random MDPs domain uses four parameters for the MDP generation: the number of states, the number of actions in each state, the connectivity of the transition function stating how many states are reachable after performing a given action in a given state, and the discount factor $\gamma$. In our experiments, we fix the number of actions to 4, the connectivity to 2, and the discount factor $\gamma$ to 0.99.

The initial state is arbitrarily set to be $s_0$, we then search with dynamic programming the performance of the optimal policy for all potential terminal state $s_f \in \mathcal{S}/s_0$. We select the terminal state for which the optimal policy yields the smaller value function and set it as terminal: $r(s, a, s_f) = 1$ and $P(s|s_f, a) = 0$ for all $s \in \mathcal{S}$ and $a \in \mathcal{A}$. The reward function is set to 0 everywhere else. We found that the optimal value-function is on average 0.6, amounting to an average horizon of 10, and has surprising low variance. Below, we write this environmental MDP $\langle \mathcal{S}, \mathcal{A}, p, p_0, r, \gamma \rangle$, its optimal state-action value function $q_\star$, its optimal performance $\mathcal{J}_\star$, and its uniform policy performance $\mathcal{J}_u$, where $\widetilde{\pi}$ denotes the uniform random policy: $\pi_u(a|s) = \frac{1}{|\mathcal{A}|}$ for all $s \in \mathcal{S}$ and all $a \in \mathcal{A}$.

In order to account for different random MDPs, we report the experimental results in terms of normalized expected return, formally computed as:

$$\overline{\mathcal{J}}_\pi = \frac{\mathcal{J}_\pi - \mathcal{J}_u}{\mathcal{J}_\star - \mathcal{J}_u}. \tag{181}$$

Note in particular that $\overline{\mathcal{J}}_\star = 1$ and $\overline{\mathcal{J}}_u = 0$.

# D Full report of finite MDPs planning experiments (under A1)

## D.1 Chain domain

In classic algorithms, the actor is conflicted between two objectives: explore and visit the unknown states, or exploit and follow the best policy under the current incomplete knowledge. The chain experiment depicted on Fig. 4 is intended to test actor-critic algorithms' ability to explore and find the optimal policy under misleading rewards, and then to commit to this optimal policy. In every state, the agent has the opportunity to choose action $a_1$ and receive a fair reward, or to go further without any reward signal and get a chance to eventually yield a higher return.

Our analysis underlines the importance of exploration and states the requirements for guaranteeing sufficient exploration. Our experiment validates these findings. We consider:

- the on-policy updates:
  - policy gradient (PG) distribution: $\frac{d_{\pi_t,\gamma}}{\|d_{\pi_t,\gamma}\|_1}$,
  - undiscounted distribution: $\frac{d_{\pi_t,1}}{\|d_{\pi_t,1}\|_1}$,
- uniform distribution: $d_u = \frac{1}{|\mathcal{S}|}$,
- and off-policy trade-offs between the uniform and policy gradient distributions: $o_t d_u + (1 - o_t)\frac{d_{\pi_t,\gamma}}{\|d_{\pi_t,\gamma}\|_1}$
  - with $o_t$ constant: 0.1 and 0.48,
  - with $o_t \in \Theta(t^{-0.48})$: $o_t = 10t^{-0.48}$,
  - with $o_t \in \Theta(t^{-1})$: $o_t = 10t^{-1}$,

For a fair comparison, all update densities $d_t$ are normalized: $\sum_s d_t(s) = 1$.

We consider four experiments:

- **Exp1:** We observe instantaneous performance of each density schedule across iterations.
- We look at the speed of discovering the optimal policy. We measure it by tracking the number of updates required to reach a normalised return of 0.48 and its dependency on:
  - **Exp2:** the number of states in the chain,
  - **Exp3:** the value rate $\beta$ between the low-hanging-fruit policy and the optimal one,
  - **Exp4:** the learning rate.

For the softmax parametrization, we also add a policy entropy regularizing term $-\lambda \log \pi(\cdot|s)$ and observe its impact in **Exp5** (in replacement to **Exp4** which delivers the same results as in the direct parametrization), as is often done in practice.

### D.1.1 Direct parametrization

**Exp1:** Its results are displayed on Fig. 5a. We observe that on-policy updates converge very fast to the low-hanging fruit policy. Fast convergence can be desirable in easy-planning tasks, but in harder tasks such as the chain it leads to suboptimal behavior. In contrast, relying on a uniform state distribution for the actor updates is efficient and eventually discovers the optimal policy as long as the off-policiness $o_t$ is strong enough. It is worth noting that even under the guarantees offered by having C4 satisfied (when $o_t \in \Omega(t^{-1})$, the policy can get stuck on the low-hanging-fruit policy for a long time (as for $o_t = 10t^{-1}$). However, once the optimal policy has been identified the convergence is fast and meets the optimal performance predicted by Thm. 3.

**Exp2:** Its results are displayed on Fig. 5b. We observe a failure mode of on-policy updates. When the chain is of size 7 or more, the policy converges to the suboptimal solution and will never get out of it. In contrast, C4 guarantees the policy will eventually be optimal. However, as illustrated by the $o_t = 10t^{-1}$ curve, it might be very long (possibly exponentially long) if the condition is only satisfied tightly. The convergence gets faster as the weight granted to the uniform density increases.

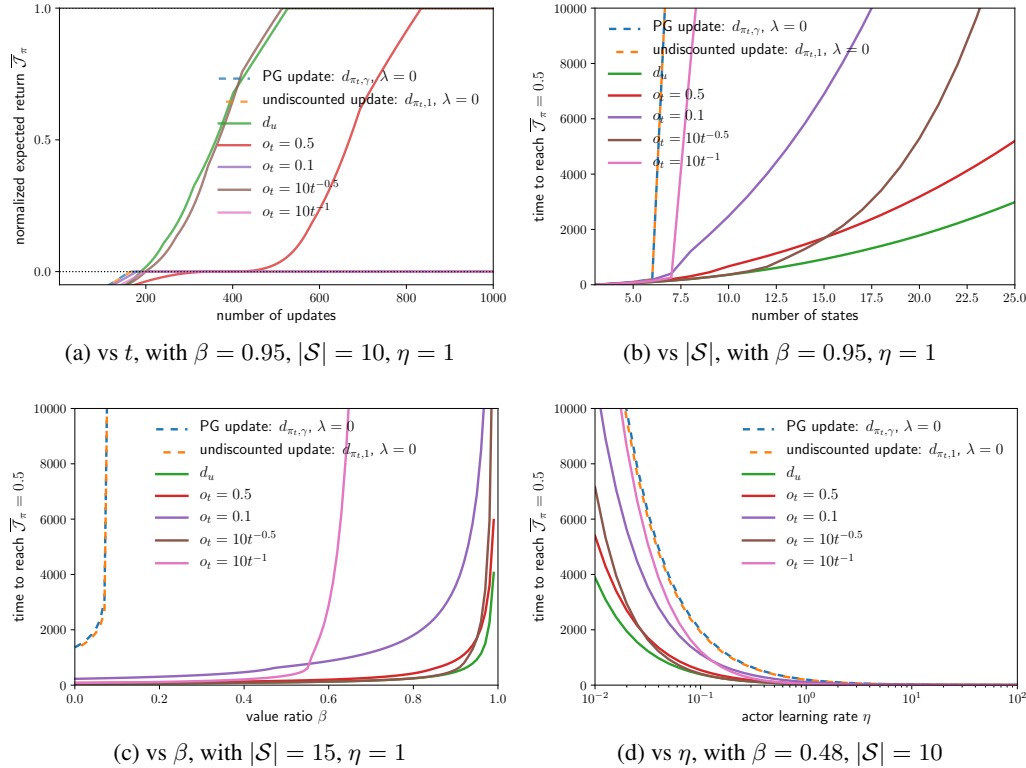

(a) vs $t$, with $\beta = 0.95$, $|\mathcal{S}| = 10$, $\eta = 1$

(b) vs $|\mathcal{S}|$, with $\beta = 0.95$, $\eta = 1$

(c) vs $\beta$, with $|\mathcal{S}| = 15$, $\eta = 1$

(d) vs $\eta$, with $\beta = 0.48$, $|\mathcal{S}| = 10$

Figure 5: Chain experiments with the direct parametrization under A1.

**Exp3:** Its results are displayed on Fig. 5c. We observe that the issues spotted in **Exp1** and **Exp2** is not due to the near-optimality of the low-hanging-fruit policy. Indeed, even with a reasonably short chain (15), and the low-hanging-fruit policy yielding an expected return 10 times smaller than optimal, the on-policy updates will still converge to it. In contrast, when the density requirement is satisfied, the updates reliably discover the optimal solution.

**Exp4:** Its results are displayed on Fig. 5d. We observe that the actor learning rate does not have any impact on the ability of the update to lead to an optimal policy[9]. Sill, making it larger speeds up convergence, so in the idealized setting we considered in this section where A1 is satisfied, the higher the actor learning rate, the better. And it has to be noted that setting it to infinity makes the update identical to a policy improvement step in the policy iteration algorithm.

### D.1.2 Softmax parametrization

Note that $\eta$ has been multiplied by 10 in the softmax experiments as compared to the direct ones, in order to get a rate of convergence comparable to that of direct parametrization.

**Exp1:** Its results are displayed on Fig. 6a. We observe similar results than for the direct parametrization, except that the algorithms never completely converge to the optimal (or suboptimal) policies. We also observe an interesting behaviour of the policy entropy regularized versions of the on-policy updates, at first converging to a policy worse than the low-hanging-fruit one. This stems from the bias induced by the policy entropy regularization: during a first stage it converged to an entropy regularized policy close to the low-hanging-fruit policy, then, after update 2000, it discovered the low-hanging-fruit policy and slowly converged to it.

**Exp2:** Its results are displayed on Fig. 6b. We observe exactly the same results as in the direct parametrization.

---

[9]We do not show the results, but on-policy updates never find the optimal policy when $\beta = 0.7$ for any $\eta$.

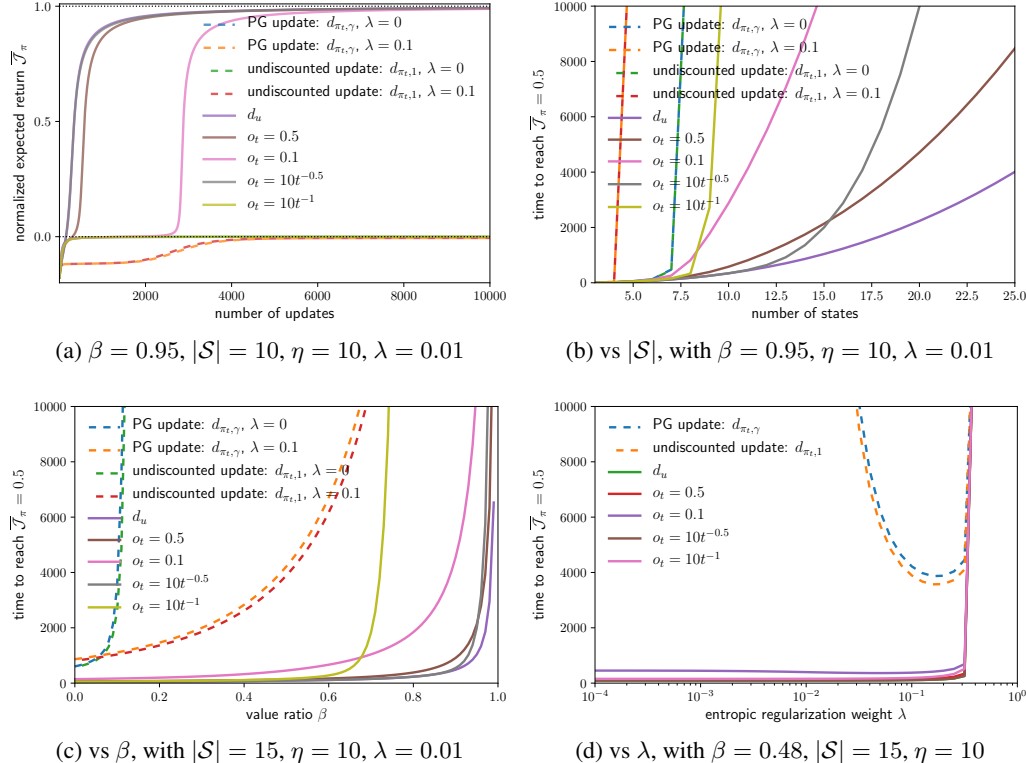

(a) $\beta = 0.95$, $|\mathcal{S}| = 10$, $\eta = 10$, $\lambda = 0.01$     (b) vs $|\mathcal{S}|$, with $\beta = 0.95$, $\eta = 10$, $\lambda = 0.01$

(c) vs $\beta$, with $|\mathcal{S}| = 15$, $\eta = 10$, $\lambda = 0.01$     (d) vs $\lambda$, with $\beta = 0.48$, $|\mathcal{S}| = 15$, $\eta = 10$

Figure 6: Chain experiments with the softmax parametrization under A1.

**Exp3:** Its results are displayed on Fig. 6c. We get results very similar to those of the direct parametrization. The policy entropy regularized updates behaviour is however quite interesting. It seems quite efficient at helping the on-policy updates find the best policy, but the introduced bias prevents learning it when $\beta$ is too high. This last remark gets very clear by looking at Fig. 6b where policy entropy regularized updates perform terribly because $\beta$ is high, even worse than the on-policy updates without the policy entropy regularization. It is also worth noting that this is the first experiment where there is a perceptible difference between on-policy updates, with the undiscounted update getting a little edge ($\gamma = 0.99$ is rather high).

**Exp5:** Its results are displayed on Fig. 6d. The first thing to notice is that, for all updates, when the policy entropy regularization gets large, $\lambda \approx 0.25$, the introduced bias is so high that the optimal regularized policy has a normalized performance lower than 0.48 (similar to what we observed on Fig. 6a). We also notice, that, for $\lambda < 0.25$, its impact on updates that satisfy C4 is near to null. Finally, for the on-policy updates, there is a sweet spot where it allows them to converge in the vicinity of the optimal policy reasonably fast, but it is rather narrow in terms of $\lambda$ values, and slow in terms of time to find the optimal policy. And finally, let us recall that it will not converge to the optimal policy but to a stochastic policy that is optimal under the policy entropy regularized objective. We elaborate this last point further in Appendix A.5.

### D.2 Random MDPs domain

The Random MDPs experiment described in Section C.2 is intended to test the updates in random MDPs, that have not been designed for the on-policy updates to fail, and where they are known to be efficient. We will consider a single experiment consisting in repeating the following process for more than 100 runs:

1. Generate a Random MDP.
2. Run the policy optimization in this MDP with the same densities as those described in Section D.1.

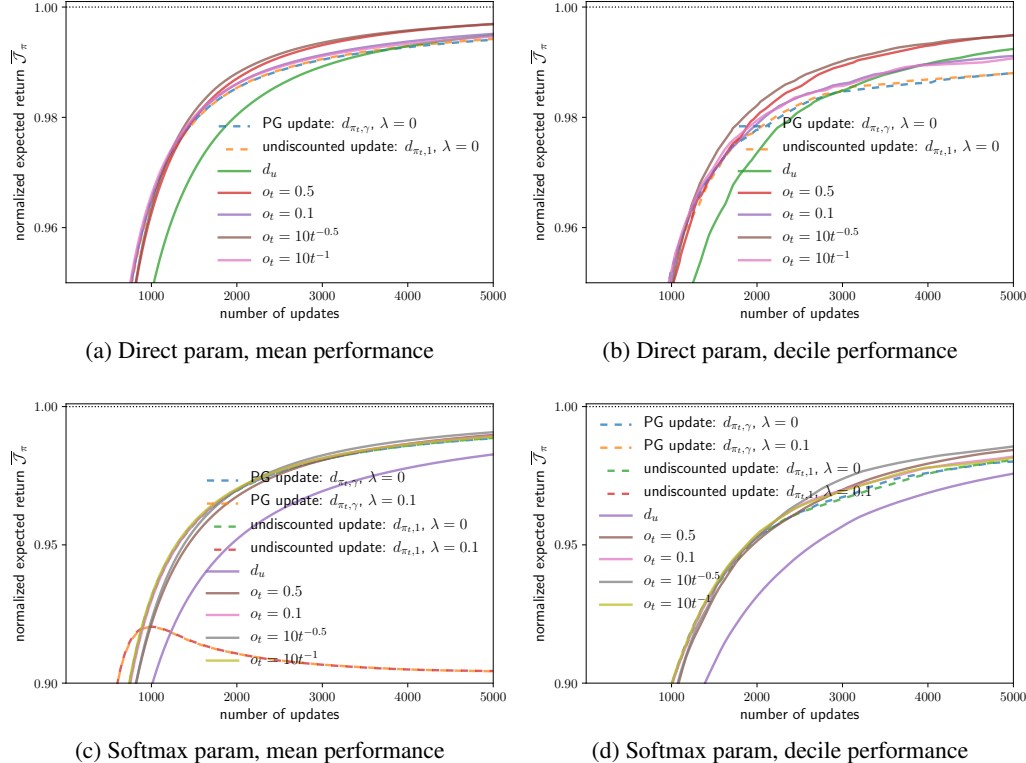

(a) Direct param, mean performance

(b) Direct param, decile performance

(c) Softmax param, mean performance

(d) Softmax param, decile performance

Figure 7: Random MDPs experiments (50 states, 4 actions) under A1.

We then report the mean (resp. decile) of the normalized expected return over the runs, as the mean (resp. decile) performance. We run the same experiment with both the direct and softmax parametrizations.

### D.2.1 Direct parametrization

The mean performance is displayed on Fig. 7a. All algorithms perform almost equally good. Indeed, in absence of purposely designed deceptive rewards, the policy gradient and the undiscounted update are rather reliable at finding strong policies. Still, we start observing from update 2,000 on that the updates with low off-policy $o_t$ updates (*i.e.* small weight on the uniform policy) start losing ground to the ones with high off-policiness $o_t$. The notable exception is $d_u$ that suffered some lateness at the start of curve, but is catching up and overtaking the low off-policy $o_t$ curves around the 4,000th update. This suggests that a proper scheduling of $o_t$ might be worth studying in the future: in early stages, follow the true gradient and increase exploration when the value improvement slows down.

The decile performance is displayed on Fig. 7b. We observe that, in addition to yielding a higher mean performance, high off-policy $o_t$ updates are also very stable: their decile performance is very close to their mean performance. In contrast, the decile performance of the on-policy updates seem to stagnate around 0.99 at the end of the curve.

### D.2.2 Softmax parametrization

The mean performance is displayed on Fig. 7c. The results are similar to that with the direct parametrization except that:

- the mean performance remains significantly further from the optimal (around 0.99 here vs 0.997 with direct parametrization). This is due to the fact that softmax parametrization takes longer to converge.

- the gap between high and low off-policy $o_t$ updates is much thinner. We believe this might also be due to the slowness of the softmax parametrization convergence. As a consequence, $d_u$ remains far from the other updates.

- the policy entropy regularized versions of the on-policy updates perform much worse. Indeed, the bias introduced by the policy entropy regularization impairs the convergence to a policy that is near optimal with respect to the normalized expected return (which is our objective function). Nevertheless, it has to be noted that it produced the faster improvement in the early stages (until update 1,000). We do not have an explanation for that.

The decile performance is displayed on Fig. 7d. We observe the same behavior as with the direct parametrization, with the exception of the narrower gaps already discussed above.

## D.3   General conclusion for this set of experiments

We test the performance across time, actor learning rate $\eta$, MDP parameters $|\mathcal{S}|$ and $\beta$, off-policiness $o_t$, and policy entropy regularization $\lambda$, with both direct and softmax parametrizations, on both the chain and random MDPs. The full report is available in Appendix D.

On the chain domain, we empirically confirm that enforcing updates with $d_t$ including an off-policy component helps path discovery and policy planning, while on-policy updates fail at converging to the optimal policy, even with policy entropy regularization, in a reasonable amount of time. By playing with the chain domain hyperparameters $\beta$ and $|\mathcal{S}|$, we observe that on-policy updates are sensitive to both: even with a small $\beta = 0.1$, a reasonable sized chain $|\mathcal{S}| = 15$ cannot be solved. In contrast, updates that are partly off-policy allow to converge fast to optimality even with $\beta = 0.95$ and $|\mathcal{S}| = 25$.

On the random MDP domain, we empirically observe that discounted and undiscounted updates perform well but still oftentimes stagnate at 99% of optimal performance, while $d_t$ including an off-policy component achieves even closer to optimality. Finally, we also note that a purely uniform $d_t$ slows down training in the random MDPs experiment. We conclude that it is best to include both on-policy and off-policy components in $d_t$. Experiments also allowed us to observe the biased convergence implied by the policy entropy regularization.

# E Full report of finite MDPs reinforcement learning experiments

## E.1 Algorithms design

This section provides the full implementation of the algorithms on the testbed.

---

**Input:** UCB reward bonus weight $\nu$ and policy entropy regularization weight $\lambda$.
**Input:** Initialization of the value $q_0$, set actor learning rate $\eta$ and critics learning rate $\eta_c$.
1: Initialize parameters $\theta = 0$ of the actor $\pi \doteq \pi_\theta$ and the critic $q \doteq q_0$.
2: **for** $t = 0$ to $\infty$ **do**
3:      Sample a transition $\tau_t = \langle s_t, a_t \sim \pi_b(\cdot|s_t), s_{t+1} \sim p(\cdot|s_t, a_t), r_t \sim r(\cdot|s_t, a_t)\rangle$.
4:      **for** once, do a single on-policy update:
5:          Update critic $q$ with a SARSA update on $\tau_t$:

$$q(s_t, a_t) \leftarrow q(s_t, a_t) + \eta_c \left( r + \gamma \sum_{a' \in \mathcal{A}} q(s', a')\pi(s', a') - q(s, a) \right) \tag{182}$$

6:          Perform a stochastic update step in state $s$ on Dr Jekyll's actor $\theta$:

$$\theta_{s_t, a_t} \leftarrow \theta_{s_t, a_t} + \eta d \left( r + \nu \sqrt{\frac{\log t}{n_{s_t, a_t}}} + \lambda \log \pi(a_t|s_t) + \sum_{a' \in \mathcal{A}} \pi(s'_y, a')q(s'_t, a') \right) \nabla_\theta \pi(a_t|s_t) \tag{183}$$

7:      **end for**
8: **end for**

---

Algorithm 2: On-policy algorithms for experiments in finite MDPs (by default $\nu = 0$ and $\lambda = 0$).

We first formally describe the on-policy algorithms in Algorithm 2. PG update has $d$ equal to $\gamma$ at the power of the episode's timestep, while $d = 1$ for undiscounted updates. Unless specified otherwise UCB exploration bonus parameter $\nu$ and policy entropy regularization parameter $\lambda$ are set to 0.

We then formally describe the implementation of J&H in Algorithm 3.

Not shown in the algorithms: in order to compare equally each update (for instance, discounted updates would observe less update amplitude than undiscounted ones), we normalize the empirical update steps $d$ by the average of their values in history. This normalizing step is not useful for the algorithms themselves, its only use is to control potential side effects of the update choice.

In order to prepare for its deep RL implementation, we extensively analyse the J&H algorithm, but only on the softmax parametrization. Indeed, there is no true equivalent to direct parametrization with neural networks. We analysed the performance of the algorithms against time (number of updates), the off-policiness $o_t$, the actor learning rate $\eta$, and the parameters of the chain domain: its size and the value ratio $\beta$. We will look at the same hyperparameters in the Reinforcement Learning setting, but also at the newly introduced hyperparameters: the critic learning rate $\eta_c$, the $q$-function initialization $q_0$, and the exploration schedule $\epsilon_t$.

But first let us specify the default hyperparameter settings:

- Environment hyperparameters:

  - Chain domain: $|\mathcal{S}| = 10$, $|\mathcal{A}| = 2$, $\gamma = 0.99$, $\beta = 0.8$.
  - Random MDPs domain: $|\mathcal{S}| = 100$, $|\mathcal{A}| = 4$, $\gamma = 0.99$.

- Algorithms hyperparameters:

  - All algorithms: $\eta = 1$, $\eta_c = 0.1$, $q_0 = 0$, $\lambda = 0$, $\nu = 0$.
  - J&H specific: $\epsilon_t$ and $o_t$ are always shown.
  - In all the experiments, Mr Hyde is trained with $Q$-learning on a $\gamma$ discounted objective from UCB rewards: $\tilde{r}(s, a) \doteq \frac{1}{\sqrt{n_{s,a}}}$.

**Input:** Scheduling of exploration $(\epsilon_t) = \frac{\epsilon_1}{t^{\alpha_\epsilon}}$, and of off-policiness $(o_t) = \frac{o_1}{t^{\alpha_o}}$.

**Input:** Initialization of the value $q_0$, set actor learning rate $\eta$ and critics learning rate $\eta_c$.

1: Initialize $\mathring{D} = \emptyset$, parameters $\mathring{\theta} = 0$ of Dr Jekyll's actor $\mathring{\pi} \doteq \pi_{\mathring{\theta}}$, and critic $\mathring{q} \doteq q_0$.

2: Initialize $\tilde{D} = \emptyset$ and Mr Hyde's value to $\tilde{q} = q_0$, and policy $\tilde{\pi}$ to uniform.

3: Set the behavioural policy to Dr Jekyll: $\pi_b \leftarrow \mathring{\pi}$ and the working replay buffer $D_b \leftarrow \mathring{D}$.

4: **for** $t = 0$ to $\infty$ **do**

5:     Sample a transition $\tau_t = \langle s_t, a_t \sim \pi_b(\cdot|s_t), s_{t+1} \sim p(\cdot|s_t, a_t), r_t \sim r(\cdot|s_t, a_t)\rangle$

6:     Add it to the working replay buffer $D_b \leftarrow D_b \cup \{\tau_t\}$.

7:     **if** $\tau$ was terminal, **then** $(\pi_b, D_b) \leftarrow (\tilde{\pi}, \tilde{D})$ w.p. $\epsilon_t$, $(\pi_b, D_b) \leftarrow (\mathring{\pi}, \mathring{D})$ otherwise.

8:     **for** once, do a single update:

9:         $\tau \doteq \langle s, a, s', r\rangle \sim \tilde{D}$ w.p. $o_t$, $\tau \doteq \langle s, a, s', r\rangle \sim \mathring{D}$ otherwise.

10:        Update Mr Hyde's critic $\tilde{q}$ with a q-learning update on $\tau$, then $\tilde{\pi}$ is greedy on $\tilde{q}$:

$$\tilde{q}(s,a) \leftarrow \tilde{q}(s,a) + \eta_c \left( \frac{1}{\sqrt{n_{s,a}}} + \gamma \max_{a' \in \mathcal{A}} \tilde{q}(s',a') - \tilde{q}(s,a) \right) \quad \text{then} \quad \tilde{\pi} \leftarrow \operatorname*{argmax}_{a \in \mathcal{A}} \tilde{q} \quad (184)$$

11:        Update Dr Jekyll's critic $\mathring{q}$ with a SARSA update on $\tau$:

$$\mathring{q}(s,a) \leftarrow \mathring{q}(s,a) + \eta_c \left( r + \gamma \sum_{a' \in \mathcal{A}} \mathring{q}(s',a')\mathring{\pi}(s',a') - \mathring{q}(s,a) \right) \quad (185)$$

12:        Perform an expected update step in state $s$ on Dr Jekyll's actor $\mathring{\theta}$:

$$\forall b \in \mathcal{A}, \qquad \mathring{\theta} \leftarrow \mathring{\theta} + \eta \mathring{q}(s,b)\nabla_{\mathring{\theta}}\mathring{\pi}(b|s) \quad (186)$$

13:     **end for**

14: **end for**

Algorithm 3: Dr Jekyll & Mr Hyde algorithm for experiments in finite MDPs.

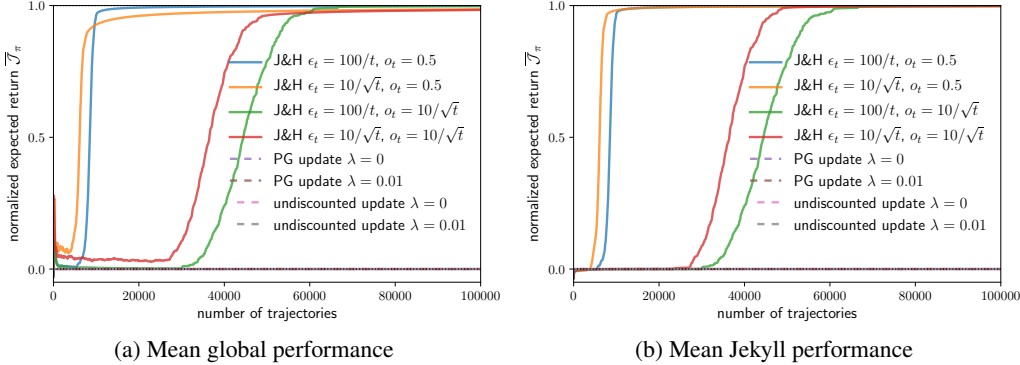

(a) Mean global performance            (b) Mean Jekyll performance

Figure 8: Chain MDPs experiments vs. number of trajectories (200+ runs).

For J&H, we generally report the *global* performance, *i.e.* the expected performance of the mixture of Dr Jekyll and Mr Hyde, but also sometimes, when mentioned, we also look at the performance of the Dr Jekyll policy, because it is the best policy known to the algorithm at a given time.

### E.2   Vs. time (number of trajectories):

In the chain experiment, we expect J&H to perform much better than on-policy updates (policy gradient and undiscounted), since they are known to easily converge to suboptimal solutions. This is indeed what we observe on Fig. 8a. All versions of J&H eventually converge, in all runs, to the optimal policy, while on-policy updates, with or without entropic regularization, never converged to the optimal solution. Looking more closely at the different J&H variants, we notice that the off-policiness is instrumental to make the convergence faster, more than the exploration ratio $\epsilon_t$, because this latter cannot be set too large in order not to compromise the global performance. Indeed, we

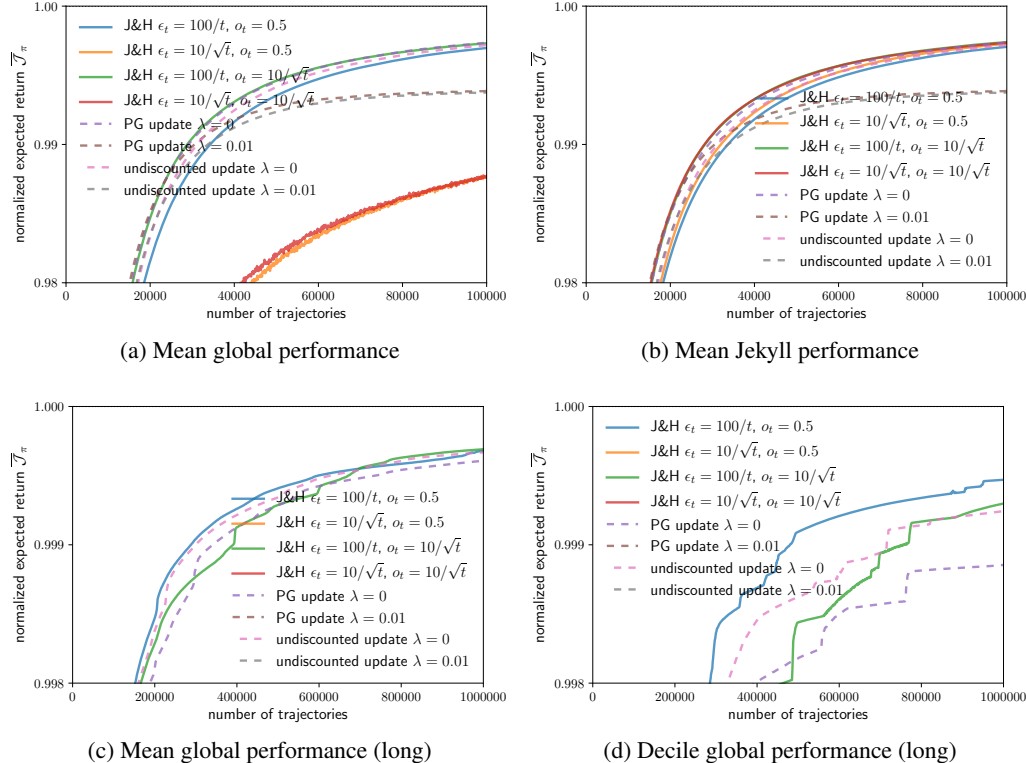

(a) Mean global performance

(b) Mean Jekyll performance

(c) Mean global performance (long)

(d) Decile global performance (long)

Figure 9: Random MDPs experiments vs. number of trajectories.

already observe with $\epsilon_t = \frac{10}{\sqrt{t}}$ that exploration prevents from converging closely to optimal in global performance. We may confirm that Dr Jekyll converges well to optimal on Fig. 8b.

In the random MDPs domain, we do not expect to see any improvement, the on-policy updates (policy gradient and undiscounted) are not *tricked* to converge to a suboptimal solution. Moreover, the high stochasticity of the domain, makes it so that every state is visited a lot of times even when following a deterministic policy. Fig. 9a confirms that J&H can keep up with the speed of the true gradient. Its versions with $\epsilon_t = \frac{10}{\sqrt{t}}$ suffer a low global performance because of its exploration that remains high, as Fig. 9b confirms. We notice that entropic regularization does not help training faster and also biases the training objective which results into a plateaued performance. Finally, we may observe that the choice of $o_t$ has a slight impact: using half Dr Jekyll, half Mr Hyde samples to train offline slows a bit the convergence in the first 100k trajectories, however, looking at a longer horizon on Fig. 9c shows that it gets better later[10]. It is also interesting to notice that the gap to optimal performance has decreased by a factor 10 with 10 times more data, which corroborates our theoretical findings that the regret should be in $\mathcal{O}(t^{-1})$. Finally, the decile performance across time shown on Fig. 9d suggests that the use of strong off-policiness mitigates the risk and improves the worst case performance.

### E.3 Vs. off-policiness $o_t$:

Off-policiness being a hyperparameter of J&H alone, we report the results for J&H only. On Fig. 10a, we observe that the higher the off-policiness, the best better it is in the chain experiment. However, it is quite the opposite in the random MDPs experiments results shown on Fig. 10b, even though it is less critical. The scheduling of $o_t$ would be an interesting focus for further research. At this point of our knowledge we conjecture that it is better to train on-policy when the expected value improvement are high, and train more off-policy when it is small.

---

[10]Note that despite using more than 100 runs to compute these curves, there remains some instability and the visible results are probably not significant enough to elect a winner, however they are enough to see that they perform comparably.

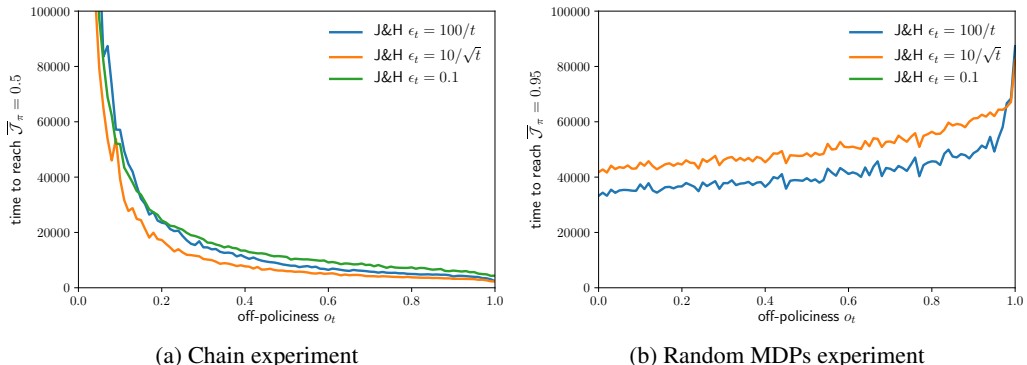

(a) Chain experiment

(b) Random MDPs experiment

Figure 10: Experiments vs. off-policiness $o_t$: average time to reach a global performance objective.

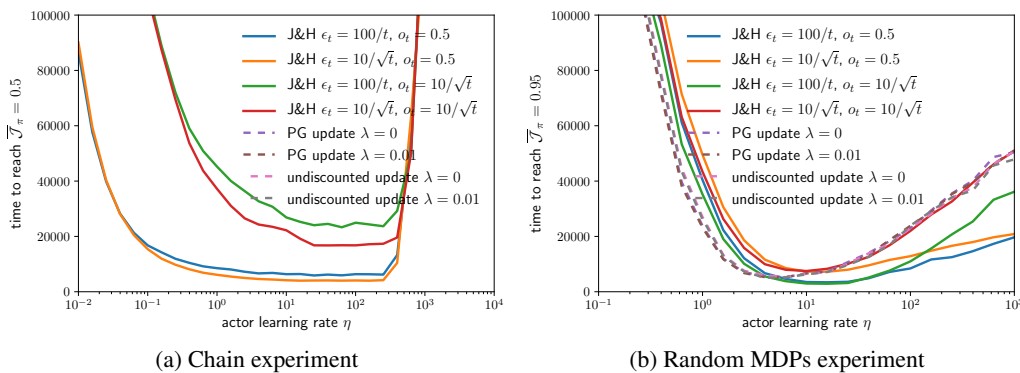

(a) Chain experiment

(b) Random MDPs experiment

Figure 11: Experiments vs. actor learning rate $\eta$: average time to reach a global performance objective.

### E.4 Vs. learning rates $\eta$ and $\eta_c$:

The study of $\eta$ on the chain domain (results on Fig. 11a reveals that the performance of J&H improves until some value where it breaks. It looks like a numerical stability issue but after checking it does not seem like anything of the sort, rather that the softmax parameters go faster away than they can go back. Indeed, as our theoretical analysis shows, there is a factor in $\pi(1-\pi)$ the updates that slows down the recovery from a run that converged too far into a suboptimal policy. Actually, this phenomenon has been observed before on policy gradients, *e.g.* on Fig. 1.d of [22]. Note that the value of this hyperparameter does not allow to help the on-policy algorithms to find the optimal solution. We made the similar experiment on easier settings (lower $\beta$ and size), and this did not improve (not shown). Note also that the default value for the actor learning rate has been set to 1 which is not its optimal value, but it was a better value to make the critic/actor learning rates consistent with each other.

The experiment on the Random MDPs shown on Fig. 11b is one of the most interesting empirical results for many reasons[11]. Firstly, we observe that all updates admit an optimal value for the actor learning rate in the range [2,30]: rather high, but not too high. This shows that there is a value for not updating too fast the policy. Secondly, off-policiness is more tolerant for higher $\eta$ values: its optimal value is obtained for $\eta \approx 20$, while on-policy updates are best with $\eta \approx 5$. Thirdly, when $\eta$ is optimally set for each update, J&H significantly outperforms on-policy updates ($\approx 2,500$ vs. 3,500). Finally, it is satisfying to observe that $\eta \approx 20$ is a also optimal in the chain experiment, suggesting that it may not be too much domain dependent.

Since the chain domain is deterministic, we would expect a high critic learning rate $\eta_c$ to be better and this is what we observe, but even a maximal critic learning rate of 1 is not enough for the on-policy updates with or without policy entropy regularization to solve the chain is the default settings

---

[11]We notice that the default value we used for this hyperparameter is not favorable for J&H.

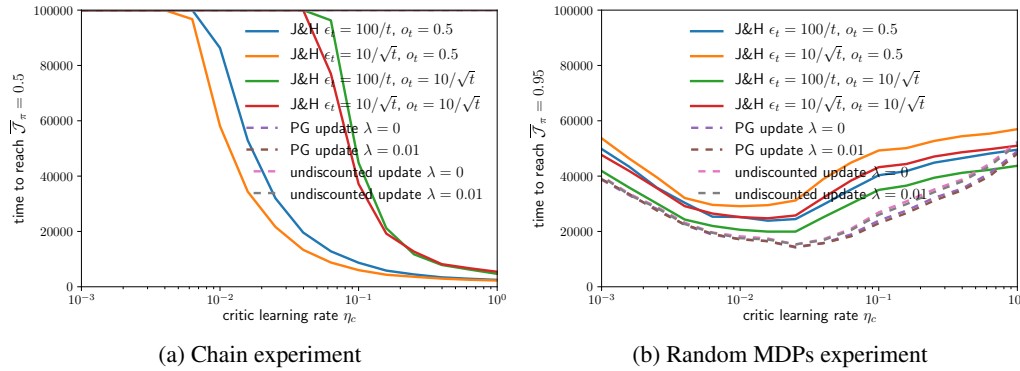

(a) Chain experiment

(b) Random MDPs experiment

Figure 12: Experiments vs. critic learning rate $\eta_c$: average time to reach a global performance objective.

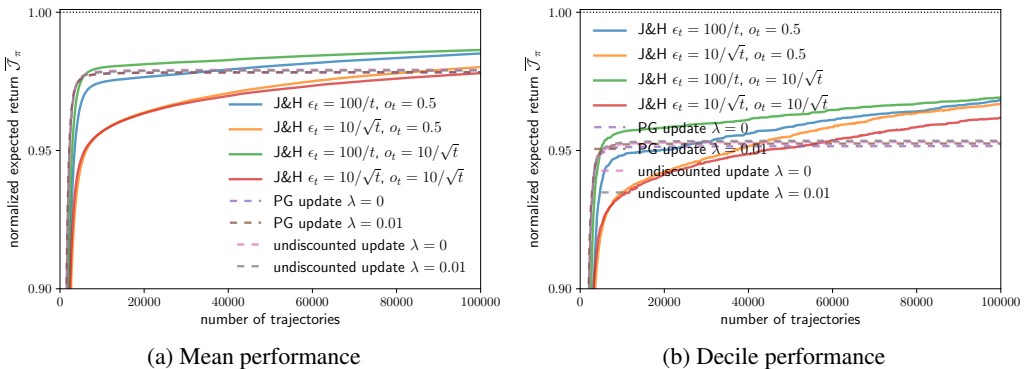

(a) Mean performance

(b) Decile performance

Figure 13: Random MDPs experiments with $\eta = 10$ and $\eta_c = 0.025$.

(size of 10 and $\beta = 0.8$). We also not that J&H is much more tolerant to a small critic learning rate when the off-policiness is kept high. The random MDP experiment (Fig. 12b) illustrates the risk of setting a too high critic learning rate. A small learning rate implies a longer but steadier time to converge, while a high learning rate might compromise the convergence at all.

One must keep in mind that these insights are for reaching a normalized performance of 0.95 in the random MDPs experiments. We tried a run with $\eta = 10$ and $\eta_c = 0.025$ for all algorithms, since they seem to all perform well with this setting in our hyperparameter search we have just presented. Fig. 13 displays the results. It clearly appears that reaching $\overline{\mathcal{J}}_\pi = 0.99$ (let alone 0.999) would take much longer than with our default setting. We judged better to stick with our default settings for this reason.

### E.5 Vs. parameters of the chain domain:

On Fig. 14a, we vary the size of the chain, and observe that J&H algorithm scales well to it as long as the off-policiness is maintained long enough for it to unlearn to suboptimal policy and move on to the optimal policy. It is worth noting that the on-policy updates fail at finding even with its easiest hyperparameters, indeed the $q$-function being initialized to 0 and the critic learning rate being of 0.1 makes it that the algorithms already converged to the suboptimal policy when the later states values are properly estimated. In fact, the policy entropy regularized versions of the on-policy updates reach $\overline{\mathcal{J}}_\pi = 0.48$ around the 150,000th episode.

On Fig. 14a, we vary the performance ratio $\beta$ and observe that the on-policy updates manage to make it work even for a chain of size 10 when $\beta$ is sufficiently small, and that the policy entropy regularization helps a bit to push further the limit, but not by much. In general, the delay occurred by the time necessary for the critic to be trained is fatal to these algorithms, hence the importance of

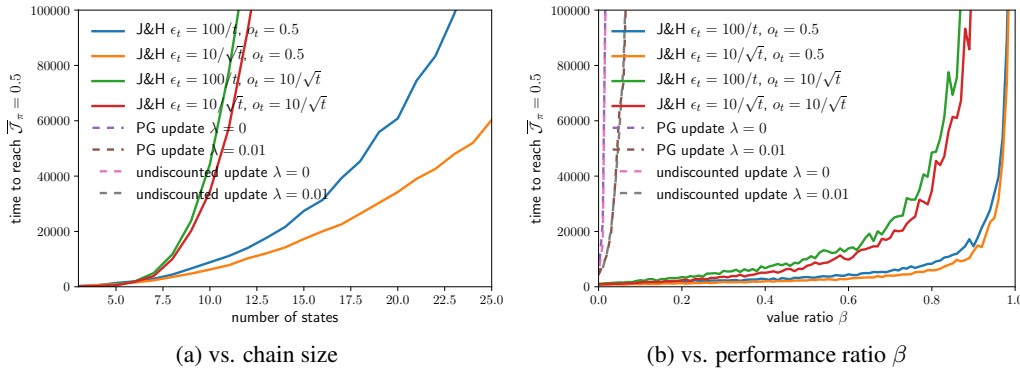

(a) vs. chain size

(b) vs. performance ratio $\beta$

Figure 14: Experiments vs. chain hyperparameters: average time to reach a global performance objective.

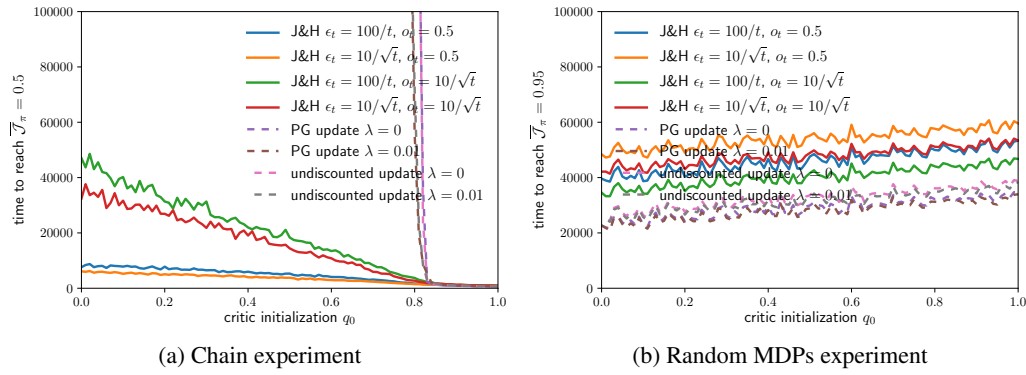

(a) Chain experiment

(b) Random MDPs experiment

Figure 15: Experiments vs. $q$-function initialization $q_0$: average time to reach a global performance objective.

prioritizing the replay of exploratory experiences. Fig. 14a also validates the reliability of the J&H algorithms, in particular with a strong off-policiness.

### E.6 Vs. $q$-function initialization $q_0$:

Initializing the $q$-function to high values has been a historic exploration scheme for $q$-learning, and we observe on Fig. 12a that is also works in the actor-critic world. As soon as $q_0$ gets larger than the value of the low-hanging-fruit policy, all updates push to explore until discovering the end of the chain. In contrast, J&H is not as much sensitive to the critic initialization, in particular when the off-policiness is high enough to fight against a hostile one.

### E.7 Vs. exploration schedule $\epsilon_t$:

Exploration schedule being a hyperparameter of J&H alone, we report the results for J&H only. In order to remove the dependence of the performance on its exploratory behaviour, contrarily to all the other experiments where we report the performance of J&H's dyad, we report here the time for Jekyll's policy to reach the normalized value.

On Fig. 16a, we observe that having a strong off-policiness alleviates the lack of exploration to a certain extent. We however want to reiterate that the chain domain is deterministic, and as such, performing off-policy updates is equivalent to exploring. In a stochastic environment such as the random MDPs, exploration cannot be replaced with off-policy updates, because statistical significance is required to find the optimal policy. With $\epsilon_t = 0$, unsurprisingly, the algorithm fails at finding the optimal reward and therefore finding the goal. From $\epsilon_t = 0.05$ to $\epsilon_t = 0.6$, the speed improves from 15000 to 6500. From $\epsilon_t = 0.6$ to $\epsilon_t = 0.95$, the speed remains more of less constant in the range [6400, 6500]. With $\epsilon_t = 1$, more surprisingly, the performance degrades quite significantly to 7300.

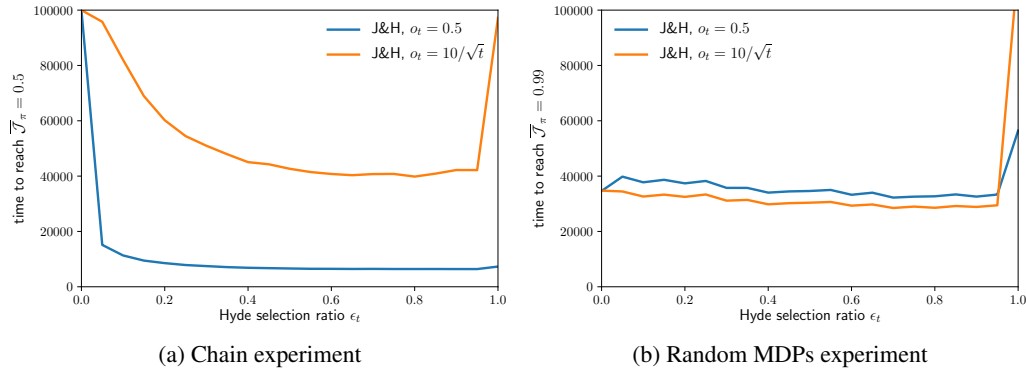

(a) Chain experiment             (b) Random MDPs experiment

Figure 16: Experiments vs. exploration schedule $\epsilon_t$: average time to reach a global performance objective.

The random MDPs experiment (Fig. 16b) displays similar behaviour at a different scale. From $\epsilon_t = 0$ to $\epsilon_t = 0.95$, the speed linearly improves from 35000 to 29000. With $\epsilon_t = 1$, again surprisingly, the performance degrades catastrophically to more than 100000.

We analyse the combined results as follows: introducing Hyde's exploratory behavior generates a large performance improvement due to a need for exploration. This need is correlated with the planning difficulty of the task, hence the much stronger effect in the chain experiment. Performance then slowly improves as a function of $\epsilon_t$: this is where the statistical significance matters. Finally, when $\epsilon_t$ equals 1, the on-policy updates do not exist anymore, slowing down significantly the convergence speed.

## E.8 General conclusion for this set of experiments

We test the performance across time, actor $\eta$ and critic $\eta_c$ learning rates, MDP parameters $|\mathcal{S}|$ and $\beta$, off-policiness $o_t$ and exploration $\epsilon_t$, and critic initialization $q_0$ with the softmax parametrization on both the chain and random MDPs. The full report is available in Appendix E.

In the chain experiments, we observe that J&H shows the same ability to converge to the optimal policy, while the on-policy updates, with or without policy entropy regularization, always fail, except when $q_0 > \mathcal{J}_\perp$ because it induces an exploratory behaviour that solves its limits. In contrast, J&H is efficient whatever $q_0$. More specifically about J&H, while the exploration $\epsilon_t$ is necessary at a small amount, the amplitude of off-policiness $o_t$ is what matters the most. Since the chain domain is deterministic, the critic learning rate $\eta_c$ is optimal set at 1. Regarding the actor learning rate, it is best set high too, but until some limit where it breaks the convergence.

Regarding the random MDPs experiments, the results are much closer, the on-policy updates being helped by the high stochasticity of the environment, providing indeed a sort of exploration. We still observe that all algorithms provide similar performance, J&H performing slightly worse when $o_t$ or $\epsilon_t$ increase. Similarly, since exploration is not necessary in this domain, the performance slightly decrease with $q_0$ for all algorithms. Regarding the critic learning rate, it is best chosen around $\eta_c = 0.025$. Maybe the most interesting result concern the dependency with respect to the actor learning rate. It seems that the off-policiness granted by J&H updates allows for higher $\eta$, which allows faster convergence when this hyperparameter is optimized independently for each algorithm. While expected, another interesting observation is that off-policiness seems to improve the worst case scenarios by allowing to recover more easily from converging to a suboptimal policy.

# F    Full report of deep reinforcement learning experiments

## F.1    The Four Rooms environment

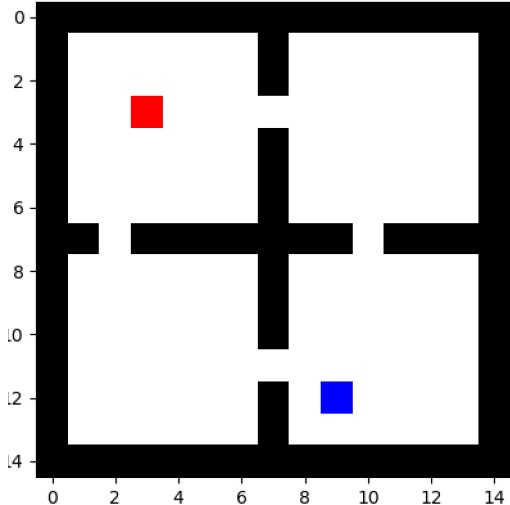

Figure 17: The Four Rooms environment, the agent (in blue) needs to navigate to the goal (in red).

Figure 17 shows an example of the environment. The agent (in blue) starts at a random place in one of the rooms. In Level 1, the initial state can be in any room, in Level 2, the initial state cannot be in the goal room. The agent then needs to navigate to the goal, using four actions, North, East, West, South. At each step in the environment, the agent receives a negative reward of $-0.1$. When reaching the goal, the agent receives a reward of $90$ and the episode terminates. If the agent hasn't reached the goal in 90 steps, the episode terminates. In the plots, we report the undiscounted sum of rewards, a quantity that belongs to $[-9, 90]$. The state consists of a random 20-dimensional embedding for each distinct position in the environment. The discount factor is set to $0.9$. After 10 episodes of training, we evaluate each agent during 10 episodes and report the average performance. Each agent was evaluated on 10 seeds, we report the average and one standard deviation over those 10 seeds. We trained the agents on an internal GPU cluster of P40 cards. Each training run took approximately 30 minutes.

## F.2    Implementation, architectures and hyperparameters

We based our code on the repo [8], that contains PyTorch implementations of various deep reinforcement learning algorithms. We used the default architectures from the repo, the Adam optimizer and performed small hyperparameters sweeps on learning rates for each algorithm (described below).

**DDQN**    We used the default implementation provided by the repo [8] for the Four Rooms environment. The q-network is a two-hidden layer MLP with 30 and 10 neurons in each. The replay buffer size is 40k and we used soft decay of the epsilon greedy exploration as well as soft updates of the target network. In terms of learning rate, we swept over $\{0.06, 0.03, 0.01, 0.006, 0.003, 0.001\}$ and found that $0.01$ performed best. We also found that decreasing the epsilon greedy decay compared to the default value in the repo gave better performance on Level 2 (it did not affect Level 1 much). We used the following probability for picking a random action: $100/e$, where $e$ is the number of episodes trained on so far (compared to 10 for the default).

**Soft-Actor Critic**    The actor and critics are two-hidden layers MLPs with 64 neurons in each. Two critics were trained, and the mininum between the two was used in the backup of the critics, as well as to train the actor. The entropy regularization hyperparameter was automatically tuned. The implementation is the standard one provided in [8]. We also swept over the same learning rate set and found that similarly $0.01$ performed best.

**RND** For the RND target network, we used a one-hidden layer MLP with 20 neurons, outputing a 20-dimensional feature vector. The predictor network contains (as is customary in RND implementations) an additional hidden layer with 20 neurons as well. We did not search over architectures. The learning rate is set to 0.003, which was found after a sweep on the set $\{0.1, 0.03, 0.01, 0.003, 0.001\}$.

**Dr Jekyll & Mr Hyde** Dr Jekyll's architecture is the same as the Soft-Actor Critic one described above. Its learning rate was set to 0.001, which was found to perform best out of the set $\{0.06, 0.03, 0.01, 0.006, 0.003, 0.001, 0.0006\}$. Mr Hyde's architecture and learning rate are the same as the DDQN architecture described above. The decay of the probability of picking Mr. Hyde (parameter $\epsilon_1$ in algorithm 4) is set to $4k$ for Level 1 and $40k$ for Level 2. Note that this corresponds to the number of steps in the environment, not the number of episodes played. Those values correspond approximately to the decay of the exploration in DDQN.

**DDQN+RND and SAC+RND** In the case of DDQN+RND, we simply augmented the reward of the environment using the exploration bonus provided by the RND network. For SAC+RND, we trained an additional critic on the exploration bonus and added it to SAC's actor update. In both cases, we weighted the exploration bonus by 1 afer sweeping over the set $\{0.1, 0.3, 0.6, 1.0, 1.3, 1.6\}$ and finding that it performed best in both cases.

### F.3 The detailed algorithm

Algorithm 4 describes the implementation used to produce Figure 3.

---

**Input:** Scheduling of exploration $(\epsilon_t) = \frac{\epsilon_1}{t}$.
1: Initialize Dr Jekyll's actor $\mathring{\pi} \doteq \pi_{\mathring{\theta}}$ and critic $\mathring{q} = q_{\mathring{\theta}}$.
2: Initialize Mr Hyde's q-network $\tilde{q} = q_{\tilde{\theta}}$ (its policy $\tilde{\pi}$ is greedy with respect to $\tilde{q}$).
3: Initialize the RND predictor $p_{RND}$ and target $t_{RND}$ networks.
4: Initialize a buffer $F = \emptyset$ of capacity $K = 128$ and a buffer $\tilde{D} = \emptyset$ of capacity $40k$.
5: Set the behavioural policy to Dr Jekyll: $\pi_b \leftarrow \mathring{\pi}$ and the working replay buffer $D_b \leftarrow \mathring{D}$.
6: **for** $t = 0$ to $\infty$ **do**
7:     Sample a transition $\tau_t = \langle s_t, a_t \sim \pi_b(\cdot|s_t), s_{t+1} \sim p(\cdot|s_t, a_t), r_t \sim r(\cdot|s_t, a_t) \rangle$.
8:     **if** Mr Hyde is in control ($\pi_b = \tilde{\pi}$), **then** add $\tau_t$ to Mr Hyde's replay buffer $\tilde{D}$ .
9:     Add $\tau_t$ to buffer $F$.
10:     **if** $\tau_t$ was terminal, **then** $\pi_b \leftarrow \tilde{\pi}$ wp $\epsilon_t$, $\pi_b \leftarrow \mathring{\pi}$ otherwise.
11:     Initialize minibatch with FIFO buffer: $B \leftarrow F$
12:     **for** $i = 1 \rightarrow K$
13:         Add $\tau_i \doteq \langle s, a, s', r \rangle \sim \tilde{D}$ to minibatch $B$
14:     **end for**
15:     For each transition $\tau_i \in B$, compute RND reward $\tilde{r}_{\tau_i} = \|p_{RND}(s') - t_{RND}(s')\|^2$.
16:     Update Mr Hyde's q-network $\tilde{q}$ with a DDQN update on $B$ where $\tilde{r}$ is used instead of $r$

$$\tilde{q}(s, a) \leftarrow \tilde{q}(s, a) + \eta_c \left( \tilde{r}(s, a) + \gamma \max_{a' \in \mathcal{A}} \tilde{q}(s', a') - \tilde{q}(s, a) \right). \tag{187}$$

17:     Update Dr Jekyll's critic $\mathring{q}$ with a SARSA update on $B$:

$$\mathring{q}(s, a) \leftarrow \mathring{q}(s, a) + \eta_c \left( r + \gamma \sum_{a' \in \mathcal{A}} \mathring{q}(s', a')\mathring{\pi}(s', a') - \mathring{q}(s, a) \right). \tag{188}$$

18:     Perform an expected update step in each state $s$ of $B$ on Dr Jekyll's actor $\mathring{\theta}$:

$$\forall b \in \mathcal{A}, \qquad \mathring{\theta} \leftarrow \mathring{\theta} + \eta \mathring{q}(s, b) \nabla_{\mathring{\theta}} \mathring{\pi}(b|s) \tag{189}$$

19:     Update $p_{RND}$ by gradient descent on $\sum_{s' \in B} \|p_{RND}(s') - t_{RND}(s')\|^2$.
20: **end for**

Algorithm 4: Dr Jekyll & Mr Hyde algorithm for experiments with a deep architecture.

# G Code description

The code for all our experiments can be found at `https://github.com/microsoft/Dr-Jekyll-and-Mr-Hyde-The-Strange-Case-of-Off-Policy-Policy-Updates`.

## G.1 Finite MDPs code

It relies on two main files:

- `main_config.py` to run experiments in a fixed setting and report average/quantile performance across time,
- and `main_hyperparam.py` to run experiments where a hyperparameter varies, and report the time to reach a fixed normalized performance target $\tilde{\mathcal{J}}$ (see Eq. (180) and (181)).

They both use the same configuration file determining the setting the experiment (the hyperparameter search overwrites the setting of this hyperparameter during the run): which setting (planning or RL), which environment (chain or random MDPs), which algorithms, with which hyperparameters, for how many steps, and for how many runs. Such configuration files may be found in the `expes` folder, where the settings of all reported experiments may be retrieved as well as their figures (and some more). Nevertheless, we deleted the experiment result files because they are voluminous (10s of Go), but we have them and can provide them on demand, or better: their random seeds.

The two environments are implemented in python files:

- `chain.py`: the chain environment described in App. C.1,
- and `garnets.py`: the random MDPs environment described in App. C.2 [18].

Two utility python files are used:

- `utils.py` contains all utility files,
- except `proj_simplex.py` implementing the projection on the simplex, that has been isolated in order to account for their authors [5].

Finally, all the algorithmic innovation that we claim belongs to `gradient_ascent.py`, including the implementation of J&H (Alg. 3).

## G.2 Deep J&H implementation

The implementation is based on the repository [8], released under the MIT License and which contains various PyTorch implementations of standard deep reinforcement learning algorithms. The entry point to train the agents is the `results/Four_Rooms.py` file. To train a given one, simply run:

```
python results/Four_Rooms.py -agent {agent} -level {level}
```

where agent is chosen in [JH_Discrete, SAC_Discrete, SAC_DiscreteRND, DDQN, DDQNRND] and level is 1 or 2.