# OpenReview forum: "Dr Jekyll & Mr Hyde: the strange case of off-policy policy updates"
_NeurIPS.cc/2021/Conference — NeurIPS 2021 Poster_

### Official Review · Reviewer_hpwC · 2021-07-14

**Rating:** 9
**Confidence:** 3

**Summary:**

The paper extends the policy gradient theorem from purely on-policy gradients to arbitrary state distributions. The authors provide proofs of convergence and (improved) rate of convergence under milder assumptions than the original policy gradient theorem. They propose a simple agent that separately performs on-policy and off-policy updates, and include a carefully-designed toy experiment that elegantly shows their algorithm achieve globally-optimal results when traditional policy gradient approaches fail. They also demonstrate their approach on a simple deep-RL benchmark.


**Limitations And Societal Impact:**

The paper deals with basic research and doesn't directly have societal impact.

**Main Review:**

## Novelty

The work is novel. It brings together a collection of previous work and resolves some open questions in the theory of policy gradient methods. It provides theoretical grounding for the long-established engineering technique of using off-policy updates to improve RL performance, an algorithm design with convergence guarantees, and more.

The authors clearly identify related work and place it in context in their paper.

## Quality

This is a comprehensive look at the theoretical underpinnings of off-policy policy gradient methods. Claims are systematically introduced, proved, and eventually experimentally verified. (In the appendix) the proofs are clearly laid out, with proof sketches explaining complicated proofs as necessary.

The paper also includes the chain-MDP experiment that elegantly demonstrates how their method learns the globally optimal policy when traditional policy gradient approaches fail. This experiment shows the theoretical advantages manifest in practice. They also demonstrate their method in a simple Deep-RL setting.

The authors clearly identify and discuss limitations of their work. The greatest limitation of the work is that convergence proofs don’t necessarily cover the deep RL case, but that is unsurprising.

## Clarity

The paper is written very clearly, and is a pleasure to read. At each stage of the derivation, the importance, implications, and potential pitfalls are identified. The derivation of their final result proceeds logically and methodically from related work.

(As a small suggestion, consider adding a brief explanation/proof sketch of Theorem 2 in the main text, because it is a key theorem and its truth is not obvious. It is explained well in the appendix.)

## Significance

The results are important, both because they provide theoretical grounding for common RL engineering strategies, and because it is likely to lead to a long line of further work in applying their results to more complex domains.


**Time Spent Reviewing:**

5

---

> ### Author Response · Authors · 2021-08-09
> **Response to review**
>
> Thank you for your positive review (with which we totally agree :D). We sincerely hope this paper opens a fruitful line of work.
>
> *The greatest limitation of the work is that convergence proofs don’t necessarily cover the deep RL case, but that is unsurprising.*
>
> Indeed. At first, our paper was meant to be almost exclusively theoretical. But in the end its theoretical findings explained some failure modes of policy gradient algorithms, and suggested algorithmic changes that led to significant performance improvement.
>
> *As a small suggestion, consider adding a brief explanation/proof sketch of Theorem 2 in the main text, because it is a key theorem and its truth is not obvious. It is explained well in the appendix.*
>
> This was our intention as well, but lack of space forced us to move our proof sketches to the appendix. We will take advantage of the extra page in the camera-ready version (in case of acceptance) to reintegrate some of them, in particular for Theorems 1 and 2.

---

### Official Review · Reviewer_rE42 · 2021-07-16

**Rating:** 7
**Confidence:** 3

**Summary:**

In this paper the authors propose an extension to policy gradient theory which considers policy updates under any state distribution density. Moreover, they present the necessary conditions for convergence to optimality under their new formulation. Subsequently, they implement the ideas supported by their theory and construct a new agent which they call J&H (Dr Jekyll and Dr Hyde). The main idea of this agent is to consider two policies one which is fully exploitative (Jekyll) and one which is completely exploratory (Hyde). This way during training they can combine on-policy and off-policy updates. Finally, they test their agent in a series of simple MDPs and show that their proposed agent is superior at  recovering from converging to a suboptimal policy.

**Limitations And Societal Impact:**

Yes

**Main Review:**

The main contribution of the paper is a new theoretical framework for designing  new actor critic methods which are more robust to the common problem of converging to suboptimal policies. By defining an update rule which can be applied under any state density, it is possible to combine data from exploratory policies which cover the state space and thus avoid early convergence to suboptimal policies. The authors show an example of such an approach with their J&K algorithm.

In their analysis the authors consider only the direct and softmax parameterization of the policy to prove their theoretical claims. This is a limitation which the authors acknowledge and is left as a future work, while they also provide empirical results to support the claim over the
generality of their method.

Minor Comments
line 35: designates ?
line 40: Define n_t before using it
line 75: Degine adv before using it
line 676: Lemmas are denoted as Theorems a, b.

**Time Spent Reviewing:**

3

---

> ### Author Response · Authors · 2021-08-09
> **Response to review**
>
> Thank you for your time reviewing our paper. We have implemented fixes to your minor comments.
>
> Could you maybe also suggest ways in which we can improve it? It is not clear from your review what motivated giving it only a 6.

---

### Official Review · Reviewer_T2tW · 2021-07-16

**Rating:** 7
**Confidence:** 4

**Summary:**

The paper addresses exploration with policy optimization methods by explicitly separating the exploration and exploitation parts of the policy. First, some theory is developed to support this idea by generalizing the policy gradient to allow arbitrary state distributions. Then, experiments in small MDPs and in deep RL demonstrates its advantages empirically.



**Limitations And Societal Impact:**

Yes.

**Main Review:**


I find that the idea of explicitly separating exploration and exploitation makes sense for policy-based RL methods. I think the theoretical contributions are particularly interesting and I would have liked to see them developed further, especially with regards to doing updates with arbitrary $d(s)$ using samples.
Generally, the experiments were well-done and thorough although I have certain concerns about the deep RL section.
The writing was satisfactory overall (didn't spot any typos!) although there were a few parts I would have like to have clarified. The proofs were well-written and clear as far as I could tell.

Detailed comments:
- There is a line of work on off-policy actor-critic algorithms that I haven't seen mentioned [1,2,3], which feature different state-distribution weightings and analyze these algorithms theoretically and empirically.
While all state-distribution weightings are subsumed in a general $d(s)$ in this paper, I would have liked some discussion on what weighting should be used then.
Currently, C4 and C4-s are fairly loose constraints on $d_t$ and $\eta_t$. I think it would be interesting to look into this further and see what forms of d_t are theoretically advantageous, beyond simply covering all the states. In the experiments, a mix of the the on-policy state distribution and an exploratory policy's state distribution is used. Perhaps there could be some theory to support this choice?

One issue to consider would be the compatibility of a specific $d(s)$ with sampling. In particular, these previous algorithms [1,2,3] can usually be run in an online fashion (updating at every step or at every episode). In this paper, it hasn't been discussed much although there is the mention of experience-replay to help. Even with experience replay, there are many sampling strategies that could be used and it would be interesting if the theory could guide it.

[1] Off-Policy Actor-Critic, Degris et al. (2012),
[2] An Off-policy Policy Gradient Theorem Using Emphatic Weightings, Imani et al. (2018),
[3] Generalized Off-Policy Actoc-Critic, Zhang et al. (2019)

- I find theorem 2 to be a nice result since it gives necessary and sufficient conditions for arbitrary state distributions $d_t(s)$. I have various questions about the lines that follow and the interpretations of this result though.
- Line 107-110: Could you clarify how this is related to generalized policy iteration? I'm not sure how this theorem is helping towards a generalized policy iteration theorem or what this GPI theorem would say.
- Line 118-119: "the convergence propoerties of policy gradient and undiscounted updates require the same set of assumptions and conditions". Could you clarify this statement?
- Line 123: "prove that off-policy updates... are necessary to guarantee convergence to optimality". Why is this true? I don't see how this follows from the theorem. There are also convergence results for on-policy PG algorithms e.g. "On the Global Convergence Rate of Softmax Policy Gradient" Mei et al.
- Line 125-128: Although the initial state distribution doesn't have to cover the space, it isn't clear that this approach solves the exploration issue theoretically. If my interpretation is correct, the open problem outlined in the citation ("Optimality and Approximation..."Agarwal et al.) is not to relax A2 with any algorithm but with the standard policy gradient. That is, can we get the convergence rates for vanilla PG in the survey by Agarwal et al. without having A2? So, I don't think the claim of solving this open problem would be justified.

- Concerning condition C4 and theorem 3 (the convergence rate), why is that we can't just choose eta_t to be extremely large (even increasing over time)? Then we would satisfy C4 or C4-s easily for any d_t that is nonzero everywhere. Also, we would seemingly get the best convergence rate.

- In the deep RL experiments, J&H uses value-based methods for each part. This confuses me since the original motivation seemed to be to improve the behaviour of policy gradient algorithms. Hence, I would have expected, at least, Dr Jekyll (the exploitation agent) to be a policy-based algorithm. As presented currently, I don't really see any connection between the theory and the practical method proposed. For example, the point about the policy gradient being weighted by the stationary distribution doesn't apply to value-based methods anymore. Similarly, the point about policy-based methods having a slow-moving policy (lines 211-218) aren't relevant.
To me, this disconnect between the theory and implementation casts doubt on whether these experiments actually support the rest of the paper.

I appreciate that the other experiments were done with an actor-critic method though and those seem to be properly done. The thorough study of various parameters was a good addition.

- About the finite MDP experiments, I didn't see the performance of just Q-learning with exploration bonuses. How does that perform compared to J&H? In a similar vein, what would the performance of just Mr Hyde look like for these experiments? While Mr Hyde only seeks to explore perhaps that would be enough to do well in these domains?

- In the finite MDP experiments, how important is it to decay epsilon for the performance? All the runs for J&H decay epsilon over time, indicating that Jekyll gets chosen more often over time.

- Have you tried a version where Hyde (the exploratory agent) gets to choose all the actions and Jekyll only learns off-policy from the data? In principle, it seems like with the right replay mechanism, this could do well too.


**Time Spent Reviewing:**

8

---

> ### Author Response · Authors · 2021-08-09
> **Response to review**
>
> Thank you for your time and suggestions / comments. They will definitely help improve the paper.
>
> *Off-policy actor-critic line of work:* Thank you for pointing out those references, we will add and discuss them. These papers study the off-policy, continuing setting paired with the average value objective. That objective is either expressed with a state visitation coming from the behavioral policy (the excursion objective) or from the trained policy (the alternative life objective). Its gradient can be computed exactly or approximated and used to optimize the objective. The weights assigned to each state during an update stem from the gradient computation (exactly as in the discounted return gradient we study). It is indeed an interesting question to understand whether those weights have desirable properties. However, that question is far from trivial (at least to us), in particular because of the different objective being optimized. It is likely that a condition akin to C4 might exist in the continuing objective case, proving that would make an interesting follow-up paper. Also note that apart from Degris et al (for a discrete parametrization), no convergence proofs are given, only equivalents to the policy gradient theorem in the off-policy case. Finally, it is interesting to see that focusing solely on the gradient of the objective function (like those papers do) is not neither sufficient nor necessary, as we see in our results.
>
> *One issue to consider would be the compatibility of a specific  with sampling. In particular, these previous algorithms [1,2,3] can usually be run in an online fashion (updating at every step or at every episode). In this paper, it hasn't been discussed much although there is the mention of experience-replay to help. Even with experience replay, there are many sampling strategies that could be used and it would be interesting if the theory could guide it.*
>
> We agree that the sampling strategy is of prime importance for the efficiency of the algorithm. Note that our experiments in the planning setting (appendix D) empirically investigate what would be the best $d(s)$. The answer seems problem dependent, and might vary across time, but fortunately a good balance between off-policy (uniform distribution) and on-policy (current policy state visitation distribution) updates seems to be steadily competitive. In the sample setting, we did not want to overload the paper that we already find rather long, and decided to go simple with the dual experience replay buffer trick, that alone, guarantees to satisfy C4. But indeed, we would advise to improve the off-policy sampling strategy so that it conforms better to a uniform sampling and maximises $\min_s d(s)$, while prioritizing to some extent the on-policy states. In any case, much more should be done in this direction: it is a natural and exciting follow-up topic to try to go beyond the necessary and sufficient condition C4, and to understand better how to plan efficiently in a policy gradient landscape.
>
> *Th.2 and below:* We agree! From our perspective, Th. 2 is a powerful result.
>
> **L107-110:** For a learning rate $\eta$ going to infinity, update (3) corresponds to becoming greedy with respect to the current value function (and the algorithm becomes standard policy iteration). On the other hand, existing convergence results from e.g. Agarwal et al [2] (that Th.2 subsumes) need to assume small learning rates (as is customary with gradient methods). By allowing the use of arbitrary learning rates $\eta$ (as long as C4 is verified), Th.2 allows to bridge the gap between standard policy iteration (large $\eta$) and policy gradient methods (small $\eta$).
>
> With respect to GPI, Th.2 gives sufficient conditions on the policy improvement step. By combining it with a thorough theoretical analysis of the policy evaluation step (which we discuss in Section 3.1, in particular L169-176), we see a path towards a theorem describing conditions on both parts of GPI that guarantee convergence to optimality. We will make this clearer in the paper.
>
> **L118-119:** Th.2 shows convergence under condition C4. C4 encompasses standard policy gradient and the undiscounted update rule: convergence to optimality of both is guaranteed as long as C4 is verified. Conversely, it is possible to prove convergence to sub-optimal policies of either when C4 is violated. In other words, Th.2 allows to reduce the study of specific algorithms to the study of whether they verify C4. We will improve those two lines.
>
> **L123:** Mei et al [19] rely on Assumption A2, ie their results only apply to a certain class of MDPs for which the initial state distribution covers the whole state space. In particular, this implies that there cannot be off-policy state: they will all be visited in $\Theta(T)$ from initialization. Without A2, standard policy gradient converges to suboptimal policies in some hard planning tasks such as the chain environment from Section 4. The need for off-policy updates follows from the necessary part of Th.2. Without off-policy updates, it is possible to build counter-examples where C4 is not verified with policy gradients because the on-policy state-visitation density will vanish too fast in some off-policy states.
>
> **Line 125-128:** Indeed, your interpretation makes sense. For instance, it could be possible to search for a class of MDPs larger than the one verifying A2 where PG also converges (by e.g. showing that C4 is verified on that class). We will tone down the claim that we solve this open-problem and simply say that our theory could be used to make progress on it if that sounds reasonable?
>
> **C4 + Th3:** note that those are asymptotic convergence rates, so they only apply after a certain timestep $t_0$. $t_0$ depends on the learning rates used prior to it. In particular, larger learning rates early during training might increase it by e.g. focusing too much on a suboptimal action. In particular, our experiments have shown that a larger learning rate makes the algorithm converge too fast on suboptimal solutions, and that it then needs a very long time (in particular under the softmax parametrization) to "unlearn" this convergence. Our hyperparameter analysis shows that performance degrades past some value for the learning rate (see Figure 11(b)).
>
> **Deep RL experiment:** Dr Jekyll is an actor-critic / policy-based agent with the same architecture as e.g. SAC (ie an actor $\pi$ and a critic $q$). As you said, it wouldn't have made much sense otherwise. It is mentioned in the appendix, L1340 and Alg.4. We apologize for the confusion and will clarify this in the main text.
>
> *Finite MDP experiments:*
>
> **Q-learning with exploration bonus:** We have not been able to complete additional experiments yet, we hope to do so during the discussion period. Here are tentative answers based on the best of our knowledge:
> - We did not run value-based methods as we build on actor-critic methods, we do not claim anything about the opposition valued-based versus actor-critic algorithms.
> - We think the value based method might be more efficient on the chain environment where deep exploration is efficient and worse on the random MDP environment, where dithering exploration is efficient.
> - Mr Hyde never sees the environmental reward, only the exploration bonus, so it cannot learn any efficient policy (though pure exploration has been shown to perform decently in some tasks).
> - Note that we included experiments with an actor-critic algorithm where the critic is based on the Q-learning with exploration bonuses (that we call UCB critic). Their results are shown in Figure 2a-c with PG update for three different $\mu$ hyperparameters.
>
> **Decay of epsilon:** We decayed $\epsilon$ because we used the J&H mixture for evaluation and wanted to show its convergence to the optimal policy. Consequently, we needed to converge to a full exploitation behaviour over time. Had we used Jekyll only for evaluation (many papers actually report evaluation after removing the exploratory component of their algorithm), then we would have recommended a constant ratio. Nevertheless, as mentioned above, there are probably better scheduling schemes that would need to be investigated in future work.
>
> **Acting by Hyde-only:** Yes, in the planning experiments described in Appendix D, the full off-policy setting (where the policy update distribution is $d_u$) is studied. We will add mentions of this in the main text. We observe that it is most often slower than a trade-off between on-policy and off-policy (to be absolutely rigorous, the uniform distribution is already a trade-off between on-policy and off-policy). In the finite sample setting, we ran an experiment with various constant $\epsilon$ in the appendix (Figures 16 a-b), but we did not push to $\epsilon=1$ because, here too, we evaluated on the J&H mixture and obtaining a high score with $\epsilon$ too high is impossible. We could repeat this experiment with evaluation on Jekyll only.

---

> > ### Comment · Reviewer_T2tW · 2021-08-26
> > **Thank you for the response**
> >
> > Based on this response and the ones to the other reviews, I have a clearer picture of the contributions of this paper. After making the previously mentioned clarifications and changes to text, I think this paper would be a good addition to the existing literature. I've increased my score accordingly.
> >
> > Trying certain experiments with Jekyll for evaluation only could be an interesting addition. This would match the evaluation procedure in many deep RL papers with policy gradient methods and possibly simplifiy the tuning of epsilon (no decaying).
> >
> > I find that studying off-policy policy gradient algortihms is an interesting research direction and I would be interested in seeing some followup works. From the theory side, it could be interesting to analyze the effects of experience replay (i.e. an algorithm closer to the practical version of J&H). In that case, there would need to be a way to deal with the fact that updates from the replay buffer are not the same as updates on fresh samples. Perhaps as an easier step, one could analyze the expected (model-aware) update that uses a state-distribution consisting of a mixture of the on-policy and replay distributions.

---

> > > ### Author Response · Authors · 2021-08-28
> > > **Discussion 2**
> > >
> > > ***Trying certain experiments with Jekyll for evaluation only could be an interesting addition. This would match the evaluation procedure in many deep RL papers with policy gradient methods and possibly simplifiy the tuning of epsilon (no decaying).***
> > >
> > > We reproduced the constant exploration hyperparameter search experiment ($\epsilon_t$ ranging with 21 values from 0 to 1: 0, 0.05, 0.1,...,0.95,1) with evaluation on Jekyll solely, as you proposed it during the initial review. As we cannot share figures in the discussion and are not allowed to use any other means, we will describe with words our new findings below (sorry for the wordiness of it):
> > > - **chain experiment (size=10)** we report time to reach $\overline{\mathcal{J}}_\pi = 0.5$ with $o_t=0.5$ (decaying $o_t$ displays similar behavior with an overall significantly degraded performance):
> > >    - with $\epsilon_t=0$, unsurprisingly, the algorithm fails at finding the optimal reward and therefore finding the goal.
> > >    - from $\epsilon_t=0.05$ to $\epsilon_t=0.6$, the speed improves from 15000 to 6500.
> > >    - from $\epsilon_t=0.6$ to $\epsilon_t=0.95$, the speed remains more of less constant in the range [6400, 6500].
> > >    - with $\epsilon_t=1$, more surprisingly, the performance degrades quite significantly to 7300.
> > > - **random MDPs experiment (size=100)** we report time to reach $\overline{\mathcal{J}}_\pi = 0.95$ with $o_t=0.5$ (decaying $o_t$ displays similar behavior with an overall significantly degraded performance):
> > >    - from $\epsilon_t=0$ to $\epsilon_t=0.95$, the speed linearly improves from 35000 to 29000.
> > >    - with $\epsilon_t=1$, again surprisingly, the performance degrades catastrophically to more than 100000.
> > >    - we will report tomorrow the time to reach $\overline{\mathcal{J}}_\pi = 0.99$ if noteworthy. We expect that selecting Jekyll more often is going to be helping because this is where more samples are required to achieve such a precise near-optimality. (we might push to $\overline{\mathcal{J}}_\pi = 0.999$ to prove our point)
> > >
> > > We interpret these experiment results as a confirmation that more deep exploration leads to a faster identification of the optimal path, which helps in hard planning tasks, and that more dithering exploration leads to a faster statistical significance around the optimal path, which helps in strongly stochastic environments. Therefore, both should walk together hand in hand to obtain the best sample efficiency.
> > >
> > > This constant exploration hyperparameter search experiment is certainly more informative that the one we did at submission with evaluation on the J&H mixture that did not provide any insight besides the obvious statement "selecting Hyde hurts performance". We thank you for this idea and we intend to replace it in the final version. Do you think that some other experiments should be rerun with this evaluation?
> > >
> > > ***From the theory side, it could be interesting to analyze the effects of experience replay (i.e. an algorithm closer to the practical version of J&H). In that case, there would need to be a way to deal with the fact that updates from the replay buffer are not the same as updates on fresh samples. Perhaps as an easier step, one could analyze the expected (model-aware) update that uses a state-distribution consisting of a mixture of the on-policy and replay distributions.***
> > >
> > > Thank you for the suggestion. This is certainly something that could be worth investigating. However, we are more inclined to a methodology of building a theory (such as off-policy policy updates) that inspires the design of new algorithms (such as J&H), rather than the opposite: justify an algorithm with a theoretical analysis. So we are more targeting an algorithm implementing the uniform sampling of the state action set, that is more or less prescribed by the constants in our bounds.

---

### Official Review · Reviewer_622C · 2021-07-16

**Rating:** 6
**Confidence:** 3

**Summary:**

This paper studies the effects of off-policy updates for policy gradient methods in RL. The authors show that under certain conditions on the state update distribution and learning rate, exact PG methods will converge to the optimal solution. Based on this result, the authors propose an RL algorithm where the policy is trained on states from two distributions: 1) the on-policy data distribution and 2) the distribution of an "exploratory" policy. This method is demonstrated on two exceedingly simple tasks.


**Main Review:**

Edit: I have updated my score following the author rebuttal. Please see my remaining concerns in my comment to the author rebuttal.


It is difficult to ascertain the significance of the results / ideas in the paper.

The claimed primary contribution seems to be the convergence of Eq 3 under some step-size schedule; however, this is not particularly novel and seems to follow quickly from the policy improvement theorem and standard results in stochastic approximation (unless I am misunderstanding the result).

All of the results are exclusively in the fully-parameterized regime with exact access to Q functions; while in this regime, off-policy updates do converge to the optimal policy, this breaks as soon as either condition is broken, for example when dealing with arbitrary function classes F or imperfect estimates of Q (e.g. approximate policy iteration). Given that both of these things hold when applying RL on any non-toy problem, I am not convinced about the relevance of the ideas / results derived in the paper.

It would be very interesting to see analysis as to how quickly (or slowly) these results break under these more realistic assumptions. I think something of this form would strengthen the paper significantly.

It would be useful to explicitly highlight why using the on-policy distribution is bad based on the formal results in your paper. While it is clear intuitively that the on-policy distribution will visit some states sparingly, it is not clear from the text to what degree this slows down convergence.

This paper proposes J&H as a mechanism to ensure that the policy is updated on a wide state distribution instead of the narrow on-policy distribution -- why should this mechanism be preferred over the standard mechanism of using a replay buffer to achieve the same effect? The replay buffer is a very common mechanism, and used with almost all recent practical actor-critic methods (e.g SAC, DQN)

The current experiments in the chain / random MDP / 4 room domains are in very toy settings (all either completely tabular, or in general, no need for generalization). Echoing my previous comment earlier, while these results do affirm the theoretical results, they provide very little intuition for how this kind of method will do on more standard domains that policy gradient and actor critic methods are applied to (as an example, Gym or Atari). For example, I imagine in many domains of interest with large state spaces, a dithering based strategy will perform better than this decoupled exploration / exploitation behavior, because the number of interesting states is simply dwarfed by the number of less useful states.

Minor notes:
- Is a formal definition of $\delta(s)$ provided in the main text?

**Time Spent Reviewing:**

3

---

> ### Author Response · Authors · 2021-08-09
> **Response to review**
>
> Thank you for the time you spent on the paper. Below we reply to your various comments.
>
> *The claimed primary contribution seems to be the convergence of Eq 3 under some step-size schedule; however, this is not particularly novel and seems to follow quickly from the policy improvement theorem and standard results in stochastic approximation (unless I am misunderstanding the result).*
>
> We would like to clarify that the contribution is not only about the step-size schedule but mostly about the state visitation density. Indeed, we do consider Th.2 as our primary contribution (though the improved asymptotic rates also have value). Previous work needed the strong assumption (A2) on the nature of the task in order to resolve the planning issue of policy gradient. We relax this assumption (our result is applicable to any finite MDP), and define a necessary and sufficient condition on the algorithm for guaranteed convergence to the optimal policy.
>
> Th.2 is far from direct and takes us several pages of formal proofs. If you see a shortcut proving this result in a trivial way, please share it with us. Also note that the convergence proof for the softmax in Agarwal et al [2] (and Mei et al [19], though they give convergence rates too) is rather lengthy and tricky (9 pages). They have to make Assumption A2 which guarantees that C4 is verified. A simple proof of our result would de facto extend and trivialize theirs.
>
> In terms of novelty, while condition C4 is classic from a stochastic approximation perspective, it is nevertheless the first time (to the best of our knowledge) it is stated in this context. It also offers a fresh perspective on actor-critic algorithms and various tricks used in practice to improve their performance (undiscounted updates, experience replay, etc), as well as suggesting avenues for future developments.
>
> *All of the results are exclusively in the fully-parameterized regime with exact access to Q functions [...] strengthen the paper significantly.*
>
> We agree that finite sample analysis and generalization are important matters and natural follow-up questions to our work. They are nevertheless beyond the scope of this paper (already 49 pages long) which introduces a framework subsuming existing results, and that we consider novel and promising. We have tried to make the submission as clear as possible about our focus and its limitations, but we would be happy to clarify it further if you have any suggestions.
>
> We would also like to stress that Mei et al [19] was published at ICML 2020 and is a theoretical paper in the tabular setting with known Q-function. In contrast, beyond our theoretical contributions, we introduce a novel algorithm and our experiments empirically validate the approach in the finite sample setting.
>
> *It would be useful to explicitly highlight why using the on-policy distribution is bad based on the formal results in your paper. While it is clear intuitively that the on-policy distribution will visit some states sparingly, it is not clear from the text to what degree this slows down convergence.*
>
> A short explanation can be found on lines 28-32, and a longer one in Appendix A.5. In a gist, without extra assumptions (e.g. A2), on-policy updates do not satisfy condition C4 because in that case $d_t$ will decrease too fast. We will expand this point in the main text.
> In addition, this limitation does not simply slow down convergence, it fundamentally breaks it in hard planning tasks like the chain environment (simple in appearance but already too difficult for standard policy gradient). On the chain, PG never finds the optimal policy and instead converges to a suboptimal one.
>
> *This paper proposes J&H as a mechanism to ensure that the policy is updated on a wide state distribution instead of the narrow on-policy distribution -- why should this mechanism be preferred over the standard mechanism of using a replay buffer to achieve the same effect? The replay buffer is a very common mechanism, and used with almost all recent practical actor-critic methods (e.g SAC, DQN).*
>
> First, note that DQN is not an actor-critic method.
>
> Second, using a replay buffer is not sufficient: as stated by condition C4, it needs to appropriately cover the state-action space, which has no reason to happen without a forcing mechanism. In J&H, it is the role of Mr Hyde.
>
> Finally, most actor-critic algorithms with experience replay reweight the samples to fit the on-policy distribution, which we prove to be counterproductive (or at least not particularly well motivated apart from matching the policy gradient update). More general sample reweightings (aka prioritized replay) may be used to enforce C4. Lines 194-198 discuss it.
> We would like to add that we do not claim novelty on (prioritized) replay buffer, expected policy gradient, nor deep exploration. Rather, we use these well- known tools to satisfy the conditions prescribed by our analysis, and in doing so, we provide a principled way justifying their application to actor-critic algorithms.
>
> *For example, I imagine in many domains of interest with large state spaces, a dithering based strategy will perform better than this decoupled exploration / exploitation behavior, because the number of interesting states is simply dwarfed by the number of less useful states.*
>
> Dithering exploration can indeed be valuable is such environments. Our point is different though: we state (and we are not the first ones to make this point) that it is insufficient to ensure full coverage of the state-action space. Actor-critic algorithm are naturally efficient at dithering exploration and bad at deep exploration. Our analysis allowing and advocating for off-policy updates allows us to design an algorithm combining both.
>
> *Minor note:* $\delta(s)$ is introduced informally on line 138, we will add a proper definition.
>
> [2] Alekh Agarwal, Sham M Kakade, Jason D Lee, and Gaurav Mahajan. Optimality and approximation with policy gradient methods in markov decision processes. COLT 2020.
>
> [19] Jincheng Mei, Chenjun Xiao, Csaba Szepesvari, and Dale Schuurmans. On the global convergence rates of softmax policy gradient methods. ICML 2020.

---

> > ### Comment · Reviewer_622C · 2021-08-23
> > **Thank you for the clarifications**
> >
> > Thank you for the clarifications! I have read your response, and gone back to the paper to try to re-evaluate my original concerns with the paper. If there is no response to a particular point, you may consider that point addressed. Holistically, I am more positive in my assessment of the paper and will update my score appropriately.
> >
> > I now understand Theorem 2 better, and it is clear to me that the result is not simply an application of stochastic approximation in the off-policy policy gradient setting.
> >
> > _"A simple proof of our result would de facto extend and trivialize theirs."_
> >
> > I did not mean to imply that a freestanding proof of Theorem 2 would be simple. Rather, my concern was that it was unclear to me how different the proof needed to be from that of Theorem 5.1 in [Agarwal et al], which provides a proof of convergence for softmax policy gradients. My understanding of [Agarwal et al] is that while their theorem is stated for the specific technical assumption of A2, the proof used works under more generic assumptions on the state distribution (for example, maybe something of the form $\exists c \forall s: d_t(s) > c > 0$). I would appreciate if an updated version of the manuscript clarified why the proof of Agarwal et al does not transfer, and what needs to be accounted for in order to strengthen the proof to only require C.4.
> >
> > _"All of the results are exclusively in the fully-parameterized regime with exact access to Q functions [...] strengthen the paper significantly."_
> >
> > My general concerns about the brittleness of this line of work to the assumptions of tabular parameterizations and exact Q functions still remain. I agree that the paper already has a lot of technical content, and it would be perhaps unreasonable to ask for formal analysis of the necessity of these assumptions for this current manuscript. Nonetheless, I would encourage the authors to, in lieu of formal analysis, at least include discussion in the paper that clarifies the potential limitations of such a "off-policy" approach in the presence of function approximation, where performing PG updates using off-policy data distributions in general does not lead to optimal solutions.

---

> > > ### Author Response · Authors · 2021-08-24
> > > **Thank you for your response**
> > >
> > > We really appreciate you took the time to re-evaluate our submission and to give us another feedback. We are really enthusiastic about our submission and would like to make it the most accurate and complete possible. Thank you for your help to this end. We try and address your latest concerns/suggestions below:
> > >
> > > ***I would appreciate if an updated version of the manuscript clarified why the proof of Agarwal et al does not transfer, and what needs to be accounted for in order to strengthen the proof to only require C.4.***
> > >
> > > Thank you for the suggestion, we will add a discussion highlighting the following differences.
> > > - Agarwal et al study gradient ascent, and as such, can rely on standard optimization results. They do so in two parts of their proof:
> > >    - Their Lemma C2, which uses the strong convexity of the objective function to prove that following the gradient results in a value improvement (assuming a learning rate sufficiently small). We provide a more general (we do not need the condition on the learning rate) and, in our admittedly biased opinion, more elegant proof of that lemma in our Theorem 1.
> > >    - Their Lemma C5, which uses (i) the convergence of the gradient to 0 (a standard result with gradient as/des-cent), and (ii) the assumption that all states have a non-vanishing density to infer that, in the limit, the learnt policy does not assign any mass to states that have a non-zero advantage. Neither (i) nor (ii) hold in our setting and we had to leverage our condition C4 instead. It is not quite possible to pinpoint a specific part of our proof showing that particular result as we took a different road, but it was non-trivial to bypass.
> > > - Indeed, their proof can be extended to policy densities lower-bounded by a strictly positive constant. However, this still only applies to the case of on-policy policy gradient. Also, guaranteeing the lower-boundedness for all policies along the optimization trajectory (ie uniform over t) is very hard to do on-policy (which is why A2 is usually made). Moving off-policy and verifying C4 via Jekyll&Hyde is our way to bypass these difficulties.
> > >
> > > Also, note that we provide improved asymptotic rates compared to Mei et al (Agarwal et al do not provide any for the softmax). We discuss the difference extensively in Appendix A6, and will move part of that discussion to the main text.
> > >
> > > ***I would encourage the authors to, in lieu of formal analysis, at least include discussion in the paper that clarifies the potential limitations of such a "off-policy" approach in the presence of function approximation, where performing PG updates using off-policy data distributions in general does not lead to optimal solutions.***
> > >
> > > We would be happy to strengthen our coverage of the potential adverse effect of off-policy updates.
> > > - **critic estimation:** The paragraph at lines 169-176 provides some intuition regarding the setting with an estimated critic. Since the submission, we have finished the analysis for a critic learned with SARSA. And indeed, while it is still in the tabular case, it required new technical tools and 30 pages of proofs. Our new results confirm what we claimed, but they required significant effort.
> > > - **function approximation:** The paragraph at lines 177-183 discusses [Nota2020]. They show that under state aliasing, using off-policy updates can lead to strongly suboptimal solutions as you said. We will emphasize it in the submission. Nevertheless, we believe that state aliasing is a worst-case scenario that should not happen with neural networks, thanks to their very high expressive capacity. Also note that this concern with respect to distribution shift is general and could be formulated for any neural model, including supervised models, or in RL the purely value-based ones such as DQN that are frequently trained off-policy. The consensus in the literature is that neural models do not suffer too much from distribution shift as long as the testing set distribution is well covered by the training set. Since appropriate coverage of the state-action space is actually the final objective of our off-policy policy updates, we expect minimal impact on this dimension, though it does remain to be formally demonstrated.
> > >
> > > Did you have something else in mind regarding the potential limitations?
> > >
> > > Thanks again for your time,

---

### Decision · Program_Chairs · 2021-09-27

**Decision:**

Accept (Poster)

**Comment:**

The reviewers discussed this paper and came to a consensus that it is a useful contribution and should be accepted.

There were concerns about the fact that this work is primarily in the tabular setting and novelty relative to existing work. The restriction to the tabular setting is quite limited, in that convergence behavior is known to be quite different for (biased) policy gradient methods in the tabular setting (namely that convergence is still guaranteed, as opposed to under function approximation where there are known counterexamples). An issue raised by a reviewer is in the omission of previous work for off-policy policy gradients, that highlight some of these issues, and that consider a different behavior distribution in the objective (somewhat similarly to what is done here, though it is less general since it is restricted to the distribution under a fixed behavior). This connection should be discussed, and the issue of potential divergence more clearly highlighted. The authors do acknowledge this in the paper, refering to the work on biased actor-critic, but it is somewhat buried. This paper would be improved with a much more explicit discussion about this, and if or how the authors think these results might extend to function approximation.

As mentioned, despite some of these issues, the insights provided, particularly about convergence rates and relationships to the state weightings, are useful.